# Node Similarities under Random Projections: Limits and Pathological Cases

**Tvrtko Tadić**
Microsoft Search, Assistant and Intelligence
tvrtkota@microsoft.com

**Cassiano O. Becker**
Microsoft Search, Assistant and Intelligence
casbecker@microsoft.com

**Jennifer Neville**
Microsoft Research
jenneville@microsoft.com

## Abstract

Random Projections have been widely used to generate embeddings for various graph learning tasks due to their computational efficiency. The majority of applications have been justified through the Johnson-Lindenstrauss Lemma. In this paper, we take a step further and investigate how well dot product and cosine similarity are preserved by random projections when these are applied over the rows of the graph matrix. Our analysis provides new asymptotic and finite-sample results, identifies pathological cases, and tests them with numerical experiments. We specialize our fundamental results to a ranking application by computing the probability of random projections flipping the node ordering induced by their embeddings. We find that, depending on the degree distribution, the method produces especially unreliable embeddings for the dot product, regardless of whether the adjacency or the normalized transition matrix is used. With respect to the statistical noise introduced by random projections, we show that cosine similarity produces remarkably more precise approximations.

## 1 Introduction

*Random projections* (RP) provide a simple and elegant approach to dimensionality reduction (Vempala, 2004). Leveraging concentration of measure phenomena in high-dimensional statistics (Vershynin, 2018), RP's rely on the well-known Johnson-Lindenstrauss (JL) lemma (Johnson et al., 1986; Dasgupta & Gupta, 2003) to provide data-independent guarantees for approximation quality. As originally developed, the JL lemma showed that Euclidean distances between any two points in a dataset are preserved with high probability if their vectors are projected using appropriately constructed random matrices. Remarkably, the JL lemma shows that the required projection dimension of the random matrix needs to grow only with the logarithm of the ambient dimension of the dataset. As such, the JL lemma has found wide application in diverse fields such as information retrieval (Kleinberg, 1997; Indyk & Motwani, 1998), machine learning (Nachum et al., 2022; Durrant & Kabán, 2013; Cannings & Samworth, 2017; Boutsidis et al., 2010; Cardoso & Wichert, 2012; Makarychev et al., 2019), privacy (Liu, 2005; Wang & Plataniotis, 2010), and numerical linear algebra (Halko et al., 2011; Martinsson & Tropp, 2020).

*In the context of graphs*, the JL lemma can be applied to generate low-dimensional encodings of a node's connectivity to other nodes in the graph. In that case, nodes are usually represented as vectors extracted as the rows of a connectivity matrix (Hamilton, 2020), which is obtained from the adjacency matrix (or some analytic function of it). For example, it is known that the entries of the power of an adjacency matrix encode the number of walks of length of that prescribed power between any two nodes (Newman, 2018). Likewise, the transition matrix can be used to compute the corresponding random-walk probability between any pair of nodes (Bianchini et al., 2005). This notion is key to algorithms such as PageRank (Page et al., 1999), which computes the steady-state random walk probability as the limit of a weighted sum over all possible lengths. In these cases, although the adjacency matrix is typically sparse –in that it registers only first-order immediate neighbors– higher-

Table 1: Notation and symbols used throughout the paper

| $n$ | number of nodes | $c, \gamma$ | low and high degree parameters |
|---|---|---|---|
| $G$ | graph with $V$ and $E$ | $L_c$ | set of low degree nodes wrt. $c$ |
| | the sets of nodes and edges | $H_c^\gamma$ | set of high degree nodes wrt. $c$ and $\gamma$ |
| $A, T$ | adjacency matrix, transition matrix | $q$ | random projection dimension |
| $P$ | matrix polynomial $p(A)$ or $p(T)$ | $R$ | random projection matrix |
| $d_u$ | degree of node $u$ | $X$ | node embedding matrix |
| $n_{uv}$ | connectivity between nodes $u$ and $v$ | $\mathcal{T}_q$ | $t$-distribution with parameter $q$ |

order connectivity matrices will often become dense and pose representation challenges in term of the storage space required. We recall that the inherent dimensionality of such representations based on connectivity matrices would otherwise grow quadratically with the number of nodes in the graph, which in practical data applications is often of the order of millions or higher (Tang et al., 2009). To that end, random projections consists of a promising dimensionality reduction approach. In the context of representation learning, the use of random projections to generate node embeddings for graphs was introduced in (Zhang et al., 2018) and further developed in (Chen et al., 2019), becoming a popular and fast way to produce embeddings.

*The problem addressed in this paper* is the one of graph representation for tasks over large graphs, and can be summarized by the following steps:

1. We use random projections to generate *low-dimensional node embeddings* capturing (high order) connectivity induced by (otherwise computationally-expensive) adjacency matrix polynomials.

2. With the node embeddings, we compute arbitrary *pairwise node similarities* as a feature in models for graph inference tasks (e.g., ranking for recommendation).

*The contributions of this paper* as threefold:

1. We show that the *degree of the node has significant influence on the quality of approximation similarity*, according to the type of matrix (e.g., adjacency or transition) and function adopted (e.g., dot product or cosine). By expressing the JL Lemma in terms of the graph's degree distribution, we show that it becomes data dependent for dot product, yielding especially weak guarantees for low and high-degree nodes. Contrary to intuition, this happens not only for the adjacency matrix, but also when the transition matrix is considered (i.e., when the matrix is row-normalized).

2. We provide extensive analyses of node embedding approximation quality under cosine similarity. By crafting a rotation argument followed by a Gram-Schmidt orthogonalization procedure, we develop *novel asymptotic and finite-sample results* that show that cosine similarity produces more precise approximations with respect to the graph's degree distribution.

3. We extend our theoretical results to the case of *ranking based on node embeddings*. Specifically, we derive the probability of random projections flipping the node ordering induced by their embeddings, and show that cosine similarity produces remarkably more stable rankings. These findings are shown to occur and impact practical applications, as we illustrate with an example using a Wikipedia dataset.

*Related work* on random projections has focused predominantly on proposing alternative random projection matrix constructions and seeking sparser configurations (Freksen et al., 2018; Jagadeesan, 2019; Achlioptas, 2003). Other works were dedicated to obtaining tighter bounds for the preservation of Euclidean distance (Sobczyk & Luisier, 2022; Li et al., 2006). In the case of the dot product, work in (Kaban, 2015; Vempala, 2004) obtained general guarantees depending on the angle between the vectors. In the case of cosine similarity, other guarantees have been provided in (Arpit et al., 2014). In this paper, we refine and improve some of those results. Other studies have focused on the preservation of margin in the formulation of support vector machines (Shi et al., 2012). Additional contributions for the analysis of clustering have been given by (Becchetti et al., 2019; Cohen-Addad et al., 2022; Bucarelli et al., 2024).

## 2 RANDOM PROJECTIONS IN GRAPHS

In the case of large graphs $G = (V, E)$, random projections have been applied to polynomials of the nonnegative integer[1] adjacency matrix $A$ and transition matrix $T$. If $R = [R_{ij}] \in \mathbb{R}^{q \times n}$, $q \ll n$, is a random matrix generated in a specific way, then $X = p(A)R^\top$ or $X = p(T)R^\top$ preserves many of the properties $p(A)$ and $p(T)$ for a matrix polynomial $p(Y) = \sum_{l=1}^m \alpha_l Y^l$. In our paper, we construct $R$ by sampling $(R_{ij} : i = 1, \ldots, q, j = 1, \ldots, n)$ as i.i.d. normal random variables with mean zero and variance $1/q$. A sequence of row vectors $p_1, \ldots, p_k$ in $\mathbb{R}^{1 \times n}$ is then mapped to row-vectors $p_1 R^\top, \ldots, p_k R^\top$ in $\mathbb{R}^{1 \times q}$. An additional benefit is that $p(A)R^\top$ and $p(T)R^\top$ can be calculated faster than $p(A)$ and $p(T)$, since a given factor $A^l R^\top$ is calculated as $A(\ldots(A(AR^\top))$, with $A$ sparse and $q \ll n$. In contrast, $A^l$ computed separately may incur an $n \times n$ full-dimensional and dense $A^l$ for higher $l$. This property becomes even more useful in the case of heterogeneous graphs, where we have polynomials of several variables associated with the so-called metapaths (Sun et al., 2011).

Let $P = p(A)$ or $P = p(T)$; and denote $P_{w*}$ as the row in the matrix $P$ representing node $w \in V$. For $u, v \in V$, the interpretation is that $\mathrm{rel}_{uv} := P_{v*} P_{u*}^\top$ is the relevance between $u$ and $v$ and

$$\mathrm{rel}_{uv}^R := X_{v*} X_{u*}^\top \approx P_{v*} P_{u*}^\top. \tag{1}$$

When (1) holds, a vertex $v$ in the graph can be effectively represented by the embedding $X_{v*}$. In the finite-sample regime, this approximation is generally justified by the JL Lemma. The following version is from the book (Boucheron et al., 2013):

**Theorem 2.1.** *Let $\varepsilon, \delta \in (0, 1)$, and $p_1, \ldots, p_k$ be non-zero vectors in $\mathbb{R}^{1 \times n}$. If $q \geq \frac{4}{\varepsilon^2} \log \left[ \frac{k^2}{\delta} \right]$ then, with probability at least $1 - \delta$, the inequality $(1 - \varepsilon) \|p_i - p_j\|^2 \leq \|p_i R^\top - p_j R^\top\|^2 \leq (1 + \varepsilon) \|p_i - p_j\|^2$ holds for all $i < j$.*

Due to the randomness of $R$, we will show that, for a large sparse graph, the assumption (1) will fail in two important cases:

*(i) $P = A$*: for a low-degree node $v$, $X_{v*} X_{u*}$ will overvalue $P_{v*} P_{u*}^\top$ for many high-degree nodes $u$;

*(ii) $P = T$*: for a high-degree node $u$, $X_{v*} X_{u*}$ will overvalue $P_{u*} P_{v*}^\top$ for many low-degree nodes $v$.

In contrast, we will show that $\frac{X_{v*} X_{u*}^\top}{\|X_{v*}\| \|X_{u*}\|} \approx \frac{P_{v*} P_{u*}^\top}{\|P_{v*}\| \|P_{u*}\|}$ holds more consistently. This is why we propose to use $\frac{X_{v*}}{\|X_{v*}\|}$ as the embedding for a vertex $v$. We call this method *RP Cosine Similarity*.

To develop our specific results, we begin by denoting, for all $u, v \in V$, $d_u := \sum_{w \in V} A_{uw}$ as the *degree* of the vertex $u$. Likewise, we define $n_{uv} := A_{u*} A_{v*}^\top$ as the *2-hop connectivity* between $u$ and $v$, corresponding to the number of paths of length two between those vertices. The following lemma provides basic information about $n_{uv}$.

**Lemma 2.2.** *For all $u, v \in V$ we have*

$$d_u \leq n_{uu} \leq d_u^2; \tag{2} \qquad\qquad \frac{n_{uv}}{d_u d_v} = T_{u*} T_{v*}^\top. \tag{3}$$

*Proof.* Since all entries of the matrix $A$ are nonnegative integers: $n_{uu} = \sum_{w \in V} A_{uw}^2 \geq \sum_{w \in V} A_{uw} = d_u$. On the other hand, $d_u^2 = (\sum_{w \in V} A_{uw})^2 \geq \sum_{w \in V} A_{uw}^2 = A_{u*} A_{u*}^\top = n_{uu}$. Note, $T_{u*} = \frac{1}{d_u} A_{u*}$ for all $u \in V$. Hence, $T_{u*} T_{v*}^\top = \frac{A_{u*} A_{v*}^\top}{d_u d_v} = \frac{n_{uv}}{d_u d_v}$. This shows (2) and (3). □

Further, we note that $d_u$ has a linear dependence on the entries of $A$, while $n_{uv}$ has a quadratic one. Thus, in order to relate $d_u$ to $n_{uv}$, we define the following graph-wide property:

$$\gamma := \max_{u, v \in V} \frac{n_{uv}}{d_v}. \tag{4}$$

---

[1]Although we assume throughout the paper that the elements of the adjacency matrix are nonnegative integers, all the proofs would work if entries were from the set $\{0\} \cup [1, \infty)$. The results can also be further modified to allow the elements of $A$ to be from $[0, \infty)$ by introducing additional constants.

This property enables our results to generalize to weighted adjacency matrices $A \in (\mathbb{Z}_0^+)^{n \times n}$. In general, from (2), we have $1 \leq \gamma \leq \max_{u,v \in V} A_{uv}$. In particular, if $A \in \{0,1\}^{n \times n}$, then $\gamma = 1$.

Next, we will define what does it mean for the node to be of *high* or *low* degree in the context of this paper. Let

$$L_c := \{v \in V : d_v \leq c\}. \tag{5}$$

This will be called the set of nodes of low degree. In particular, for a graph following a power law distribution, $L_c$ will contain the majority of the nodes. Conversely, let

$$H_c^\gamma := \{u \in V \ : \ d_u \geq \gamma^2 cq\}. \tag{6}$$

We will call this the set nodes of high degree. We are interested analyzing graphs where $n = |V|$ is large using results of the Johnson-Lindenstrauss type. For that, we consider $q = \mathcal{O}(\log n)$. Using the power law distribution as described in equation (10.8) from (Newman, 2018), which is often observed in natural and social networks, we have $\frac{|\{u:d_u \geq z\}|}{n} \sim Cz^{-\alpha}$, where $\alpha > 0$. Consequently, $\frac{|H_c^\gamma|}{n} \sim C[\log n]^{-\alpha}$. For large $n$, the set $H_c^\gamma$ should contain a number of nodes on the order of $\mathcal{O}(n[\log n]^{-\alpha})$.

## 2.1 RP Dot Product when $P = T$

To avoid numerical instability issues associated with higher degree nodes and higher powers of $A$, practitioners often consider the transition matrix $T$, since it consists of a bounded, row-stochastic normalization of $A$. We will show that even in this case, random projections may yield especially poor approximations for nodes $u \in H_c^\gamma$ and $v \in L_c$.

**Theorem 2.3.** *Let $X = TR^\top$. The following statements hold:*
*(a) Asymptotic result. For $u,v \in V$ and large $q$*

$$X_{u*}X_{v*}^\top \overset{a}{\sim} \mathcal{N}\left(\frac{n_{uv}}{d_u d_v}, \frac{1}{q}\left[\frac{n_{uu}n_{vv}}{d_u^2 d_v^2} + \left(\frac{n_{uv}}{d_u d_v}\right)^2\right]\right), \tag{7}$$

*(b) Finite-sample result. For $\varepsilon \in (0,1)$ and $\delta \in (0,1)$, if $q \geq 4\frac{1+\varepsilon}{\varepsilon^2}\log\left[\frac{n(n-1)}{\delta}\right]$, then*

$$|X_{u*}X_{v*}^\top - \frac{n_{uv}}{d_u d_v}| < \varepsilon\frac{\sqrt{n_{uu}n_{vv}}}{d_u d_v} \tag{8}$$

*for all $u,v \in V$ holds with probability at least $1 - \delta$.*

*Remark.* We denote by $Y_q \overset{a}{\sim} \mathcal{N}(\mu, \frac{\sigma^2}{q})$ the fact that $\frac{Y_q - \mu}{\sigma}\sqrt{q} \overset{d}{\to} \mathcal{N}(0,1)$ as $q \to \infty$. The "$\overset{a}{\sim}$" is interpreted as meaning an approximate distribution for large $q$.

*Proof.* Using the rotation argument from §A.II, we represent $X_{u*}X_{v*}^\top$ as sum of $q$ i.i.d random variables and a convenient term. This is stated in part (a) of Theorem A.6 in the Appendix.

Using the Central Limit Theorem and Law of Large Numbers, we get the following asymptotic result: $T_u R^\top R T_v^\top \overset{a}{\sim} \mathcal{N}(T_u T_v^\top, (T_u T_v^\top)^2/q + \|T_u\|^2\|T_v\|^2/q)$. For details, see part (a) of Proposition A.1 in the Appendix. Applying (3) we have $T_u T_v^\top = \frac{n_{uv}}{d_u d_v}$, $\|T_v\|^2 = T_v T_v^\top = \frac{n_{vv}}{d_v^2}$ and $\|T_u\|^2 = T_u T_u^\top = \frac{n_{uu}}{d_u^2}$ and part (a) follows.

Using the representation $X_{u*}X_{v*}^\top$ mentioned at the beginning of the proof, we can produce concentration results that lead to a JL Lemma-type result in part (b) of Proposition A.1 in the Appendix. From there, we set $n = k = |V|$ and get that $|T_u R^\top R T_v^\top - T_u T_v^\top| < \varepsilon\|T_v\|\|T_u\|$ for all $u,v \in V$ under the same conditions and assumptions of part (b) of this Theorem. The full claim follows from the calculation we did to prove part (a). $\square$

The rotation argument mentioned above (and fully developed in §A.II), allows us to directly analyze $P_{u*}R^\top R P_{v*}^\top$. This improves on the typical approach (Vempala, 2004; Kaban, 2015; Amirov et al., 2023) of representing the dot product by the expansion $\frac{1}{4}(\|P_{u*}R^\top + P_{v*}R^\top\| - \|P_{u*}R^\top - P_{v*}R^\top\|)$, since these factors are usually not independent. Doing so enables us to produce more precise bounds, as discussed in §A.III.5.

The asymptotic result (7) is often missed when using random projections in graphs. As mentioned, versions of (8) are often used. However, they miss telling us when and how often the method will produce poor estimates. This does not seem to have been addressed in the literature. To do so, we will fix $u \in H_c$ and look at the values $\{X_{u*}X_{v*}^\top : v \in L_c\}$ that may misapproximate $\{\frac{n_{uv}}{d_u d_v} : v \in L_c\}$.

**Corollary 2.4.** *If $v \in L_c$ and $u \in H_c^\gamma$, then the standard deviation in (7) is greater than its expectation, i.e.:*

$$\frac{n_{uv}}{d_u d_v} \leq \sqrt{\frac{1}{q}\left[\frac{n_{uu}n_{vv}}{d_u^2 d_v^2} + \left(\frac{n_{uv}}{d_u d_v}\right)^2\right]}. \tag{9}$$

*Proof.* We have $\frac{n_{uu}n_{vv}+n_{uv}^2}{q} \overset{(2)}{\geq} \frac{d_u d_v + n_{uv}^2}{q} \overset{u \in H_c}{\geq} \frac{\gamma^2 c q d_v + n_{uv}^2}{q} \geq \gamma^2 c d_v \overset{v \in L_c}{\geq} (\gamma d_v)^2 \overset{(4)}{\geq} n_{uv}^2$. Multiplying the last inequality with $d_u^{-2}d_v^{-2}$ and taking the square root, (9) follows. $\square$

Under the conditions of the Corollary above, when can generate the one-sigma interval $\left[\frac{n_{uv}}{d_u d_v} - \sqrt{\frac{1}{q}\left[\frac{n_{uu}n_{vv}}{d_u^2 d_v^2} + \left(\frac{n_{uv}}{d_u d_v}\right)^2\right]}, \frac{n_{uv}}{d_u d_v} + \sqrt{\frac{1}{q}\left[\frac{n_{uu}n_{vv}}{d_u^2 d_v^2} + \left(\frac{n_{uv}}{d_u d_v}\right)^2\right]}\right]$ in which $(X_{u*}X_{v*}^\top : v \in L_c)$ will take values with probability of less than 69%. This means that getting a value outside that interval is not unlikely. In that case, we know from (9) that such values will either double $\frac{n_{uv}}{d_u d_v}$ or be less than $0$, which is clearly a poor approximation.

We now interpret the finite-sample result, and set $q = \lceil (1+\varepsilon)/\varepsilon \log n(n-1)/\delta \rceil$ such that part (b) of Theorem 2.3 holds. The expression $n_{uv}/(d_u d_v)$ in the numerator of (8) depends on $d_u$ with order $(d_u)^{-1}$, and denominator has a lower-order dependence $(d_u)^{-\frac{1}{2}}$, as described in Lemma B.4 in the Appendix. Thus, higher values of $d_u$ lead to an unfavorable regime in which the guarantees from Theorem 2.3 become increasingly weaker. As a simple numerical illustration, we assume $A \in \{0,1\}^{n \times n}$ so that $n_{uu} = d_u$ and $n_{vv} = d_v$, and let $\varepsilon = 10^{-2}$, $d_u = 10^7$, $d_v = 10$ and $n_{uv} = 1$. Then, (8) yields $X_{u*}X_{v*}^\top \in 10^{-8}(1-100, 1+100)$, which is clearly a severely biased estimate.

Similar results hold in the case $P = A$. For this case, we provide analogous asymptotic normality and finite sample results in §B.I of the Appendix. There, we also express similarities in the terms of values of $(n_{uv})$ and $(d_u)$. In the absence of normalization, the disruption of the assumption $X_{u*}X_{v*}^\top \approx P_{u*}P_{v*}^\top$ is more obvious.

## 2.2 RP COSINE SIMILARITY

In this section, we present asymptotic normality and finite-sample analyses to show how RP Cosine Similarity produces significantly better approximations than those of RP Dot Product for vertices $u \in H_c^\gamma$ and $v \in L_c$ (both when $P = T$ and $P = A$). We begin by showing that it will not matter for RP Cosine Similarity if we take $P = A$ or $P = T$.

**Lemma 2.5.** *For all $u, v \in V$ we have*

$$\cos(T_{u*}R^\top, T_{v*}R^\top) = \cos(A_{u*}R^\top, A_{v*}R^\top) \quad and \quad \cos(T_{u*}, T_{v*}) = \cos(A_{u*}, A_{v*}) = \frac{n_{uv}}{\sqrt{n_{uu}n_{vv}}}.$$

*Proof.* First equality follows from $\frac{T_{u*}RR^\top T_{v*}^\top}{\|T_{u*}R^\top\|\|T_{v*}R^\top\|} = \frac{(d_u^{-1}A_{u*})R^\top R(d_v^{-1}A_{v*}^\top)}{\|d_u^{-1}A_{u*}R^\top\|\|d_v^{-1}A_{v*}R^\top\|} = \frac{A_{u*}R^\top RA_{v*}^\top}{\|A_{u*}R\|\|A_{v*}R\|}$. Using parts (a) and (b) of Lemma 2.2 we can calculate that $\cos(A_{u*}, A_{v*}) = \frac{n_{uv}}{\sqrt{n_{uu}n_{vv}}}$ and $\cos(T_{u*}, T_{v*}) = \frac{n_{uv}}{\sqrt{n_{uu}n_{vv}}}$. $\square$

We will state our main result for this method now.

**Theorem 2.6.** *For $P \in \{A, T\}$ and $X = PR^\top$, the following claims hold:*
*(a) Asymptotic result. For $u, v \in V$ and large $q$*

$$\cos(X_{u*}, X_{v*}) \overset{a}{\sim} \mathcal{N}\left(\frac{n_{uv}}{\sqrt{n_{uu}n_{vv}}}, \frac{1}{q}\left(1 - \frac{n_{uv}^2}{n_{uu}n_{vv}}\right)^2\right). \tag{10}$$

*(b) Finite-sample result. Let $\varepsilon \in (0, 0.05]$ and $\delta \in (0, 1)$. If $q \geq \dfrac{2 \ln \left[ \dfrac{2n(n-1)\left(1+\frac{\varepsilon^2}{4}\right)}{\delta} \right]}{\ln \left[ 1 + \frac{\varepsilon^2}{2(1+\varepsilon\sqrt{2})} \right]}$, then*

$$\left| \cos(X_{u*}, X_{v*}) - \frac{n_{uv}}{\sqrt{n_{uu}n_{vv}}} \right| \leq \varepsilon \left( 1 - \frac{n_{uv}^2}{n_{uu}n_{vv}} \right)$$

*holds for all $u, v \in V$ with probability at least $1 - \delta$.*

*Proof.* Using the rotation argument from §A.II, we can get a very useful representation of $\cos(X_{u*}, X_{v*})$ stated in part (b) of Theorem A.6 in the Appendix.

From the Central Limit Theorem and the Law of Large Numbers, we show that $\cos(X_{u*}, X_{v*}) \overset{a}{\sim} \mathcal{N}\left( \cos(P_{u*}, P_{v*}), \frac{1}{q}\left(1 - \cos^2(P_{u*}, P_{v*})\right)^2 \right)$. For details, see part (a) of Proposition A.3 in the Appendix. Claim (a) follows from Lemma 2.5.

Using the representation mentioned at the beginning, we can get concentration results and a results of the JL Lemma type. This is stated in part (b) of Proposition A.3 in the Appendix, whence we have $|\cos(X_{u*}, X_{v*}) - \cos(P_{u*}, P_{v*})| < \varepsilon(1 - \cos^2(P_{u*}, P_{v*}))$ under the same assumptions and conditions. Claim (b) follows from Lemma 2.5. $\qquad\square$

Let us compare the asymptotic result for cosine similarity with the previous cases. From (10), we can produce the approximate three-sigma confidence interval $\left[ \cos(P_{u*}, P_{v*}) - \frac{3}{\sqrt{q}}(1 - \cos^2(P_{u*}, P_{v*})), \; \cos(P_{u*}, P_{v*}) + \frac{3}{\sqrt{q}}(1 - \cos^2(P_{u*}, P_{v*})) \right]$ where $\cos(X_{u*}, X_{v*})$ is expected to take values with 99.7% probability. Differently from the previous cases, we see that the interval endpoints *depend only on the cosine similarity* being estimated. Thus, for a reasonably large $q$, the value of $\cos(P_{u*}, P_{v*})$ will be well-approximated. Further, the closer $\cos(P_{u*}, P_{v*})$ is to 1, the smaller the standard deviation, and the narrower the confidence interval. The confidence interval is the widest when $\cos(P_{u*}, P_{v*}) = 0$. In that case, it can happen that it takes negative values under random projections, but this is likely not to be lower than $-\frac{3}{\sqrt{q}}$.

With Theorem 2.6, we can state the following additional properties.

**Proposition 2.7.** *(a) $\cos(X_{u*}, X_{v*}) \in [-1, 1]$ for all $u, v \in V$.*

*(b) Almost surely $\cos(X_{u*}, X_{v*}) = \pm 1$ if and only if $\cos(P_{u*}, P_{v*}) = \pm 1$.*

*(c) For $\varepsilon \in (0, 0.05]$ and $\delta \in (0, 1)$, if $q \geq \dfrac{2 \ln \left[ \dfrac{2n(n-1)\left(1+\frac{\varepsilon^2}{4}\right)}{\delta} \right]}{\ln \left[ 1 + \frac{\varepsilon^2}{2(1+\varepsilon\sqrt{2})} \right]}$, then*

$$\cos(X_{u*}, X_{v*}) \in [\cos(P_{u*}, P_{v*}) - \varepsilon, \cos(P_{u*}, P_{v*}) + \varepsilon]$$

*for all $u, v \in V$ with probability $1 - \delta$.*

*Proof.* Part (a) follows by definition of cosine similarity. Part (b) follows from the Cauchy-Schwarz inequality. The full discussion can be found in the Corollary A.20 in the Appendix. Part (c) follows from Theorem 2.6 part (b) since $0 \leq 1 - \dfrac{n_{uv}^2}{n_{uu}n_{vv}} = 1 - \cos^2(P_{u*}, P_{v*}) \leq 1$. $\qquad\square$

Part (b) of Proposition 2.7 tells us that random projection will preserve values of exactly 1. Part (c) of Proposition 2.7 provides the desired property of *uniformity* with respect to node degrees for the absolute error on the difference between the estimate and the true value.

## 3   APPLICATION TO RANKING

Here, we specialize the results presented in Section 2 to a ranking application. Recall that, for polynomials $P = p(A)$ or $P = p(T)$, the interpretation is that for $u, v \in V$ $\text{rel}_{uv} := P_{v*}P_{u*}^{\top}$. For

computational reasons, we apply the random projection $X = PR^\top$ and work with $\mathrm{rel}_{uv}^R := X_{v*}X_{u*}^\top$ under the assumption that $\mathrm{rel}_{uv}^R \approx \mathrm{rel}_{uv}$.

We will next examine what happens if we fix $w \in V$ and use the approximate relevance $(\mathrm{rel}_{wh}^R : h \in V)$ for ranking. We will use the well-known NDCG metric: $\mathrm{NDCG}_w^R@K := \frac{\mathrm{DCG}_w^R@K}{\mathrm{DCG}_w@K}$ where

$$\mathrm{DCG}_w^R@K := \sum_{h:\mathrm{rank}_w^R(h) \leq K} \frac{\mathrm{rel}_{wh}}{\log(\mathrm{rank}_w^R(h) + 1)}, \mathrm{DCG}_w@K := \sum_{h:\mathrm{rank}_w(h) \leq l} \frac{\mathrm{rel}_{wh}}{\log(\mathrm{rank}_w(h) + 1)}. \tag{11}$$

Here, $\mathrm{rank}_w(h)$ (resp. $\mathrm{rank}_w^R(h)$) denotes the rank order of $h \in V$ with respect to $w$ induced by $\mathrm{rel}_{wh}$ (resp. $\mathrm{rel}_{wh}^R$). Note that $\mathrm{DCG}_w^R@K$ takes $\mathrm{rel}_{wh}$ and not $\mathrm{rel}_{wh}^R$ in its numerator, since we want to compute the effect of selecting $h$ according to the approximated RP order $\mathrm{rank}_w^R(h)$ in terms of the original relevance $\mathrm{rel}_{wh}$. The following theorem will help us quantify how often the approximation $\mathrm{rel}_{wh}^R$ can *flip* the order of relevance between nodes $u$ and $v$ with respect to $w$.

**Theorem 3.1.** *Let $u, v \in V$ be such that $\mathrm{rel}_{wu} > \mathrm{rel}_{wv}$. Then,*

$$\mathbb{P}(\mathrm{rel}_{wu}^R < \mathrm{rel}_{wv}^R) = \mathbb{P}\left(\mathcal{T}_q > \frac{\cos(P_{w*}, P_{u*} - P_{v*})\sqrt{q}}{\sqrt{1 - \cos^2(P_{w*}, P_{u*} - P_{v*})}}\right) \tag{12}$$

*where $\mathcal{T}_q$ is the t-distribution with parameter q.*

*Proof.* To have $\mathrm{rel}_{wu} < \mathrm{rel}_{wv}$, we need $P_{w*}(P_{u*} - P_{v*})^\top > 0$. The right-hand side is the probability that this dot product will change sign under the random projection, i.e., $\mathbb{P}(P_{w*}R^\top R(P_{u*} - P_{v*})^\top < 0)$. This is a simple consequence that follows from the representation in part (a) of Theorem A.6, which is obtained using the rotation argument and the definition of the $t$-distribution. For details, see Proposition A.2 in the Appendix. $\square$

### 3.1 Instability of Ranking for DotProduct when $P = T$

For a given $u \in V$, we provide information on the set of relevance values $\{\mathrm{rel}_{uv} : v \in V\}$.

**Proposition 3.2.** *For all $u, v \in V$, we have $\mathrm{rel}_{uv} = \frac{n_{uv}}{d_u d_v} \leq \frac{\gamma}{d_u}$. Further,*

$$\mathrm{rel}_{uv} \in \begin{cases} \{0\}, & \text{if } n_{uv} = 0; \\ \left[\frac{1}{c^2}, 1\right], & \text{if } u, v \in L_c, n_{uv} \geq 1; \\ \left[\frac{1}{d_{\max}^2}, \frac{1}{\gamma cq}\right], & \text{if } u \in H_c^\gamma, n_{uv} \geq 1; \end{cases} \tag{13}$$

*where $d_{\max} = \max_{h \in V} d_h$.*

*Proof.* We have $\mathrm{rel}_{uv} = T_{u*}T_{v*}^\top \overset{(3)}{=} \frac{n_{uv}}{d_u d_v} \overset{(4)}{\leq} \frac{\gamma}{d_u}$. To prove (13), we look at each case. If $n_{uv} = 0$, by definition $\mathrm{rel}_{uv} = 0$. If $n_{uv} \geq 1$ and $u, v \in L_c$, then $d_u, d_v \leq c$ and hence $\mathrm{rel}_{uv} = \frac{n_{uv}}{d_u d_v} \geq \frac{1}{d_u d_v} \geq \frac{1}{c^2}$. Finally, if $u \in H_c^\gamma$ and $n_{uv} \geq 1$, we have $d_u \geq \gamma^2 cq$ and $d_u, d_v \leq d_{\max}$; therefore $\frac{1}{d_{max}^2} \leq \frac{1}{d_u d_v} \leq \frac{n_{uv}}{d_u d_v} = \mathrm{rel}_{uv} \leq \frac{\gamma}{d_u} \leq \frac{1}{\gamma cq}$. $\square$

We will have $\gamma \geq 1$ and $q \gg c$, hence the intervals in (13) are all disjoint in this case. This shows that relevance has different values in three non-trivial cases.

Note that the upper bound $\frac{\gamma}{d_u}$ in Proposition 3.2 can be particularly small for a high degree $d_u$. However, the estimate might be larger than that.

**Corollary 3.3.** *For $u \in H_c^\gamma$ and $v \in L_c$, we have $\mathrm{rel}_{uu}^R \overset{a}{\sim} \mathcal{N}\left(\frac{n_{uu}}{d_u^2}, \frac{2n_{uu}^2}{d_u^4 q}\right)$ and $\mathrm{rel}_{uv}^R$ will be asymptotically normal with nonnegative expectation and standard deviation greater than $\frac{\gamma}{d_u}$.*

*Proof.* Asymptotic result $\mathrm{rel}_{uu}^R \overset{a}{\sim} \mathcal{N}\left(\frac{n_{uu}}{d_u^2}, \frac{2n_{uu}^2}{d_u^4 q}\right)$ follows from Theorem 2.3 part(a), when we take $u = v$. It can be shown using definitions of $\gamma$ and $c$, under the assumptions that $u \in H_c^\gamma$ and $v \in L_c$, that the variance in (7), and hence the variance of $\mathrm{rel}_{uv}$, is greater than $(\gamma/d_u)^2$. Details can be found in Lemma B.4 (b) in the Appendix. $\square$

The last result tells us that $\mathrm{rel}_{uu}^R$ will have values in $\left[\frac{n_{uu}}{d_u^2}\left(1-3\sqrt{\frac{2}{q}}\right), \frac{n_{uu}}{d_u^2}\left(1+3\sqrt{\frac{2}{q}}\right)\right]$ with probability more than 99%. This means, for example, if $q \geq 100$, $\mathrm{rel}_{uv}^R \in [1.5\frac{\gamma}{d_u}, \infty) \subset \left[\frac{n_{uu}}{d_u^2}\left(1+3\sqrt{\frac{2}{q}}\right), \infty\right)$ can happen with probability $\mathbb{P}(\mathcal{N}(0,1) > 1.5) \approx 6.7\%$.

For higher $q$, this probability will be higher. Roughly, we can anticipate that $\mathrm{rel}_{uu}^R < \mathrm{rel}_{uv}^R$ happens when $\mathrm{rel}_{uu} > \mathrm{rel}_{uv}$ frequently. The following result will show that this can happen more often than one would like.

**Corollary 3.4.** *Let $u \in H_c^\gamma$ and $v \in L_c$ be two vertices with no common neighbors, i.e., such that $\mathrm{rel}_{uu} > 0$ and $\mathrm{rel}_{uv} = 0$. Then $\mathbb{P}(\mathrm{rel}_{uu}^R < \mathrm{rel}_{uv}^R) > \mathbb{P}(\mathcal{T}_q > \gamma^{-1/2}) \geq \mathbb{P}(\mathcal{T}_q > 1) \approx 15.8\%$.*

**Remark 3.5.** *The previous estimate is based on the fact that for $q \geq 30$, in practice, $\mathcal{N}(0,1)$ is taken as an approximation for $\mathcal{T}_q$.*

*Proof.* Since $P_{u*}P_{v*}^\top = 0$, we have $\cos(P_{u*}, P_{u*} - P_{v*}) = \frac{\|P_{u*}\|}{\sqrt{\|P_{u*}\|^2 + \|P_{v*}\|^2}}$ and (12) becomes

$$\mathbb{P}(\mathrm{rel}_{uu}^R < \mathrm{rel}_{uv}^R) = \mathbb{P}\left(\mathcal{T}_q > \frac{\|P_{u*}\|}{\|P_{v*}\|}\sqrt{q}\right). \tag{14}$$

Further,

$$\frac{\|P_{u*}\|}{\|P_{v*}\|}\sqrt{q} = \frac{\sqrt{n_{uu}/d_u^2}}{\sqrt{n_{vv}/d_v^2}}\sqrt{p} = \frac{d_v\sqrt{n_{uu}}}{d_u\sqrt{n_{vv}}}\sqrt{q} \overset{(2)}{\leq} \frac{d_v\sqrt{n_{uu}}}{d_u\sqrt{d_v}}\sqrt{q} = \frac{\sqrt{d_v n_{uu} q}}{d_u} \overset{(4)}{\leq} \frac{\sqrt{d_v\gamma d_u q}}{d_u}$$

$$= \sqrt{\frac{d_v\gamma q}{d_u}} \overset{v \in L_c}{\leq} \sqrt{\frac{\gamma c q}{d_u}} \overset{u \in H_c^\gamma}{\leq} \frac{1}{\sqrt{\gamma}}.$$

From the last estimate, we have $\mathbb{P}\left(\mathcal{T}_q > \frac{\|P_{v*}\|}{\|P_{u*}\|}\sqrt{q}\right) \geq \mathbb{P}\left(\mathcal{T}_q > \gamma^{-1/2}\right)$, and the claim follows from (14) and Remark 3.5. As mentioned in the paragraph after (4), $\gamma \geq 1$. $\square$

Similar results would follow if we took two vertices $w, u$ of high degree with many common neighbors and a low degree vertex $v$ that has no common neighbors with $w$ and $u$. For simpler calculations, we took $w = u$.

We provide corresponding results for the case of $P = A$ in §B.II of the Appendix. The only notable difference is that, while Corollaries 3.3 and 3.4 will result in poor ranking results for high-degree nodes, their analogous versions (Corollaries B.7 and B.8) will result in poor ranking for low-degree nodes. This is illustrated on Figure 1. Due to the high number of low-degree nodes, this phenomenon is easier to notice empirically. Conversely, it is harder to note such a pathology for higher-degree nodes, since they are usually scarcer.

## 3.2 STABILITY OF RANKING FOR COSINE SIMILARITY WHEN $P = A$ OR $P = T$

In both cases, for the dot product with respect to a given node, when using random projection for relevance estimation, a node of low relevance can be estimated with higher relevance than the node of low relevance. We will show that this is less likely to happen with cosine similarity.

In the case of cosine similarity, for relevance we use $\mathrm{rel}_{uv} = \cos(P_{u*}, P_{v*})$ and $\mathrm{rel}_{uv}^R = \cos(X_{u*}, X_{v*})$. The following proposition, shows the variety of values that $(\mathrm{rel}_{uv} : u, v \in V)$ can have.

**Proposition 3.6.** *For $u, v \in V$ we have $\mathrm{rel}_{uv} \in \begin{cases} \{0\}, & \text{if } n_{uv} = 0; \\ \left[\frac{1}{c^2}, 1\right], & \text{if } u, v \in L_c, n_{uv} \geq 1; \\ \left[\frac{1}{d_{\max}^2}, \frac{1}{\sqrt{q}}\right], & \text{if } u \in H_c^\gamma, v \in L_c, n_{uv} \geq 1; \end{cases}$*

*where $d_{\max} = \max_{h \in V} d_h$.*

*Proof.* If $n_{uv} = 0$, by definition $\mathrm{rel}_{uv} = 0$. If $n_{uv} \geq 1$ and $u, v \in L_c$, from $d_u, d_v \leq c$ and $n_{uu} \overset{(2)}{\leq} d_u^2$, we have $\frac{1}{c^2} \leq \frac{1}{d_u d_v} \leq \frac{n_{uv}}{\sqrt{n_{uu}n_{vv}}} = \mathrm{rel}_{uv} \leq 1$. If $u \in H_c^\gamma$, $v \in L_c$ and $n_{uv} \geq 1$,

from $d_u, d_v \leq d_{\max}$ and $n_{uu} \overset{(2)}{\leq} d_u^2$, we have: $\frac{1}{d_{max}^2} \leq rel_{uv} = \frac{n_{uv}}{\sqrt{n_{uu}n_{vv}}} \overset{(2),(4)}{\leq} \frac{\gamma d_v}{\sqrt{d_u d_v}} = \frac{\gamma \sqrt{d_v}}{\sqrt{d_u}} \leq \frac{\gamma \sqrt{c}}{\sqrt{\gamma^2 cq}} = \frac{1}{\sqrt{q}}$. $\qquad \square$

Again, we will have $\gamma \geq 1$ and $q \gg c$. Hence, the intervals in Proposition 3.6 are all disjoint in this case. This shows that relevance has different values in three non-trivial cases.

From Proposition 2.7, the next result follows.

**Proposition 3.7.** *For all $u \in V$ $\mathrm{rel}_{uu} = \mathrm{rel}_{uu}^R = 1$. Further, both the relevance $(\mathrm{rel}_{uv} : u, v \in V)$ and the approximation $(\mathrm{rel}_{uv}^R : u, v \in V)$ can be at most $1$.*

This gives us a clear interpretation of the relevance: a value of $1$ implies a strong connection between node, and one that is is preserved; a value of $0$ implies no neighbor overlap.

**Corollary 3.8.** *For $u, v, w, h \in V$, the following holds.*

*(a) If $\mathrm{rel}_{uv} = 0$, then $\mathrm{rel}_{uu} = 1 > 0 = \mathrm{rel}_{uv}$ and $\mathrm{rel}_{uu}^R > \mathrm{rel}_{uv}^R$ almost surely.*

*(b) Under the conditions of Proposition 2.7 (c), if $\mathrm{rel}_{uv} - \mathrm{rel}_{wh} > 2\varepsilon$, then $\mathbb{P}(\mathrm{rel}_{uv}^R > \mathrm{rel}_{wh}^R) \geq 1 - \delta$.*

*Proof.* (a) From Proposition 3.7 (c), we have $\mathrm{rel}_{uu} = \mathrm{rel}_{uu}^R = 1$. Since $\mathrm{rel}_{uv}^R$ is a continuous random variable, $\mathbb{P}(\mathrm{rel}_{uv}^R = 1) = 0$. Hence,

$$1 = \mathbb{P}(\mathrm{rel}_{uv}^R \leq 1) = \mathbb{P}(\mathrm{rel}_{uv}^R < 1) + \mathbb{P}(\mathrm{rel}_{uv}^R = 1) = \mathbb{P}(\mathrm{rel}_{uv}^R < 1) + 0 = \mathbb{P}(\mathrm{rel}_{uv}^R < 1).$$

(b) With probability $1 - \delta$, we have $\mathrm{rel}_{uv}^R > \mathrm{rel}_{uv} - \varepsilon$ and $\mathrm{rel}_{wh} + \varepsilon > \mathrm{rel}_{wh}^R$. Therefore, $\mathrm{rel}_{uv}^R > \mathrm{rel}_{uv} - \varepsilon > (\mathrm{rel}_{wh} + 2\varepsilon) - \varepsilon = \mathrm{rel}_{wh} + \varepsilon > \mathrm{rel}_{wh}^R$. $\qquad \square$

Part (a) of Corollary 3.8 shows that the phenomenon from Corollary 3.4 does not happen in the case of cosine similarity (see also a similar result for the adjacency matrix $A$ in Corollary B.8 of the Appendix). Part (b) of Corollary 3.8 shows stability, i.e., if two relevance values are not close, their estimate will highly likely keep their order. Part (a) of Theorem 2.6 provides similar asymptotic guarantees for stability.

The computational experiments in the next section will show, using real graphs, that the standard measure for ranking NDCG will depend on node degrees in the case of the dot product for both $P = A$ and $P = T$, while it will be stable in case of the cosine similarity.

### 3.3 COMPUTATIONAL EXPERIMENTS

We illustrate the results developed in the previous section for a ranking application over a real graph. To that end, we consider the Gleich/wikipedia-20060925 dataset[2] from the University of Florida Sparse Matrix Collection (Davis & Hu, 2011). The dataset consists of a web crawling extraction of Wikipedia with $2,983,494$ web pages and $37,269,096$ web links. We define $A_{ij} = 1$ if page $i$ links to or is linked by page $j$, and $0$ otherwise.

We compare the rankings induced by the original and approximated relevance by evaluating the effect of RP by computing their $\mathrm{NDCG}_i^R @ K =: \eta_i$ as in (11). In particular, we evaluate the effect of $R$ on the quality of the ranking $\eta_i$ induced by their relevance $\mathrm{rel}_{ij}$ when computed according to the three variants of similarity discussed in this paper: $\eta_i^T$ for RP Dot Product when $P = T$, $\eta_i^A$ for RP Dot Product when $P = A$, and $\eta_i^C$ for RP Cosine Similarity. Because the number of possible pairwise node combinations is very large, we evaluate a representative sample of the nodes using a stratified sampling strategy. To assure a meaningful mixture of node degrees, we split the set of nodes into three segments of size $L = 1 \times 10^6$ (low, medium, and high) based on their ordered degrees. Then, we select nodes into three subsets by sampling $300$ out of $L$ nodes without replacement from each of these segments, and take $\mathcal{S} = \mathcal{S}_{\mathrm{low}} \cup \mathcal{S}_{\mathrm{med}} \cup \mathcal{S}_{\mathrm{high}}$ as our evaluation sample. Finally, for each node $i$ in $S$ (and with respect to every other node $j$ in $S$), we compute $\eta_i^T$, $\eta_i^A$, and $\eta_i^C$ considering the types of similarity *(T, A, C)* defined previously.

---

[2]https://www.cise.ufl.edu/research/sparse/matrices/Gleich/wikipedia-20060925.html

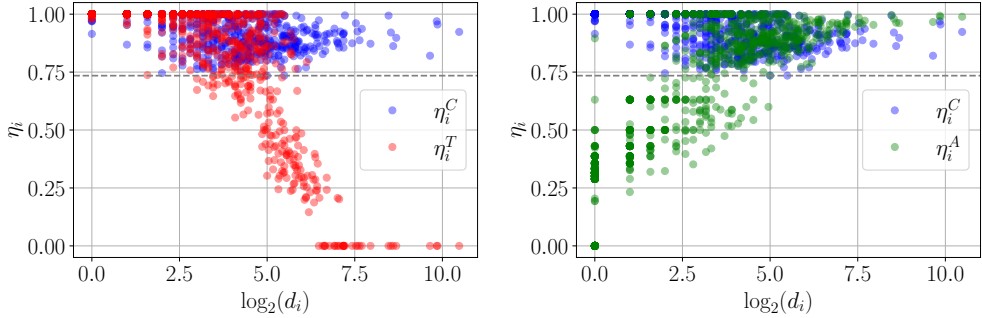

Figure 1: *Distribution of $\eta_i := \text{NDCG}_i^R@10$, versus node degree $d_i$.* On the left, we consider *RP Dot Product when $P = T$*, denoted by $\eta_i^T$. We compare it with *RP Cosine Similarity*, denoted by $\eta_i^C$. Both scores are computed using the same random projection matrix, with dimension $q = 256$. The dotted line marks the lowest value observed for $\eta_i^C$, of approximately 0.75. It can be seen that $\eta_i^T$ often takes low values for *higher* degrees (i.e., region below dotted line), especially when $\log_2(d_i) \geq 4$. On the right, we display the equivalent plot for *RP Dot Product when $P = A$*, denoted by $\eta_i^A$. The NDCG scores often take low values for *lower* degrees, especially when $\log_2(d_i) \leq 6$.

With the previous definitions, we can compute the empirical distributions of $\eta_i$ over the node degrees $d_i$ for the three variants of similarity considered, with $q = 256$ for the random projection dimension. In Figure 1 (left), we compare RP Dot Product when $P = T$ with RP Cosine Similarity. As expected (c.f., §3.1), we can see a strong disruption of NDCG when the degrees are *high*. Correspondingly, in Figure 1 (right), we display the corresponding comparison for RP Dot Product when $P = A$. As expected (c.f., §B.II), we can see a strong disruption of NDCG when the degrees are *low*. RP Cosine Similarity is observed to be largely immune to both of the aforementioned effects, and is able to preserve the quality of the rankings, as anticipated in §3.2. These results are summarized numerically on Table 2, considering the samples taken from $S_{\text{high}}$ (left) and $S_{\text{low}}$ (right), for $K \in \{1, 5, 10\}$.

Table 2: Empirical mean (std) of ranking quality $\eta_i := \text{NDCG}_i^R@K$ for $K \in \{1, 5, 10\}$, and degrees $\log_2(d_i)$ (last row), for $i \in S_{\text{high}}$ (left) and $i \in S_{\text{low}}$ (right). Bold face denotes lower mean values.

| K | $\eta_i^T, i \in S_{\text{high}}$ | $\eta_i^A, i \in S_{\text{high}}$ | $\eta_i^C, i \in S_{\text{high}}$ | $\eta_i^T, i \in S_{\text{low}}$ | $\eta_i^A, i \in S_{\text{low}}$ | $\eta_i^C, i \in S_{\text{low}}$ |
|---|---|---|---|---|---|---|
| 2 | **0.589 (0.426)** | 0.954 (0.061) | 0.964 (0.025) | 1.000 (0.003) | **0.057 (0.200)** | 0.999 (0.008) |
| 5 | **0.607 (0.344)** | 0.920 (0.072) | 0.924 (0.047) | 0.999 (0.004) | **0.168 (0.250)** | 0.999 (0.010) |
| 10 | **0.602 (0.315)** | 0.899 (0.078) | 0.895 (0.063) | 0.999 (0.004) | **0.327 (0.188)** | 0.999 (0.010) |
| | $\log_2(d_i), i \in S_{\text{high}}$: 5.274 (1.103) | | | $\log_2(d_i), i \in S_{\text{low}}$: 0.260 (0.439) | | |

## 4 CONCLUSION

Based on the discussion and numerical results presented in this paper, we have characterized the different estimation behaviors of cosine similarity and dot product under random projections. While cosine similarity tends to show more stable behavior, the choice of similarity measure should be problem-dependent. For applications where norm information is not crucial, cosine similarity can be preferred due to its favorable estimation behavior. However, in scenarios where norm information is essential, the dot product may remain necessary, despite its estimation challenges. Our findings highlight the trade-offs imposed by random projections and provide insights into the potential pathologies for nodes of varying degrees, contributing to a better understanding of the estimation difficulties associated with different similarity metrics. As future work, we envision extending our analysis to the sparse random projection setting by exploring, where applicable, possible similarities between Gaussian and Rademacher random variables.

ACKNOWLEDGMENTS

This paper is an extension of the work conducted by the Graph Intelligence Sciences team at Microsoft, focusing on the relationships and connections of Office 365 entities. The authors wish to express gratitude to their colleagues for their unwavering support. Additionally, the first author would like to acknowledge the hospitality of the Department of Mathematics at the University of Washington in Seattle during the work on this paper.

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

# Appendix

This appendix is divided in two parts:

- In Part A, we provide detailed treatment and proofs for the general results stated and used in the main paper;
- In Part B, we provide results for RP DotProduct when $P = A$. This consists of a parallel development to that presented in §2.1 and §3.1 of the main paper, when we had $P = T$.

## A   DOT PRODUCT AND COSINE SIMILARITY UNDER RANDOM PROJECTIONS

In this part, we provide an in-depth study into the random projection mapping results discussed in the main paper. Our presentation focuses on general properties that are not exclusive to graphs.

- In §A.I, we introduce the notation and define random projection as a mapping.
- In §A.II, we use the rotation argument to provide new representation for the dot product and cosine similarity under the random projections.
- In §A.III, we provide several asymptotic normality results for the dot product. We also use the rotation argument to derive an improved JL Lemma, and compare it with existing versions in the literature.
- In §A.IV, we provide several asymptotic normality results for the cosine similarity, as well novel JL Lemmas, based on the rotation argument. Cosine similarity is a popular quantity to measure similarity. Although random projections have been studied extensively, there are not many papers that include consideration of the cosine similarity.
- In §A.V, we provide a few technical inequalities and concentration results that were used in other sections, for a complete reference to the reader.

We begin by summarizing the main results of Part A in the following proposition.

**Proposition A.1.** *(a) For every $i, j$ we have*

$$(Rp_i, Rp_j) \overset{a}{\sim} \mathcal{N}\left( (p_i, p_j), \frac{1}{q}[\|p_j\|^2 \|p_i\|^2 + (p_j, p_i)^2] \right). \tag{15}$$

*(b) Let $\varepsilon \in (0, 1)$ and $\delta \in (0, 1)$, then for $q \geq 4 \cdot \frac{1+\varepsilon}{\varepsilon^2} \log \frac{k(k-1)}{\delta}$ we have that*

$$|(Rp_j, Rp_i) - (p_j, p_i)| < \varepsilon \|p_j\| \|p_i\|$$

*for $i < j$ with probability at least $1 - \delta$.*

Recall that we used a random matrix $R = [R_{ij}] \in \mathbb{R}^{q \times n}$, where $(R_{ij} : i = 1, \ldots, n, j = 1, \ldots, d)$ are i.i.d. normal random variables with mean zero and variance $1/q$. Here, $p_1, \ldots, p_k$ will be vectors in $\mathbb{R}^n$ who are then mapped to vectors $Rp_1, \ldots, Rp_k$ in $\mathbb{R}^q$. Other notation in this appendix is described in Table 3.

*Proof.* Item (a) is the statement of Corollary A.11. Item (b) is the statement of Theorem A.17. □

We note that if $p_i$ and $p_j$ are orthogonal or $(p_i, p_j)$ is small, then $\|p_i\| \|p_j\|$ can still be large. Hence, by only the dot product we have very little control on the size of the error in both parts (a) and (b) of Proposition A.1.

In the main part, $p_i$ and $p_j$ had non-negative entries, and therefore, the dot $(p_i, p_j)$ products are always non-negative. What is the probability that $(Rp_i, Rp_j)$ will be negative? Remarkably, we are able to provide a closed-formula solution dependent only on the cosine similarity $\frac{(p_j, p_i)}{\|p_j\| \|p_i\|} =: \cos(p_j, p_i)$ of $p_j$ and $p_i$:

Table 3: Notation and symbols used in Appendix A

| | | | |
|---|---|---|---|
| $n$ | original dimension | $M_{2\ldots q}$ | vector with entries $(M_2, \ldots, M_q)$ |
| $q$ | dimension of the projection space | $x, y, p_1 \ldots p_k$ | vectors in $\mathbb{R}^n$ |
| $R$ | $n \times q$ random projection matrix | $(\cdot, \cdot)$ | dot product in $\mathbb{R}^n$ or $\mathbb{R}^q$ |
| $Q$ | equal to or distributed as $\sqrt{q}R$ | $\rho, \rho_{x,y}$ | cosine similarity |
| $M, N$ | std. normal random vectors in $\mathbb{R}^q$ | $\mathcal{T}_q$ | $t$-distribution with parameter $q$ |

**Proposition A.2.** *If $(p_j, p_i) > 0$ and $\cos(p_j, p_i) \neq 1$, then*

$$\mathbb{P}\left((Rp_j, Rp_i) < 0\right) = \mathbb{P}\left(\mathcal{T}_q > \frac{|\cos(p_j, p_i)|\sqrt{q}}{\sqrt{1 - \cos(p_j, p_i)^2}}\right),$$

*where $\mathcal{T}_q$ is the $t$-distribution with the parameter $q$.*

*Proof.* Fact $(p_j, p_i) > 0$ implies $\cos(p_j, p_i) \neq -1$. Therefore, the claim follows from Proposition A.8. □

**Proposition A.3.** *(a) For all $i, j$ such that $p_i \neq 0$ and $p_j \neq 0$, we have:*

$$\cos(Rp_i, Rp_i) \overset{a}{\sim} \mathcal{N}\left(\cos(p_i, p_j), \frac{1}{q}(1 - \cos^2(p_i, p_j))^2\right). \tag{16}$$

*(b) Let $\varepsilon \in (0, 0.05]$, $\delta \in (0, 1)$ and $p_1, \ldots, p_k$ be non-zero vectors in $\mathbb{R}^n$. For*

$$q \geq \frac{2\ln\left[\frac{2k(k-1)\left(1 + \frac{\varepsilon^2}{4}\right)}{\delta}\right]}{\ln\left[1 + \frac{\varepsilon^2}{2(1 + \varepsilon\sqrt{2})}\right]} \tag{17}$$

*the inequality*

$$|\cos(Rp_j, Rp_i) - \cos(p_j, p_i)| \leq \varepsilon(1 - \cos(p_j, p_i)^2)$$

*holds for all $i < j$ with probability at least $1 - \delta$.*

*Proof.* Item (a) is the statement of Theorem A.21. Item (b) is a modified statement of Theorem A.28. □

Note that Proposition A.3 give us guarantees that the error between the projected value and original value is absolute. Additionally, the closer $|\cos(p_j, p_i)|$ is to 1, the smaller the error.

---

### A.I GENERAL RESULTS AND REMARKS

#### A.I.1 VECTOR REPRESENTATION

For any vector $x$ in $\mathbb{R}^n$ we will denote $\tilde{x} := x/\|x\|$.

Note that for non-zero vectors $x$ and $y$ in $\mathbb{R}^n$, cosine similarity has the property

$$\frac{(x, y)}{\|x\|\|y\|} = (x/\|x\|, y/\|y\|) = (\tilde{x}, \tilde{y}). \tag{18}$$

Also, for any matrix $Q \in \mathbb{R}^{q \times n}$

$$\frac{(Qx, Qy)}{\|Qx\|\|Qy\|} = \frac{(Q(x/\|x\|), Q(y/\|y\|))}{\|Q(x/\|x\|)\|\|Q(y/\|y\|)\|} = \frac{(Q\tilde{x}, Q\tilde{y})}{\|Q\tilde{x}\|\|Q\tilde{y}\|} \tag{19}$$

For unit vectors $\tilde{x}$ and $\tilde{y}$, we know from the Gram-Schmidt process that there exists a unique vector $r_{\tilde{x}, \tilde{y}}$ such that

$$\tilde{x} - (\tilde{x}, \tilde{y})\tilde{y} =: r_{\tilde{x}, \tilde{y}}.$$

Recall that $r_{\tilde{x},\tilde{y}}$ is perpendicular to $\tilde{y}$. Hence, we have $\|r_{\tilde{x},\tilde{y}}\| = \sqrt{1 - (\tilde{x}, \tilde{y})^2}$. Hence, we can set $\tilde{r}_{\tilde{x},\tilde{y}} = r_{\tilde{x},\tilde{y}}/\|r_{\tilde{x},\tilde{y}}\|$ and we have

$$\tilde{x} = (\tilde{x}, \tilde{y})\tilde{y} + \sqrt{1 - (\tilde{x}, \tilde{y})^2}\tilde{r}_{\tilde{x},\tilde{y}}. \tag{20}$$

Note that $\tilde{y}$ and $\tilde{r}_{\tilde{x},\tilde{y}}$ are unit and orthogonal vectors.

We will often use $\rho = \rho_{xy} = \frac{(x,y)}{\|x\|\|y\|}$ to denote the cosine similarity of $x$ and $y$.

We can now summarize our findings.

**Proposition A.4.** *For any two non-zero vectors $x$ and $y$ in $\mathbb{R}^n$ there exists a unique unit vector $\tilde{r}_{\tilde{x},\tilde{y}}$ orthogonal to $\tilde{y}$ such that*

$$\tilde{x} = \rho\tilde{y} + \sqrt{1 - \rho^2}\tilde{r}_{\tilde{x},\tilde{y}},$$

*where $\rho$ is the cosine similarity between $x$ and $y$.*

### A.I.2 RANDOM PROJECTION PROPERTIES

In this paper, $Q$ will denote a random $q \times n$ matrix whose entries $Q_{ij} \sim \mathcal{N}(0, 1)$ are i.i.d. Random vectors consisting of i.i.d. $\mathcal{N}(0, 1)$ random variables we will call standard normal random vectors. Note, $Q$ can be obtained by setting $Q := \sqrt{q}P$.

**Lemma A.5.** *Let $x$ and $y$ be two orthogonal unit vectors in $\mathbb{R}^n$, i.e. $(x, y) = 0$ and $\|x\| = \|y\| = 1$. Then $Qx$ and $Qy$ are two independent identically distributed standard normal random vectors of dimension $q$.*

*Proof.* We can show that $(Qx)_1, \ldots, (Qx)_q, (Qy)_1, \ldots, (Qy)_q$ are i.i.d and $\mathcal{N}(0, 1)$.

Note that

$$(Qx)_k = \sum_{j=1}^{n} Q_{kj}x_j \sim \mathcal{N}(0, \|x\|^2), \tag{21}$$

and

$$(Qy)_k = \sum_{j=1}^{n} Q_{kj}y_j \sim \mathcal{N}(0, \|y\|^2). \tag{22}$$

Therefore, all of these random variables have the distribution $\mathcal{N}(0, 1)$. From the definition of $Q$, (21) and (22) we can conclude that the sequences $(Qx)_1, (Qx)_2, \ldots, (Qx)_q$ and $(Qy)_1, (Qy)_2, \ldots, (Qy)_q$ are i.i.d.

Finally, let us look at the covariance between $(Qx)_k$ and $(Qy)_l$.

$$\mathbb{E}[(Qx)_k(Qy)_l] = \mathbb{E}\left[\left(\sum_{j=1}^{n} Q_{kj}x_j\right)\left(\sum_{j=1}^{n} Q_{lj}y_j\right)\right] \tag{23}$$

If $l \neq k$ then two sums under the expectation are independent and hence

$$(23) = \mathbb{E}\left[\left(\sum_{j=1}^{n} Q_{kj}x_j\right)\right]\mathbb{E}\left[\left(\sum_{j=1}^{n} Q_{lj}y_j\right)\right] = 0 \cdot 0.$$

Else, if $l = k$ then:

$$(23) = \mathbb{E}\left[\sum_{j=1}^{n} Q_{kj}^2 x_j y_j + 2\sum_{i \neq j} Q_{kj}Q_{ki}x_j y_i\right]$$

$$= \sum_{j=1}^{n} \underbrace{\mathbb{E}[Q_{kj}^2]}_{1} x_j y_j + 2\sum_{i \neq j} \underbrace{\mathbb{E}[Q_{kj}Q_{ki}]}_{0} x_j y_i$$

$$= \sum_{j=1}^{n} x_j y_j = (x, y) = 0.$$

The claim now follows. $\square$

Recall that $R = q^{-1/2}Q$ is a random $q \times n$ matrix whose entries $R_{ij} \sim \mathcal{N}(0, 1/q)$ are i.i.d. Note that, due to the scaling properties,

$$\frac{(Qx, Qy)}{\|Qx\|\|Qy\|} = \frac{(Rx, Ry)}{\|Rx\|\|Ry\|}. \tag{24}$$

We will state all the main results in the term of the matrix $R$, however $Q$ will let us simplify some proofs.

---

### A.II   ROTATION ARGUMENT AND ORTHOGONAL REPRESENTATION

Let $x$ and $y$ be vectors in $\mathbb{R}^n$ and $\rho = \frac{(x,y)}{\|x\|\|y\|}$. We will show the following result.

**Theorem A.6.** *There exist independent standard normal random vectors $M, N$ of length $q$, such that*

*(a) For the dot product we have*

$$(Qx, Qy) = (\sqrt{q}Rx, \sqrt{q}Ry) = \|x\|\|y\|(\rho\|N\|^2 + M_1\|N\|\sqrt{1-\rho^2}). \tag{25}$$

*(b) For the cosine similarity*

$$\frac{(Qx, Qy)}{\|Qx\|\|Qy\|} = \frac{(Rx, Ry)}{\|Rx\|\|Ry\|} = \frac{\rho\|N\| + M_1\sqrt{1-\rho^2}}{\sqrt{(\rho\|N\| + M_1\sqrt{1-\rho^2})^2 + (1-\rho^2)\|M_{2\ldots q}\|^2}}, \tag{26}$$

*where $M_j$ is the $j$-th component of the vector $M$ and $M_{2\ldots q} = (M_2, \ldots, M_q)$.*

The rotation argument is powered by the the following Lemma. Recall that $\tilde{N} = N/\|N\|$.

**Lemma A.7.** *Let $H$ and $N$ be independent standard normal random vectors of dimension $q$, and $U_{\tilde{N}}$ an orthogonal $q \times q$ matrix such that $U_{\tilde{N}}\tilde{N} = e_1$. Then random vectors $U_{\tilde{N}}H$ and $N$ are independent standard normal.*

*Proof.* Note that for every orthogonal matrix $U$, vector $UH$ is normal random with expectation $0$ and identity covariance matrix. Therefore, it has the same distribution as $H$.

We will show that $(U_{\tilde{N}}H, N)$ has the same distribution as $(H, N)$. Let $f_N$ be the density of $N$ and since $N$ is independent of $H$ we have

$$\mathbb{P}(U_{\tilde{N}}H \in A, N \in B) = \int_{z \in B} \mathbb{P}(U_{\tilde{z}}H \in A)f_N(z)dz$$

$$\overset{U_{\tilde{z}}H \overset{d}{=} H}{=} \int_{z \in B} \mathbb{P}(H \in A)f_N(z)dz = \mathbb{P}(H \in A, N \in B)$$

for all measurable subsets $A, B$ of $\mathbb{R}^q$. $\qquad \square$

*Proof of Theorem A.6.* Note that $Q\tilde{x} = \rho Q\tilde{y} + \sqrt{1-\rho^2}Q\tilde{r}_{\tilde{x},\tilde{y}}$. Since, $(\tilde{y}, \tilde{r}_{\tilde{x},\tilde{y}}) = 0$, by Lemma A.5 we know that $H = Q\tilde{r}_{\tilde{x},\tilde{y}}$ and $N = Q\tilde{y}$ are independent standard normal random variables. Hence, we have

$$\frac{(Qx, Qy)}{\|Qx\|\|Qy\|} \overset{(19)}{=} \frac{(Q\tilde{x}, Q\tilde{y})}{\|Q\tilde{x}\|\|Q\tilde{y}\|} = \frac{(\rho N + \sqrt{1-\rho^2}H, N)}{\|\rho N + \sqrt{1-\rho^2}H\|\|N\|} = \frac{(\rho N + \sqrt{1-\rho^2}H, \tilde{N})}{\|\rho N + \sqrt{1-\rho^2}H\|}. \tag{27}$$

We now pick the orthogonal $q \times q$ matrix $U_{\tilde{N}}$ for which $U_{\tilde{N}}\tilde{N} = e_1$. Applying the $U_{\tilde{N}}$-transformation to all arguments in (27) we get

$$\frac{(Qx, Qy)}{\|Qx\|\|Qy\|} = \frac{(U_{\tilde{N}}(\rho\|N\|\tilde{N} + \sqrt{1-\rho^2}H), U_{\tilde{N}}(\tilde{N}))}{\|U_{\tilde{N}}(\rho\|N\|\tilde{N} + \sqrt{1-\rho^2}H)\|} = \frac{(\rho\|N\|e_1 + \sqrt{1-\rho^2}U_{\tilde{N}}H, e_1)}{\|\rho\|N\|e_1 + \sqrt{1-\rho^2}U_{\tilde{N}}H\|}.$$

Setting $M := U_{\tilde{N}}H$. By Lemma A.7, $M$ remains standard normal random vector independent of $N$.

We can apply the same arguments to the dot product:

$$
\begin{aligned}
(Q\tilde{x}, Q\tilde{y}) &= (\rho N + \sqrt{1-\rho^2}H, N) = \|N\|(\rho N + \sqrt{1-\rho^2}H, \tilde{N}) \\
&= \|N\|(U_{\tilde{N}}(\rho\|N\|\tilde{N} + \sqrt{1-\rho^2}H), U_{\tilde{N}}(\tilde{N})) = \|N\|(\rho\|N\|e_1 + \sqrt{1-\rho^2}U_{\tilde{N}}H, e_1) \\
&= \rho\|N\|^2 + \sqrt{1-\rho^2}(U_{\tilde{N}}H, e_1)\|N\| = \rho\|N\|^2 + \sqrt{1-\rho^2}M_1\|N\|.
\end{aligned}
$$

$\square$

GENERAL APPROACH

The results can be generalized using the properties of the Wishart distribution from the theory of random matrices[3]. We will outline the approach, based on to several results from §2.4 in (Kollo & von Rosen, 2005).

Let $p_1, p_2, \ldots, p_k$ be vectors in $\mathbb{R}^n$ that form the columns of matrix $P$. The matrix $P^\top Q^\top QP$ is a random matrix with a central Wishart distribution $W_k(P^\top P, q)$. Note that $(P^\top P)_{ij} = (p_i, p_j)$. Let $L^\top L$ be the Cholesky decomposition of $P^\top P$. Using the properties of the Wishart distribution (Theorem 2.4.2 from (Kollo & von Rosen, 2005)), we have $W_k(P^\top P, q) = W_k(L^\top L, q)$. Let $T^\top T$ be the Bartlett decomposition of a Wishart matrix with the distribution $W_k(I, n)$ (Corollary 2.4.2.1 from (Kollo & von Rosen, 2005)). Then,

$$
P^\top Q^\top QP \stackrel{d}{=} L^\top T^\top TL.
$$

Finally, we know that the elements $(T_{ij} : i, j = 1, \ldots, p)$ are independent, where $T_{ii}^2 \sim \chi^2(n-i+1)$ $i = 1, \ldots, k$; $T_{ij} \sim N(0,1)$ for $i < j$; and the rest of the matrix is zero.

For $k = 2$, when $p_1 = x$ and $p_2 = y$ we have $L^\top = \begin{bmatrix} \|x\| & 0 \\ \rho\|x\|\|y\| & \sqrt{1-\rho^2}\|y\| \end{bmatrix}$. Hence, we get

$$
P^\top Q^\top QP \stackrel{d}{=}
$$
$$
\begin{bmatrix} \|x\|^2 T_{11}^2 & \|x\|\|y\|(\rho T_{11}^2 + \sqrt{1-\rho^2}T_{11}T_{12}) \\ \|x\|\|y\|(\rho T_{11}^2 + \sqrt{1-\rho^2}T_{11}T_{12}) & \|y\|^2\left((1-\rho^2)T_{11}^2 + \left(\sqrt{1-\rho^2}T_{12} + \rho T_{22}\right)^2\right) \end{bmatrix}.
$$

Note that since $(\|N\|, \|M_{2\ldots q}\|, M_1) \stackrel{d}{=} (T_{11}, T_{22}, T_{12})$, last equality provides the representation obtained in Theorem A.6.

---

A.III    ASYMPTOTIC AND FINITE-SAMPLE RESULTS FOR THE DOT PRODUCT

In this section we analyze how the dot product changes under random projection. Some of the results here are well known, but for completeness and to demonstrate the power of the representation in Theorem A.6, we will prove them.

A.III.1    PROBABILITY OF SIGN CHANGE

One of the practical questions when doing the random projections is will the dot product or cosine similarity change the sign. In practical settings vectors $x$ and $y$ can have all non-negative components and as such their dot product is also non-negative. Can the projection change that? The representation for the dot product gives us an exact answer to that question.

**Proposition A.8.** *(a) If $(x, y) = 0$ and $\|x\|\|y\| \neq 0$ then $\mathbb{P}((Rx, Ry) < 0) = \mathbb{P}((Rx, Ry) > 0) = \frac{1}{2}$.*

---

[3]We thank the reviewers for pointing this to us during the review process. Since we were not able to find the result explicitly in the literature we are outlining the approach here.

*(b) If $(x, y) \neq 0$ and $|\rho| \neq 1$ then*

$$\mathbb{P}\left(\frac{(Rx, Ry)}{(x, y)} < 0\right) = \mathbb{P}\left(\mathcal{T}_q > \frac{|\rho|\sqrt{q}}{\sqrt{1 - \rho^2}}\right),$$

*where $\mathcal{T}_q$ is the $t$-distribution with the parameter $q$.*

*Proof.* In the case (a) $\rho = 0$ and the claim follows from the (25), since the sign depends on the sign of $M_1$ which can be positive or negative with probability $\frac{1}{2}$. In (b) we can assume $(x, y) > 0$ and therefore $\rho > 0$. Hence,

$$\mathbb{P}\left(\frac{(Rx, Ry)}{(x, y)} < 0\right) = \mathbb{P}\left((Rx, Ry) < 0\right) = \mathbb{P}\left((q^{-1/2}Qx, q^{-1/2}Qy) < 0\right)$$

$$= \mathbb{P}\left((Qx, Qy) < 0\right) = \mathbb{P}\left(\rho\|N\|^2 + M_1\|N\|\sqrt{1 - \rho^2} < 0\right)$$

$$= \mathbb{P}\left(\frac{M_1}{\|N\|} < -\frac{\rho}{\sqrt{1 - \rho^2}}\right) = \mathbb{P}\left(\frac{M_1}{\sqrt{\|N\|^2/q}} < -\frac{\rho\sqrt{q}}{\sqrt{1 - \rho^2}}\right).$$

Since $\mathcal{T}_q := \frac{M_1}{\sqrt{\|N\|^2/p}} \sim t(q)$ the claim follows from the fact that $t(q)$ is symmetric random variable for the last expression we have

$$= \mathbb{P}\left(\mathcal{T}_q < -\frac{\rho\sqrt{q}}{\sqrt{1 - \rho^2}}\right) = \mathbb{P}\left(\mathcal{T}_q > \frac{\rho\sqrt{q}}{\sqrt{1 - \rho^2}}\right).$$

$\square$

As we can see from the graph on Figure 2 the probability of changing a sign is the highest - $\frac{1}{2}$ for $\rho = 0$ and and quickly becomes really low. From Corollary 3.2.2. in (Kaban, 2015) we know that the $\mathbb{P}\left(\frac{(Rx, Ry)}{(x, y)} < 0\right) \leq \exp(-q\rho^2/8)$ and the result here gives the exact value of the probability.

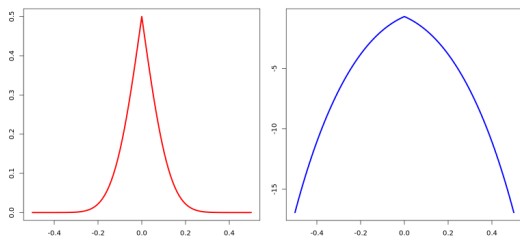

Figure 2: Graphs of functions $\rho \mapsto \mathbb{P}\left(\mathcal{T}_q > \frac{|\rho|\sqrt{q}}{\sqrt{1 - \rho^2}}\right)$ and $\rho \mapsto \log \mathbb{P}\left(\mathcal{T}_q > \frac{|\rho|\sqrt{q}}{\sqrt{1 - \rho^2}}\right)$ for $q = 100$ on the interval $[-0.5, 0.5]$.

This results tells us that the probability of the dot product sign change under random projection is a function of $\rho$, which in practice we don't know. If we wanted to calculate a probability of a cosine similarity or dot product changing signs in case of a large number of vectors it would be hard to do so. However we can still conclude that if $\rho$ is close to zero we can expect the sign to change, while if it is reasonably far from 0 it is unlikely that the sign will change.

### A.III.2 CENTRAL LIMIT THEOREM

The representation given in (25) has many consequences. For example, we can simply calculate the expectation and the variance of the dot product.

**Proposition A.9.** *We have $\mathbb{E}\left[\frac{(Rx, Ry)}{\|x\|\|y\|}\right] = \rho$ and $\mathrm{Var}\left[\frac{(Rx, Ry)}{\|x\|\|y\|}\right] = \frac{1+\rho^2}{q}$.*

*Proof.* Recall that $\mathbb{E}[M_1] = 1$ and $\mathbb{E}[M_1^2] = 1$; $\mathbb{E}[\|N\|^2] = q$ and $\mathbb{E}[\|N\|^4] = q(q+2)$. Using (25) we have

$$\mathbb{E}\left[\frac{(Rx, Ry)}{\|x\|\|y\|}\right] = q^{-1}(\rho\mathbb{E}[\|N\|^2] + \mathbb{E}[\|N\|]\mathbb{E}[M_1]\sqrt{1-\rho^2})$$

$$= q^{-1}\rho q + \mathbb{E}[\|N\|] \cdot 0 \cdot \sqrt{1-\rho^2} = \rho,$$

and

$$\mathbb{E}\left[\left(\frac{(Rx, Ry)}{\|x\|\|y\|}\right)^2\right]$$

$$= q^{-2}\rho^2\mathbb{E}[\|N\|^4] + 2q^{-2}\mathbb{E}[\|N\|^3]\mathbb{E}[M_1]\rho\sqrt{1-\rho^2} + q^{-2}\mathbb{E}[\|N\|^2]\mathbb{E}[M_1^2](1-\rho^2)]$$

$$= q^{-2}\rho^2 q(q+2) + 0 + q^{-2} \cdot q \cdot 1 \cdot (1-\rho^2) = \rho^2 + \frac{1+\rho^2}{q}.$$

Now, since $\mathrm{Var}\left[\frac{(Rx, Ry)}{\|x\|\|y\|}\right] = \mathbb{E}\left[\left(\frac{(Rx, Ry)}{\|x\|\|y\|}\right)^2\right] - \left(\mathbb{E}\left[\frac{(Rx, Ry)}{\|x\|\|y\|}\right]\right)^2$, the claim follows. $\qquad\square$

We can also get the following asymptotic result relatively easy.

**Theorem A.10.** *We have*

$$\sqrt{q}\left[\frac{(Rx, Ry)}{\|x\|\|y\|} - \rho\right] \xrightarrow{d} \mathcal{N}(0, 1+\rho^2),$$

*as $q \to \infty$.*

*Proof.* Form the Central Limit Theorem we have $\frac{\|N\|^2 - q}{\sqrt{q}} = q^{-1/2}\sum_{j=1}^{q}(N_1^2 - 1) \xrightarrow{d} \mathcal{N}(0, 2)$ and by the Law of Large Numbers we have $\frac{1}{\sqrt{q}}\|N\| = \sqrt{\frac{N_1^2 + \ldots + N_q^2}{q}} \to 1$ almost surely. Since $M_1$ is independent, we have

$$\sqrt{q}\left[\frac{(Rx, Ry)}{\|x\|\|y\|} - \rho\right] = \left[\rho \cdot \frac{\|N\|^2 - q}{\sqrt{q}} + M_1\frac{\|N\|}{\sqrt{q}}\sqrt{1-\rho^2}\right]$$

$$\xrightarrow{d} \hat{\mathcal{N}}(0, 2\rho^2) + \mathcal{N}(0, 1-\rho^2) \overset{d}{=} \mathcal{N}(0, 1+\rho^2).$$

$\qquad\square$

We can write the last result in the following form, which we stated in part (a) of Proposition A.1.

**Corollary A.11.** *We have*

$$\sqrt{q}[(Rx, Ry) - (x, y)] \xrightarrow{d} \mathcal{N}(0, \|x\|^2\|y\|^2 + (x, y)^2),$$

*as $q \to \infty$.*

Although, $(Rx, Ry)$ is an unbiased and asymptotically normal estimator for $(x, y)$, the control of the standard deviation can be a problem. $(x, y)$ could be small compared to the value of $\sigma = \sqrt{\frac{\|x\|^2\|y\|^2 + (x, y)^2}{q}}$. Let $x = (1, 0, 1, 0, \ldots)$ and $y = (0, 1, 0, 1, \ldots)$, then $(x, y) = 0$ and $\sigma = \frac{n}{2\sqrt{q}}$. However, since $q \ll n$ the random projection might be nowhere near the value of $(x, y)$. Simulation in Figure 3 illustrates this issue.

### A.III.3 CONCENTRATION RESULTS

So far we have shown that $\frac{(Rx, Ry)}{\|x\|\|y\|}$ is an unbiased and asymptotically normal estimator for $\rho$. However when we have many vectors and we want to be sure that all of their similarities get randomly projected to approximate values we will need to use concentration results.

First, using (25) we calculate the Laplace transform.

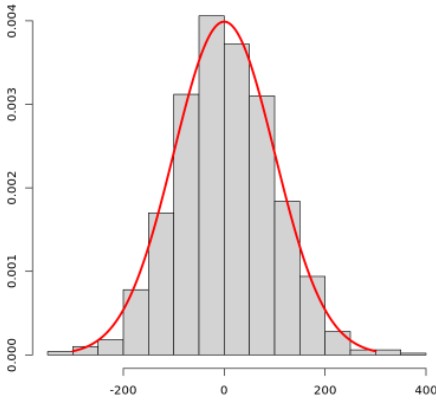

Figure 3: Simulated dot product $(R\tilde{x}, R\tilde{y})$ for $x = (1, 0, 1, 0, \ldots)$ and $y = (0, 1, 0, 1, \ldots)$ for $n = 2000$ and $q = 100$.

**Proposition A.12.** *We have*

$$\mathbb{E}[\exp(\lambda(Q\tilde{x}, Q\tilde{y}))] = [(1 - \lambda(1 + \rho))(1 - \lambda(\rho - 1))]^{-q/2} \tag{28}$$

*for $\lambda \in (-(1-\rho)^{-1}, (1+\rho)^{-1})$*

*Proof.* Recall that $\mathbb{E}(e^{\lambda M_1}) = e^{\lambda^2/2}$ for all $\lambda \in \mathbb{R}$ and $\mathbb{E}(e^{\lambda \|N\|^2}) = (1 - 2\lambda)^{-q/2}$ for $\lambda < 1/2$. By applying the Laplace transform first to $M_1$ and then to $\|N\|^2$ we get

$$
\begin{aligned}
\mathbb{E}[\exp(\lambda(Q\tilde{x}, Q\tilde{y}))] &= \mathbb{E}\left[ e^{\lambda\rho\|N\|^2} \mathbb{E}\left[ e^{\lambda\|N\|\sqrt{1-\rho^2}M_1} \big| \|N\| \right] \right] \\
&= \mathbb{E}\left[ e^{\lambda\rho\|N\|^2} e^{\lambda^2\|N\|^2(1-\rho^2)/2} \right] = \mathbb{E}\left[ e^{(\lambda\rho + \lambda^2(1-\rho^2)/2)\|N\|^2} \right] \\
&= (1 - (2\lambda\rho - \lambda^2(1 - \rho^2)))^{-q/2} = [(1 - \lambda(1 + \rho))(1 - \lambda(\rho - 1))]^{-q/2}
\end{aligned}
$$

$\square$

We will now use the the Laplace Transform to obtain Chernhov bounds. The following Lemma will be a useful estimate.

**Lemma A.13.** *For $\lambda \in (0, (1+\rho)^{-1})$ we have*

$$\log \mathbb{E}[\exp(\lambda[(Q\tilde{x}, Q\tilde{y}) - q\rho])] \leq \frac{\lambda^2 q(1 + \rho^2)}{2(1 - \lambda(1 + \rho))}, \tag{29}$$

*and $\lambda \in (0, (1-\rho)^{-1})$*

$$\log \mathbb{E}[\exp(-\lambda[(Q\tilde{x}, Q\tilde{y}) - q\rho])] \leq \frac{\lambda^2 q(1 + \rho^2)}{2(1 - \lambda(1 - \rho))}. \tag{30}$$

*Proof.* In case $\rho = 1$ or $\rho = -1$ expression in (28) will simplify and we will be able to use Proposition A.35 on $X$ and $-X$ respectively.

One can show that for $s \in (0, 1)$ we have (see Lemma A.38 (a)):

$$(-\log(1 - s) - s) \leq \frac{s^2}{2(1 - s)}, \tag{31}$$

and for $s < 0$ (see Lemma A.38 (a)):

$$(-\log(1 - s) - s) \leq \frac{s^2}{2}. \tag{32}$$

We will prove (29), (30) is shown in the same way.

$$\log \mathbb{E}[\exp(\lambda(Q\tilde{x}, Q\tilde{y}) - q\rho)]$$

$$= -\lambda q\rho - \frac{q}{2}\log(1 - \lambda(1+\rho)) - \frac{q}{2}\log(1 - \lambda(\rho-1))$$

$$= \frac{q}{2}\left[-\log(1-\lambda(1+\rho)) - \lambda(1+\rho)\right] + \frac{q}{2}\left[-\log(1-\lambda(\rho-1)) - \lambda(\rho-1)\right]$$

$$\overset{(31),(32)}{\leq} \frac{q}{2}\left[\frac{(\lambda(1+\rho))^2}{2(1-\lambda(1+\rho))} + \frac{(\lambda(\rho-1))^2}{2}\right]$$

$$\leq \frac{q}{2}\left[\frac{(\lambda(1+\rho))^2}{2(1-\lambda(1+\rho))} + \frac{(\lambda(\rho-1))^2}{2(1-\lambda(1+\rho))}\right]$$

$$= \frac{q}{2}\left[\frac{(\lambda(1+\rho))^2 + (\lambda(\rho-1))^2}{2(1-\lambda(1+\rho))}\right] = \frac{\lambda^2 q(1+\rho^2)}{2(1-\lambda(1+\rho))}$$

This proves (29), the equation (30) can be proved in the same way. $\qquad\square$

**Theorem A.14.** *For $t \geq 0$ we have*

$$\mathbb{P}((Q\tilde{x}, Q\tilde{y}) - q\rho > t) \leq \exp\left(\frac{-t^2}{2q(1+\rho^2) + 2(1+\rho)t}\right), \tag{33}$$

*and*

$$\mathbb{P}((Q\tilde{x}, Q\tilde{y}) - q\rho < -t) \leq \exp\left(\frac{-t^2}{2q(1+\rho^2) + 2(1-\rho)t}\right). \tag{34}$$

*Proof.* Note that $\mathbb{E}[(Q\tilde{x}, Q\tilde{y})] = q\rho$.

Since (29) holds we can use Theorem A.33 by setting $X = (Q\tilde{x}, Q\tilde{y})$ to obtain (33).

Since (30) holds we can use Theorem A.33 by setting by setting $X = -(Q\tilde{x}, Q\tilde{y})$ to obtain (34). $\quad\square$

**Proposition A.15.** *For any function $f : [-1, 1] \to [0, \infty)$ we have:*

$$\mathbb{P}((Q\tilde{x}, Q\tilde{y}) - q\rho > q\varepsilon f(\rho)) \leq \exp\left(\frac{-qf(\rho)^2}{2(1+\rho^2)}\left[\varepsilon^2 - \varepsilon^3\frac{1+\rho}{1+\rho^2}f(\rho)\right]\right) \tag{35}$$

$$\mathbb{P}((Q\tilde{x}, Q\tilde{y}) - q\rho < -q\varepsilon f(\rho)) \leq \exp\left(\frac{-qf(\rho)^2}{2(1+\rho^2)}\left[\varepsilon^2 - \varepsilon^3\frac{1-\rho}{1+\rho^2}f(\rho)\right]\right) \tag{36}$$

**Remark A.16.** *Note that for $\rho = 1$ and $f(\rho) = \varepsilon$ we get the usual bounds for $\chi^2(q)$ used to prove Johnson -Lindenstrauss Lemma as given, for example in Lemma 1.3. of (Vempala, 2004) and (Ghojogh et al., 2021).*

### A.III.4 JOHNSON-LINDENSTRAUSS-TYPE RESULT

**Theorem A.17.** *Let $p_1, \ldots p_k$ be a finite set of vectors in $\mathbb{R}^n$ and let $\varepsilon \in (0, 1)$ and $\delta \in (0, 1)$, then for $q \geq 4 \cdot \frac{1+\varepsilon}{\varepsilon^2}\log\frac{k(k-1)}{\delta}$*

$$\left|\frac{(Rp_j, Rp_i)}{\|p_j\|\|p_i\|} - \frac{(p_j, p_i)}{\|p_j\|\|p_i\|}\right| < \varepsilon$$

*for $i < j$ with probability at least $1 - \delta$.*

We have $\binom{k}{2}$ cosine similarities in the statement above and in practice we will have no way of estimating[4] them. To prove this result we need a probablity estimate that doesn't depend on the value of the cosine similarity. The following Proposition will help us in that by simplifying the statement of Theorem A.14.

---

[4]One exception is the case when $x_1, \ldots, x_k$ are all in $[0, \infty)^n$, i.e. have nonnegative coordinates. In that case we know cosine similarities will be in interval $[0, 1]$.

**Proposition A.18.** *For $\varepsilon > 0$ we have*

$$\mathbb{P}\left(\frac{(Rx, Ry)}{\|x\|\|y\|} - \rho > \varepsilon\right) \leq \exp\left(\frac{-q\varepsilon^2}{4(1+\varepsilon)}\right), \tag{37}$$

*and*

$$\mathbb{P}\left(\frac{(Rx, Ry)}{\|x\|\|y\|} - \rho < -\varepsilon\right) \leq \exp\left(\frac{-q\varepsilon^2}{4(1+\varepsilon)}\right). \tag{38}$$

*Proof.* By setting $t = q\varepsilon$ in (33) we get

$$\mathbb{P}\left(\frac{(Rx, Ry)}{\|x\|\|y\|} - \rho > \varepsilon\right) \leq \exp\left(\frac{-q\varepsilon^2}{2(1+\rho^2) + 2(1+\rho)\varepsilon}\right),$$

Using the fact that $|\rho| \leq 1$ we get

$$\frac{-q\varepsilon^2}{2(1+\rho^2) + 2(1+\rho)\varepsilon} \leq \frac{-q\varepsilon^2}{4(1+\varepsilon)}.$$

This proves (37). Bound in (38) can be shown in a same way. $\qquad\square$

*Proof of Theorem A.17.* Define sets

$$A_{ij} = \left(\left|\frac{(Rp_j, Rp_i)}{\|p_j\|\|p_i\|} - \frac{(p_j, p_i)}{\|p_j\|\|p_i\|}\right| < \varepsilon\right)$$

$$A_{ij}^{c+} = \left(\frac{(Rp_j, Rp_i)}{\|p_j\|\|p_i\|} - \frac{(p_j, p_i)}{\|p_j\|\|p_i\|} > \varepsilon\right)$$

$$A_{ij}^{c-} = \left(\frac{(Rp_j, Rp_i)}{\|p_j\|\|p_i\|} - \frac{(p_j, p_i)}{\|p_j\|\|p_i\|} < -\varepsilon\right)$$

Note that $A_{ij}^c = A_{ij}^{c+} \cup A_{ij}^{c-}$. Using the union bound tehnicque we get

$$P\left(\bigcap_{i<j} A_{ij}\right) = 1 - P\left(\bigcup_{i<j} A_{ij}^c\right) = 1 - P\left(\bigcup_{i<j} A_{ij}^{c+} \cup A_{ij}^{c-}\right) \geq 1 - \sum_{i<j}[\mathbb{P}(A_{ij}^{c+}) + \mathbb{P}(A_{ij}^{c-})].$$

From Proposition A.18 and bound on $q$ we have that

$$\sum_{i<j}[\mathbb{P}(A_{ij}^{c+}) + \mathbb{P}(A_{ij}^{c-})] \leq k(k-1)\exp\left(-q\frac{\varepsilon^2}{4(1+\varepsilon)}\right)$$

$$\leq k(k-1)\exp\left(-4 \cdot \frac{1+\varepsilon}{\varepsilon^2}\log\frac{k(k-1)}{\delta} \cdot \frac{\varepsilon^2}{4(1+\varepsilon)}\right) = \delta.$$

$\qquad\square$

### A.III.5   COMPARISON WITH KNOWN RESULTS

Regarding known estimates for the dot product, Theorem 2.1. from (Kaban, 2015) provides the following bounds for $\varepsilon \in (0, 1)$:

$$\mathbb{P}\left(\frac{(Rx, Ry)}{\|x\|\|y\|} - \rho > \varepsilon\right) \leq e^{-\frac{q\varepsilon^2}{8}}, \quad \text{and} \quad \mathbb{P}\left(\frac{(Rx, Ry)}{\|x\|\|y\|} - \rho < -\varepsilon\right) \leq e^{-\frac{q\varepsilon^2}{8}}.$$

This a simple consequence of Proposition A.18.

Equation (4) in (Kang, 2021) and Lemma 5.7. in (Vempala, 2004) provide the following estimate:

$$\mathbb{P}\left(\left|\frac{(Rx, Ry)}{\|x\|\|y\|} - \rho\right| \leq \varepsilon\right) \geq 1 - 4e^{-\frac{q(\varepsilon^2 - \varepsilon^3)}{4}}.$$

Since $\varepsilon^2 - \varepsilon^3 \leq \frac{\varepsilon^2}{1+\varepsilon}$, again, from Proposition A.18 we can get a better estimate

$$\mathbb{P}\left(\left|\frac{(Rx, Ry)}{\|x\|\|y\|} - \rho\right| \leq \varepsilon\right) \geq 1 - 2e^{-\frac{q\varepsilon^2}{4(1+\varepsilon)}}.$$

### A.IV ASYMPTOTIC AND FINITE-SAMPLE RESULTS FOR COSINE SIMILARITY

In this section, we will examine the behavior of cosine similarity under random projection. We will begin by examining some special cases and then develop a general approach.

#### A.IV.1 THE CASE OF $\rho = \pm 1$

Recall that in the case of $\rho = \pm 1$, i.e.

$$\frac{(x, y)}{\|x\|\|y\|} = \pm 1$$

there exists $\alpha > 0$ such that $x = \pm \alpha y$. This is a known consequence of the Cauchy-Schwarz inequality.

In this case we have

$$\frac{(Rx, Ry)}{\|Rx\|\|Ry\|} = \frac{(R(\pm\alpha y), Ry)}{\|R(\pm\alpha y)\|\|Ry\|} = = \frac{\pm\alpha(Ry, Ry)}{\alpha\|Ry\|\|Ry\|} = \pm\frac{\|Ry\|^2}{\|Ry\|^2} = \pm 1 = \rho. \qquad (39)$$

It turns out the that this is preserved under random projections.

**Proposition A.19.** *If $\frac{(Rx,Ry)}{\|Rx\|\|Ry\|} = \pm 1$ then there exists $\alpha > 0$ such that $x = \pm\alpha y$ (almost surely).*

*Proof.* Since the cosine similarity of $Rx$ and $Ry$ is $\pm 1$, then there is an $\alpha > 0$ such that $Rx = \pm\alpha Ry$. Hence, $w = R(x \mp \alpha y) = 0$. Let us assume $x \mp \alpha y \neq 0$ then $R(x \mp \alpha y)$ is $q$-dimensional vector with entries distributed as $\mathcal{N}(0, \|x \mp \alpha y\|^2/q)$. The probability of which being a zero vector is 0. Hence, we have $x = \pm\alpha y$ almost surely. $\qquad\square$

Hence, the in the cosine similarity of $\pm 1$ will be preserved under random projection $Q$.

**Corollary A.20.** *If $\frac{(Rx,Ry)}{\|Rx\|\|Ry\|} = \pm 1$ almost surely if and only if $\frac{(x,y)}{\|x\|\|y\|} = \pm 1$.*

*Proof.* $\frac{(Rx,Ry)}{\|Rx\|\|Ry\|} = \pm 1$, by Proposition A.19, holds if and only if $x = \pm\alpha y$ for some $\alpha > 0$, and, Cauchy-Shcwarz inequality, this holds if and only if $\frac{(x,y)}{\|x\|\|y\|} = \pm 1$. $\qquad\square$

#### A.IV.2 CENTRAL LIMIT THEOREM

In this section we will show that the value of $\frac{(Rx,Ry)}{\|Rx\|\|Ry\|}$ is approximately normally distributed around $\rho$ with a variance $\frac{(1-\rho^2)^2}{q}$.

**Theorem A.21.** *For non-zero vectors $x$ and $y$ in $\mathbb{R}^n$ we have*

$$\sqrt{q}\left[\frac{(Rx, Ry)}{\|Rx\|\|Ry\|} - \frac{(x, y)}{\|x\|\|y\|}\right] \xrightarrow{d} \mathcal{N}\left(0, \left[1 - \left(\frac{(x, y)}{\|x\|\|y\|}\right)^2\right]^2\right), \qquad (40)$$

*as $q \to \infty$.*

Theorem A.21 can be proven using the delta-method technique. We will use the representation given in Theorem A.6 to prove it.

**Lemma A.22.** *Define $T_q = \rho\|N\| + M_1\sqrt{1 - \rho^2}$ and $B_q = \|M_{2\ldots q}\|$, then from (26) we have:*

$$\frac{(Rx, Ry)}{\|Rx\|\|Ry\|} - \rho = (1 - \rho^2) \cdot \frac{(T_q)^2 - \rho^2 B_q^2}{\sqrt{(T_q)^2 + (1 - \rho^2)B_q^2}} \cdot \frac{1}{T_q + \rho\sqrt{(T_q)^2 + (1 - \rho^2)B_q^2}} \qquad (41)$$

*Proof.* We have:

$$\frac{(Rx, Ry)}{\|Rx\|\|Ry\|} - \rho = \frac{T_q}{\sqrt{(T_q)^2 + (1-\rho^2)B_q^2}} - \rho = \frac{T_q - \rho\sqrt{(T_q)^2 + (1-\rho^2)B_q^2}}{\sqrt{(T_q)^2 + (1-\rho^2)B_q^2}}$$

$$= \frac{T_q - \rho\sqrt{(T_q)^2 + (1-\rho^2)B_q^2}}{\sqrt{(T_q)^2 + (1-\rho^2)B_q^2}} \cdot \frac{T_q + \rho\sqrt{(T_q)^2 + (1-\rho^2)B_q^2}}{T_q + \rho\sqrt{(T_q)^2 + (1-\rho^2)B_q^2}}$$

$$= \frac{(1-\rho^2)(T_q)^2 - \rho^2(1-\rho^2)B_q^2}{\sqrt{(T_q)^2 + (1-\rho^2)B_q^2}} \cdot \frac{1}{T_q + \rho\sqrt{(T_q)^2 + (1-\rho^2)B_q^2}}.$$

□

*Proof of Theorem A.21.* We will use notation and results from Lemma A.22. From the Law of Large Numbers, we have

$$\frac{T_q}{\sqrt{q}} = \rho\sqrt{\frac{1}{q}\sum_{j=1}^{q} N_j^2} + q^{-1/2}M_1\sqrt{1-\rho^2} \to \rho \cdot 1 + 0 \cdot \sqrt{1-\rho^2} = \rho \qquad (42)$$

almost surely, as $q \to \infty$. Using the same arguments, we have $\frac{B_q}{\sqrt{q}} \to 1$ almost surely. Hence,

$$q^{-1/2}\sqrt{(T_q)^2 + (1-\rho^2)B_q^2} = \sqrt{\left(\frac{T_q}{\sqrt{q}}\right)^2 + (1-\rho^2)\left(\frac{B_q}{\sqrt{q}}\right)^2}$$

$$\to \sqrt{\rho^2 + (1-\rho^2) \cdot 1^2} = 1 \qquad (43)$$

almost surely as $q \to \infty$. (42) and (43) now imply

$$q^{-1/2}(T_q + \rho\sqrt{(T_q)^2 + (1-\rho^2)B_q^2}) \to \rho + \rho = 2\rho. \qquad (44)$$

Using the Central Limit Theorem, we get

$$\frac{B_q^2 - q}{\sqrt{q}} = \frac{1}{\sqrt{q}}\sum_{j=2}^{q}(M_j^2 - 1) - \frac{1}{\sqrt{q}} \xrightarrow{d} \mathcal{N}(0, 2),$$

$$\frac{T_q^2 - q\rho^2}{\sqrt{q}} = \rho^2\frac{\|N\|^2 - q}{\sqrt{q}} + 2\rho\sqrt{1-\rho^2}\frac{\|N\|}{\sqrt{q}}M_1 + (1-\rho^2)\frac{M_1^2}{\sqrt{q}}$$

$$\xrightarrow{d} \rho^2\mathcal{N}(0, 2) + 2\rho\sqrt{1-\rho^2}\mathcal{N}(0, 1) + 0 = \mathcal{N}(0, 4\rho^2 - 2\rho^4).$$

Now, we have

$$\sqrt{q}\left[\frac{(Rx, Ry)}{\|Rx\|\|Ry\|} - \rho\right]$$

$$= \frac{\frac{T_q^2 - q\rho^2}{\sqrt{q}} - \rho^2\frac{B_q^2 - q}{\sqrt{q}}}{q^{-1/2}\sqrt{(T_q)^2 + (1-\rho^2)B_q^2}} \cdot \frac{(1-\rho^2)}{q^{-1/2}(T_q + \rho\sqrt{(T_q)^2 + (1-\rho^2)B_q^2})}$$

$$= \frac{\mathcal{N}(0, 4\rho^2 - 2\rho^4) - \rho^2\mathcal{N}(0, 2)}{1} \cdot \frac{1-\rho^2}{2\rho} = (1-\rho^2)\frac{\mathcal{N}(0, 4\rho^2 - 2\rho^4) + \mathcal{N}(0, 2\rho^4)}{2\rho}$$

$$= (1-\rho^2)\frac{\mathcal{N}(0, 4\rho^2)}{2\rho} = (1-\rho^2)\mathcal{N}(0, 1) = \mathcal{N}(0, (1-\rho^2)^2).$$

□

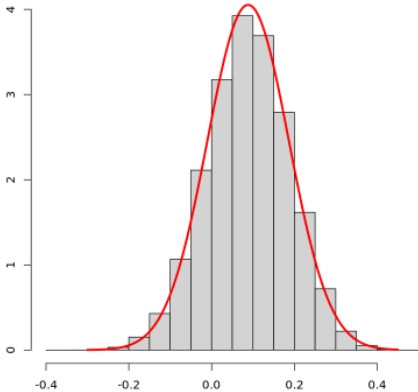

Figure 4: Simulated random projection for $n = q = 100$, where $(x, y)/\|x\|/\|y\| = \rho = 0.154$

For practical purposes, we interpret the result as given in (16) - that for *large* $q$ we have the normal distribution with small variance. Hence, it is likely that the cosine similarity of randomly projected vectors will be close to the one we would get without the random projection.

From simulations, see Figure 4, even for $q = 100$ we have *approximately* normal behavior, but the variance is large and the value might not be close to the original similarity. The result of Theorem A.21 will be more useful for larger values of $q$.

The same result holds for the empirical correlation coefficient in linear regression and in this case it is known that this is not a practical result (see Example 3.6. in (van der Vaart, 1998)). However, we are in a different setting here and the numerical simulations suggest that this result is stable and can be used in practice even for moderately large values of $q$. For example, for values of $q \geq 6400$ the error will be $\pm 0.05$ with the estimated probability of at least $0.95$. Concentration results to follow will further confirm this.

### A.IV.3    CONCENTRATION RESULT

Theorem A.21 guarantees that for large $q$ the cosine similarity random projection will be almost preserved with high probability for any pair of vectors in $\mathbb{R}^n$. However, this approach is not feasible to analyze the behavior of all pairs of vectors in a given set $y_1, \ldots, y_k$, as the computation becomes intractable and hard even for small values of $k$. If we want to get some guarantees that the similarity of all pairs of vectors is preserved within a small error with high probability under random projection, we need to use concentration inequalities.

The following theorem is the main result of this subsection.

**Theorem A.23.** *Let $x$ and $y$ be non-zero vectors in $\mathbb{R}^n$ and $\varepsilon \in (0, 0.055)$, then*

$$\mathbb{P}\left(\left|\frac{(Rx, Ry)}{\|Rx\|\|Ry\|} - \rho\right| \geq \varepsilon(1 - \rho^2)\right) \leq (4 + \varepsilon^2)\left[1 + \frac{\varepsilon^2}{2(1 + \varepsilon\sqrt{2})}\right]^{-\frac{q}{2}}$$

Recall the representation from Theorem A.6: For a given random $R$, for any vectors $x$ and $y$ in $\mathbb{R}^n$ there exist standard $q$-dimensional Gaussian vectors $N = N^{x,y}$ and $M = M^{x,y}$ such that

$$\frac{(Rx, Ry)}{\|Rx\|\|Ry\|} = \frac{\rho\|N\| + M_1\sqrt{1 - \rho^2}}{\sqrt{(\rho\|N\| + M_1\sqrt{1 - \rho^2})^2 + (1 - \rho^2)\|M_{2\ldots q}\|^2}}.$$

BOUNDS ON TAILS

We will first determine some tail bounds. We start with the following lemma.

**Lemma A.24.** *We have*

$$\mathbb{P}\left(\frac{M_1}{\|M_{2\ldots q}\|} > \varepsilon\right) = \mathbb{P}\left(\frac{M_1}{\|M_{2\ldots q}\|} < -\varepsilon\right) \leq \frac{1}{(1 + \varepsilon^2)^{\frac{q-1}{2}}}$$

*Proof.* Note that $M_1 \sim \mathcal{N}(0,1)$ and $\|M_{2\ldots q}\|^2 \sim \chi^2(q-1)$ are independent. The equality $\mathbb{P}\left(\frac{M_1}{\|M_{2\ldots q}\|} > \varepsilon\right) = \mathbb{P}\left(\frac{M_1}{\|M_{2\ldots q}\|} < -\varepsilon\right)$ follows from the fact that both $M_1$ and $-M_1$ have the same distribution. Further, by Markov's inequality we get

$$\mathbb{P}\left(\frac{M_1}{\|M_{2\ldots q}\|} > \varepsilon\right) = \mathbb{E}[\mathbb{P}\left(M_1 > \varepsilon\|M_{2\ldots q}\|\big| \|M_{2\ldots q}\|\right)]$$

$$\leq \mathbb{E}[\exp\left(-\varepsilon^2\|M_{2\ldots q}\|^2/2\right)] = \frac{1}{(1+\varepsilon^2)^{\frac{q-1}{2}}}.$$

The last equality follows from the fact that $\|M_{2\ldots q}\|^2$ has the $\chi^2(q-1)$ distribution and the Laplace transform for this distribution. $\qquad\square$

**Lemma A.25.** *For $\varepsilon > 0$ we have:*

$$\mathbb{P}\left(\frac{\|N\|}{\|M_{2\ldots q}\|} > \sqrt{1+\varepsilon}\right) \leq \sqrt{1+\frac{\varepsilon}{2}}\left[1+\frac{\varepsilon^2}{4(1+\varepsilon)}\right]^{-\frac{q}{2}} \tag{45}$$

$$\mathbb{P}\left(\frac{\|N\|}{\|M_{2\ldots q}\|} < \sqrt{1-\varepsilon}\right) \leq \sqrt{1-\frac{\varepsilon}{2}}\left[1+\frac{\varepsilon^2}{4(1-\varepsilon)}\right]^{-\frac{q}{2}} \tag{46}$$

*Proof.* Using Markov inequality, we have for all $\lambda \in (0, 1/2)$:

$$\mathbb{P}\left(\frac{\|N\|^2}{\|M_{2\ldots q}\|^2} > 1+\varepsilon\right) = \mathbb{P}(\lambda\|N\|^2 > \lambda(1+\varepsilon)\|M_{2\ldots q}\|^2)$$

$$= \mathbb{P}(\lambda\|N\|^2 - \lambda(1+\varepsilon)\|M_{2\ldots q}\|^2 > 0) = \mathbb{P}(\exp[\lambda\|N\|^2 - \lambda(1+\varepsilon)\|M_{2\ldots q}\|^2] > 1)$$

$$\leq \mathbb{E}[e^{\lambda\|N\|^2 - \lambda(1+\varepsilon)\|M_{2\ldots q}\|^2}] \leq \frac{1}{(1-2\lambda)^{\frac{q}{2}}} \cdot \frac{1}{(1+2(1+\varepsilon)\lambda)^{\frac{q-1}{2}}}.$$

Note that,

$$(1-2\lambda)(1+2(1+\varepsilon)\lambda)$$
$$= 1 + 2\varepsilon\lambda - 4(1+\varepsilon)\lambda^2$$
$$= 1 + \frac{\varepsilon^2}{4(1+\varepsilon)} - \left(\frac{\varepsilon}{2\sqrt{1+\varepsilon}} - 2\sqrt{1+\varepsilon}\lambda\right)^2$$

By setting $\lambda = \frac{\varepsilon}{4(1+\varepsilon)}$ we get

$$\frac{1}{(1-2\lambda)^{\frac{q}{2}}} \cdot \frac{1}{(1+2(1+\varepsilon)\lambda)^{\frac{q-1}{2}}}$$
$$= (1+2(1+\varepsilon)\lambda)^{\frac{1}{2}}[(1-2\lambda)(1+2(1+\varepsilon)\lambda)]^{-\frac{q}{2}}$$
$$= \sqrt{1+\frac{\varepsilon}{2}}\left[1+\frac{\varepsilon^2}{4(1+\varepsilon)}\right]^{-\frac{q}{2}}$$

This proves (45). In the similar way we prove (46):

$$\mathbb{P}\left(\frac{\|N\|^2}{\|M_{2\ldots q}\|^2} < 1-\varepsilon\right) = \mathbb{P}(\lambda\|N\|^2 < \lambda(1-\varepsilon)\|M_{2\ldots q}\|^2)$$

$$= \mathbb{P}(\lambda(1-\varepsilon)\|M_{2\ldots q}\|^2 - \lambda\|N\|^2 > 0) = \mathbb{P}(\exp[\lambda(1-\varepsilon)\|M_{2\ldots q}\|^2 - \lambda\|N\|^2] > 1)$$

$$\leq \mathbb{E}[e^{\lambda(1-\varepsilon)\|M_{2\ldots q}\|^2 - \lambda\|N\|^2}] \leq \frac{1}{(1+2\lambda)^{\frac{q}{2}}} \cdot \frac{1}{(1-2(1-\varepsilon)\lambda)^{\frac{q-1}{2}}}.$$

Note that,

$$(1+2\lambda)(1-2(1-\varepsilon)\lambda)$$
$$= 1 + 2\varepsilon\lambda - 4(1-\varepsilon)\lambda^2$$
$$= 1 + \frac{\varepsilon^2}{4(1-\varepsilon)} - \left(\frac{\varepsilon}{2\sqrt{1-\varepsilon}} - 2\sqrt{1-\varepsilon}\lambda\right)^2$$

By setting $\lambda = \frac{\varepsilon}{4(1-\varepsilon)}$ we get

$$\frac{1}{(1+2\lambda)^{\frac{p}{2}}} \cdot \frac{1}{(1 - 2(1-\varepsilon)\lambda)^{\frac{q-1}{2}}}$$
$$= (1 - 2(1-\varepsilon)\lambda)^{\frac{1}{2}} [(1+2\lambda)(1 - 2(1-\varepsilon)\lambda)]^{-\frac{q}{2}}$$
$$= \sqrt{1 - \frac{\varepsilon}{2}} \left[ 1 + \frac{\varepsilon^2}{4(1-\varepsilon)} \right]^{-\frac{q}{2}}$$

This completes the proof. $\square$

**Corollary A.26.** *We have*

$$\mathbb{P}\left( \frac{\|N\|}{\|M_{2...q}\|} > \sqrt{1+\varepsilon} \right) + \mathbb{P}\left( \frac{\|N\|}{\|M_{2...q}\|} < \sqrt{1-\varepsilon} \right) \leq 2 \left[ 1 + \frac{\varepsilon^2}{4(1+\varepsilon)} \right]^{-\frac{q}{2}} \qquad (47)$$

*Proof.* Using (45), (46) and the Cauchy-Schwarz inequality we have

$$\mathbb{P}\left( \frac{\|N\|}{\|M_{2...q}\|} > \sqrt{1+\varepsilon} \right) + \mathbb{P}\left( \frac{\|N\|}{\|M_{2...q}\|} < \sqrt{1-\varepsilon} \right)$$
$$\leq \sqrt{1 + \frac{\varepsilon}{2}} \left[ 1 + \frac{\varepsilon^2}{4(1+\varepsilon)} \right]^{-\frac{q}{2}} + \sqrt{1 - \frac{\varepsilon}{2}} \left[ 1 + \frac{\varepsilon^2}{4(1-\varepsilon)} \right]^{-\frac{q}{2}}$$
$$\leq \sqrt{1 + \frac{\varepsilon}{2} + 1 - \frac{\varepsilon}{2}} \cdot \sqrt{\left[ 1 + \frac{\varepsilon^2}{4(1+\varepsilon)} \right]^{-q} + \left[ 1 + \frac{\varepsilon^2}{4(1-\varepsilon)} \right]^{-q}}$$
$$\leq \sqrt{2} \cdot \sqrt{2 \left[ 1 + \frac{\varepsilon^2}{4(1+\varepsilon)} \right]^{-q}} = 2 \left[ 1 + \frac{\varepsilon^2}{4(1+\varepsilon)} \right]^{-\frac{q}{2}}$$

$\square$

PROOF OF THE CONCENTRATION RESULT

We need one more lemma to have everything for the proof.

**Lemma A.27.** *Given* $\varepsilon \in (0, 0.08]$, $\left| \frac{M_1}{\|M_{2...q}\|} \right| < \varepsilon/2$ *and* $\sqrt{1-\varepsilon} < \frac{\|N\|}{\|M_{2...q}\|} < \sqrt{1+\varepsilon}$, *then*

$$\left| \frac{(Rx, Ry)}{\|Rx\|\|Ry\|} - \rho \right| \leq \frac{\varepsilon}{\sqrt{2}}(1 - \rho^2).$$

*Proof.* Recall, we used $\phi(s) = \frac{s}{\sqrt{1-s^2}}$ and let us define $\psi(h) := \phi^{-1}(h) = \frac{h}{\sqrt{1+h^2}}$. Note that $\psi'(h) = \frac{1}{(1+h^2)^{3/2}} > 0$, hence $\psi$ is an increasing function. We have

$$\frac{(Rx, Ry)}{\|Rx\|\|Ry\|} = \psi\left( \frac{\rho\|N\| + M_1\sqrt{1-\rho^2}}{\|M_{2...q}\|\sqrt{1-\rho^2}} \right) = \psi\left( \frac{\rho}{\sqrt{1-\rho^2}} \frac{\|N\|}{\|M_{2...q}\|} + \frac{M_1}{\|M_{2...q}\|} \right).$$

and

$$\rho = \psi\left( \frac{\rho}{\sqrt{1-\rho^2}} \right).$$

We will prove the claim for the case when $\rho \geq 0$; the other case is proven in a similar way. Using non-negativity of $\rho$ and the fact that $\psi$ is increasing, we have

$$\psi\left(\frac{\rho}{\sqrt{1-\rho^2}}\sqrt{1-\varepsilon} - \frac{\varepsilon}{2}\right) - \psi\left(\frac{\rho}{\sqrt{1-\rho^2}}\right)$$
$$\leq \frac{(Rx, Ry)}{\|Rx\|\|Ry\|} - \rho \leq$$
$$\psi\left(\frac{\rho}{\sqrt{1-\rho^2}}\sqrt{1+\varepsilon} + \frac{\varepsilon}{2}\right) - \psi\left(\frac{\rho}{\sqrt{1-\rho^2}}\right). \quad (48)$$

Using Lemma A.42 (a) we can estimate the upper bound:

$$\psi\left(\frac{\rho}{\sqrt{1-\rho^2}}\sqrt{1+\varepsilon} + \frac{\varepsilon}{2}\right) - \psi\left(\frac{\rho}{\sqrt{1-\rho^2}}\right)$$
$$\leq \left(1 + \frac{\rho^2}{1-\rho^2}\right)^{-3/2} \cdot \left[\frac{\rho}{\sqrt{1-\rho^2}}(\sqrt{1+\varepsilon} - 1) + \frac{\varepsilon}{2}\right]$$
$$\leq (1-\rho^2)^{3/2}\left[\frac{\rho}{\sqrt{1-\rho^2}}\frac{\varepsilon}{2} + \frac{\varepsilon}{2}\right]$$
$$\leq (1-\rho^2)\frac{\varepsilon(\rho + \sqrt{1-\rho^2})}{2}. \quad (49)$$

Since $\rho + \sqrt{1-\rho^2} \leq \sqrt{2}$, we have

$$(49) \leq (1-\rho^2)\frac{\varepsilon}{\sqrt{2}}. \quad (50)$$

To prove the lower bound in (48), we first need to note that $\frac{\rho}{\sqrt{1-\rho^2}}\sqrt{1-\varepsilon} - \frac{\varepsilon}{2}$ can be negative or very close to $0$. For that reason, we will have to consider two cases: $\rho^2 \leq \varepsilon$ or $\rho^2 > \varepsilon$.

If $\rho^2 > \varepsilon$,

$$\frac{\rho}{\sqrt{1-\rho^2}}\sqrt{1-\varepsilon} \geq \sqrt{\varepsilon} > \varepsilon. \quad (51)$$

we have

$$\frac{\rho}{\sqrt{1-\rho^2}}\sqrt{1-\varepsilon} - \frac{\varepsilon}{2} \geq \frac{\varepsilon}{2} > 0, \quad (52)$$

and in particular

$$\rho\sqrt{1-\varepsilon} - \frac{\varepsilon}{2}\sqrt{1-\rho^2} > 0. \quad (53)$$

Let

$$A := \psi\left(\frac{\rho}{\sqrt{1-\rho^2}}\right) = \rho > 0,$$

and

$$B := \psi\left(\frac{\rho}{\sqrt{1-\rho^2}}\sqrt{1-\varepsilon} - \frac{\varepsilon}{2}\right) > 0.$$

From (52) we have $B \geq 0$, and hence since $\psi$ is an increasing function we have $A > B > 0$.

We obtain

$$A - B = \rho - \frac{\rho\sqrt{1-\varepsilon} - \varepsilon/2\sqrt{1-\rho^2}}{\sqrt{(\rho\sqrt{1-\varepsilon} - \varepsilon/2\sqrt{1-\rho^2})^2 + (1-\rho^2)}}$$

$$\overset{\text{Lemma A.40.}}{\leq} \rho - \frac{\rho\sqrt{1-\varepsilon} - \varepsilon/2\sqrt{1-\rho^2}}{\sqrt{1-\varepsilon\rho^2}} = \rho\underbrace{\left(1 - \frac{\sqrt{1-\varepsilon}}{\sqrt{1-\varepsilon\rho^2}}\right)}_{\leq 0} + \frac{\varepsilon\sqrt{1-\rho^2}}{2\sqrt{1-\varepsilon\rho^2}}$$

$$\leq \frac{\varepsilon\sqrt{1-\rho^2}}{2\sqrt{1-\varepsilon\rho^2}} \leq \frac{\varepsilon\sqrt{1-\rho^2}}{2\sqrt{1-\varepsilon\rho^2}} \cdot \frac{\sqrt{1-\rho^2}}{\sqrt{1-\varepsilon\rho^2}} \leq \frac{\varepsilon(1-\rho^2)}{2(1-\varepsilon\rho^2)} \leq \frac{\varepsilon(1-\rho^2)}{2(1-\varepsilon)} \tag{54}$$

Since $\varepsilon \in (0, \frac{1}{4}]$, we have $\frac{1}{2(1-\varepsilon)} \leq \frac{2}{3} < \frac{\sqrt{2}}{2} = \frac{1}{\sqrt{2}}$ and hence

$$(54) \leq \frac{\varepsilon(1-\rho^2)}{\sqrt{2}} \tag{55}$$

On the other hand, if $\rho^2 \leq \varepsilon$, Lemma A.42 (b) implies:

$$\psi\left(\frac{\rho}{\sqrt{1-\rho^2}}\right) - \psi\left(\frac{\rho}{\sqrt{1-\rho^2}}\sqrt{1-\varepsilon} - \frac{\varepsilon}{2}\right)$$

$$\leq \left[\frac{\rho}{\sqrt{1-\rho^2}}(1-\sqrt{1-\varepsilon}) + \frac{\varepsilon}{2}\right] = \left[\frac{\rho}{\sqrt{1-\rho^2}} \cdot \frac{\varepsilon}{1+\sqrt{1-\varepsilon}} + \frac{\varepsilon}{2}\right]$$

$$= \varepsilon\left[\frac{\rho}{\sqrt{1-\rho^2}}\frac{1}{1+\sqrt{1-\varepsilon}} + \frac{1}{2} + \frac{\rho^2}{\sqrt{2}}\right] - \frac{\varepsilon}{\sqrt{2}}\rho^2. \tag{56}$$

Since $\rho^2 \leq \varepsilon$, we have $\rho \in [0, \varepsilon]$ and

$$\frac{\rho}{\sqrt{1-\rho^2}}\frac{1}{1+\sqrt{1-\varepsilon}} + \frac{1}{2} + \frac{\rho^2}{\sqrt{2}} \leq \frac{\sqrt{\varepsilon}}{\sqrt{1-\varepsilon}+1-\varepsilon} + \frac{1}{2} + \frac{\varepsilon}{\sqrt{2}}.$$

This expression is increasing for $\varepsilon \in (0, 1)$, so

$$\leq \frac{\sqrt{\varepsilon}}{\sqrt{1-\varepsilon}+1-\varepsilon} + \frac{1}{2} + \frac{\varepsilon}{\sqrt{2}}\Big|_{\varepsilon=0.08} \approx 0.70708 < \frac{1}{\sqrt{2}}.$$

Therefore,

$$(56) \leq \frac{\varepsilon(1-\rho^2)}{\sqrt{2}} \tag{57}$$

Finally, the inequalities (50), (55) and (57) prove the claim. $\qquad \square$

*Proof of Theorem A.23.* Since $\varepsilon \in (0, 0.055]$, we have $\varepsilon\sqrt{2} \in (0, 0.08]$, then by Lemma A.27 we have

$$\mathbb{P}\left(\left|\frac{(Rx, Ry)}{\|Rx\|\|Ry\|} - \rho\right| \geq \varepsilon(1-\rho^2)\right) = \mathbb{P}\left(\left|\frac{(Rx, Ry)}{\|Rx\|\|Ry\|} - \rho\right| \geq \frac{\varepsilon\sqrt{2}}{\sqrt{2}}(1-\rho^2)\right)$$

$$\leq \mathbb{P}\left(\frac{M_1}{\|M_{2\ldots q}\|} > \frac{\varepsilon\sqrt{2}}{2}\right) + \mathbb{P}\left(\frac{M_1}{\|M_{2\ldots q}\|} < -\frac{\varepsilon\sqrt{2}}{2}\right)$$

$$+ \mathbb{P}\left(\frac{\|N\|}{\|M_{2\ldots q}\|} > \sqrt{1+\varepsilon\sqrt{2}}\right) + \mathbb{P}\left(\frac{\|N\|}{\|M_{2\ldots q}\|} < \sqrt{1-\varepsilon\sqrt{2}}\right).$$

Using Lemma A.24 and Corollary A.26 we have

$$\leq 2 \left[ 1 + \frac{\varepsilon^2}{2} \right]^{-\frac{q-1}{2}} + 2 \left[ 1 + \frac{\varepsilon^2}{4(1 + \varepsilon\sqrt{2})} \right]^{-\frac{q}{2}}$$

$$= (2 + \varepsilon^2) \left[ 1 + \frac{\varepsilon^2}{2} \right]^{-\frac{q}{2}} + 2 \left[ 1 + \frac{\varepsilon^2}{4(1 + \varepsilon\sqrt{2})} \right]^{-\frac{q}{2}}$$

$$\leq (2 + \varepsilon^2) \left[ 1 + \frac{\varepsilon^2}{4(1 + \varepsilon\sqrt{2})} \right]^{-\frac{q}{2}} + 2 \left[ 1 + \frac{\varepsilon^2}{4(1 + \varepsilon\sqrt{2})} \right]^{-\frac{q}{2}}$$

$$= (4 + \varepsilon^2) \left[ 1 + \frac{\varepsilon^2}{4(1 + \varepsilon\sqrt{2})} \right]^{-\frac{q}{2}}.$$

$\square$

### A.IV.4 JOHNSON-LINDENSTRAUSS-TYPE RESULT

**Theorem A.28.** *Let $\varepsilon \in (0, 0.05]$, $\delta \in (0, 1)$ and $p_1, \ldots, p_k$ be non-zero vectors in $\mathbb{R}^n$. If*

$$q \geq \frac{2 \ln \left[ \frac{2k(k-1)\left(1 + \frac{\varepsilon^2}{4}\right)}{\delta} \right]}{\ln \left[ 1 + \frac{\varepsilon^2}{2(1+\varepsilon\sqrt{2})} \right]} \tag{58}$$

*then, with probability $1 - \delta$, the inequality*

$$\left| \frac{(Rp_j, Rp_i)}{\|Rp_j\|\|Rp_i\|} - \frac{(p_j, p_i)}{\|p_j\|\|p_i\|} \right| \leq \varepsilon(1 - \rho^2)$$

*holds for all $i < j$.*

*Proof.* Define

$$W_{ij} := \left( \left| \frac{(Rp_j, Rp_i)}{\|Rp_j\|\|Rp_i\|} - \frac{(p_j, p_i)}{\|p_j\|\|p_i\|} \right| \leq \varepsilon(1 - \rho^2) \right).$$

To prove the theorem, it suffices to show that $\mathbb{P}\left( \bigcup_{i<j} W_{ij}^c \right) < \delta$.

By Theorem A.23, we have

$$\mathbb{P}\left( \bigcup_{i<j} W_{ij}^c \right) = \sum_{i<j} \mathbb{P}\left( W_{ij}^c \right) = \frac{k(k-1)}{2} \cdot 4 \left( 1 + \frac{\varepsilon^2}{4} \right) \cdot \left[ 1 + \frac{\varepsilon^2}{2(1+\varepsilon\sqrt{2})} \right]^{-\frac{q}{2}}.$$

This expression is less than $\delta$ by the choice of $q$ in (60). $\square$

### A.IV.5 RESULT COMPARISON ANALYSIS

In this section, we state Lemma 5 from (Arpit et al., 2014), which gives a similar result to Theorem A.23, and compare the two results.

**Lemma A.29.** *For $x$ and $y$ non-zero vectors in $\mathbb{R}^n$ and for every $\beta \in (0, 1/2)$, we have*

$$-\frac{\beta}{1-\beta}(1-\rho) \leq \frac{(Rx, Ry)}{\|Rx\|\|Ry\|} - \rho \leq \frac{\beta}{1+\beta}(1-\rho)$$

*if $\rho \leq -\varepsilon$,*

$$-\frac{\beta}{1-\beta}(1-\rho) \leq \frac{(Rx, Ry)}{\|Rx\|\|Ry\|} - \rho \leq \frac{\beta}{1-\beta}(1+\rho)$$

*if $-\varepsilon \leq \rho \leq \varepsilon$,*

$$-\frac{\beta}{1+\beta}(1+\rho) \leq \frac{(Rx, Ry)}{\|Rx\|\|Ry\|} - \rho \leq \frac{\beta}{1-\beta}(1+\rho)$$

*if $\rho \geq \varepsilon$. Moreover, the inequality holds true with probability at least $1 - 8e^{-\frac{q}{4}(\beta^2 - \beta^2)}$.*

**Remark A.30.** *We observe that Lemma A.29 does not capture the fact that when $\rho = \pm 1$, the random projection preserves the cosine similarity, as discussed in §A.IV.1.*

*Also, the role of $\beta$ is unclear. If we set $\varepsilon = \max\{\frac{\beta}{1-\beta}, \frac{\beta}{1+\beta}\} = \frac{\beta}{1-\beta}$, the result can be rewritten as*

$$-\varepsilon(1 + |\rho|) \leq \frac{(Rx, Ry)}{\|Rx\|\|Ry\|} - \rho \leq \varepsilon(1 + |\rho|), \tag{59}$$

*with probability at least $1 - 8e^{-\frac{q}{4} \cdot \frac{\varepsilon^2}{(1+\varepsilon)^2}}$.*

*We note that the inequality (59) is less precise than the one provided by Theorem A.23.*

*Moreover, by Lemma A.41, we have $\left[1 + \frac{\varepsilon^2}{2(1+\varepsilon\sqrt{2})}\right]^{-\frac{q}{2}} \leq e^{-\frac{q}{4} \cdot \frac{\varepsilon^2}{(1+\varepsilon)^2}}$. Therefore, the inequality in Theorem A.23 also holds with higher probability.*

*We remark that Theorem A.23 requires $\varepsilon \in (0, 0.05]$, but this is a reasonable absolute error for cosine similarity, which ranges from 0 to 1.*

We will see how the dimension selection in Theorem A.28 for cosine similarity compares with the similar result in original Johnson-Lindenstrauss Lemma. The following version is based on Theorem 2.13 and Remark 2.11 from (Boucheron et al., 2013) and we present it without proof.

**Theorem A.31.** *Let $\varepsilon, \delta \in (0, 1)$, and $p_1, \ldots, p_k$ be non-zero vectors in $\mathbb{R}^n$. If*

$$q \geq \frac{4}{\varepsilon^2} \log\left[\frac{k^2}{\delta}\right] \tag{60}$$

*then, with probability at least $1 - \delta$, the inequality*

$$(1 - \varepsilon)\|p_i - p_j\|^2 \leq \|R(p_i - p_j)\|^2 \leq (1 + \varepsilon)\|p_i - p_j\|^2 \tag{61}$$

*holds for all $i < j$.*

In the following we will use the notation $f \sim g$ that will denote

$$\lim_{\varepsilon \to 0^+} \frac{f(\varepsilon)}{g(\varepsilon)} = 1.$$

This is a standard tool to analyze asymptotic behavior.

**Proposition A.32.** *For a fixed, $\delta$ and $k$ we have*

$$\frac{2 \ln\left[\frac{2k(k-1)\left(1+\frac{\varepsilon^2}{4}\right)}{\delta}\right]}{\ln\left[1 + \frac{\varepsilon^2}{2(1+\varepsilon\sqrt{2})}\right]} \sim \frac{4}{\varepsilon^2} \ln\left[\frac{2k(k-1)}{\delta}\right].$$

*Proof.* Using L'Hospital's rule one can show $\lim_{x \to 0^+} \frac{\log(1+x)}{x} = 1$, hence

$$\lim_{\varepsilon \to 0^+} \frac{\ln\left[1 + \frac{\varepsilon^2}{2(1+\varepsilon\sqrt{2})}\right]}{\frac{\varepsilon^2}{2(1+\varepsilon\sqrt{2})}} = 1. \tag{62}$$

Simple limit calculus gives us

$$\lim_{\varepsilon \to 0^+} \frac{\frac{\varepsilon^2}{2(1+\varepsilon\sqrt{2})}}{\frac{\varepsilon^2}{2}} = \lim_{\varepsilon \to 0^+} \frac{1}{1 + \varepsilon\sqrt{2}} = 1. \tag{63}$$

Hence, multiplying expressions in (62) and (63) we get:

$$\lim_{\varepsilon \to 0^+} \frac{\ln\left[1 + \frac{\varepsilon^2}{2(1+\varepsilon\sqrt{2})}\right]}{\frac{\varepsilon^2}{2}} = 1. \tag{64}$$

Furthermore, by continuity of the function $\ln$ we have

$$\lim_{\varepsilon \to 0^+} \ln \left[ \frac{2k(k-1)\left(1+\frac{\varepsilon^2}{4}\right)}{\delta} \right] = \ln \left[ \frac{2k(k-1)}{\delta} \right]. \tag{65}$$

The claim now follows from (64) and (65). $\qquad\square$

The comparison in Proposition A.32 tells us that the dimension $q$ will be of the same order but slightly higher. This is illustrated on Figure 5.

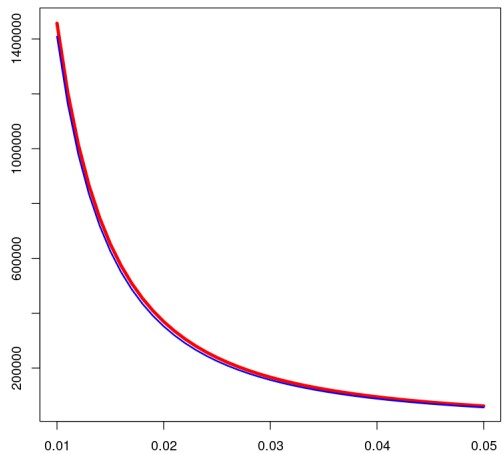

Figure 5: Simulated random projection for $k = 10,000,000$, $\delta = 0.05$ the upper (red) curve represents the graph of $\varepsilon \mapsto \dfrac{2\ln\left[\frac{2k(k-1)\left(1+\frac{\varepsilon^2}{4}\right)}{\delta}\right]}{\ln\left[1+\frac{\varepsilon^2}{2(1+\varepsilon\sqrt{2})}\right]}$ (minimum value of $q$ for cosine similarity) and the lower (blue) curve $\varepsilon \mapsto \frac{4}{\varepsilon^2} \ln\left[\frac{k^2}{\delta}\right]$ (minimum value of $q$ for Johnson-Lindenstrauss Lemma). We can see that the two curves are very close.

## A.V  TECHNICAL INEQUALITIES AND CONCENTRATION RESULTS

In this appendix, we placed some technical inequalities we used in other parts of the text, so that they can be verified by the reader.

### A.V.1  USEFUL CONCENTRATION INEQUALITIES

In this paper we will depend on the results from concentration inequality theory. The approach in this paper is adapted based on the book (Boucheron et al., 2013). In this subsection we adapted results from Chapter 2 of the book.

**Theorem A.33.** *Let $X$ be a random variable such that $v > 0$, $c > 0$ we have*

$$\log \mathbb{E}[e^{\lambda(X - \mathbb{E}X)}] \le \frac{v\lambda^2}{2(1 - c\lambda)} \tag{66}$$

*for every $\lambda \in (0, c^{-1})$. Then for $t \ge 0$ we have*

$$\mathbb{P}(X - \mathbb{E}X > t) \le \exp\left(\frac{-t^2}{2(v + ct)}\right)$$

*Proof.* Using Markov inequality, for $\lambda \in (0, c^{-1})$ we have

$$\mathbb{P}(X - \mathbb{E}X > t) \leq \frac{\mathbb{E}[e^{\lambda(X - \mathbb{E}X)}]}{e^{\lambda t}} \leq \exp\left(\frac{v\lambda^2}{2(1 - c\lambda)} - \lambda t\right).$$

Hence,

$$\mathbb{P}(X - \mathbb{E}X > t) \leq \exp\left[-\sup_{\lambda \in (0, c^{-1})}\left(\lambda t - \frac{v\lambda^2}{2(1 - c\lambda)}\right)\right].$$

Using usual calculus techniques we get (see Lemma A.39 for details)

$$\sup_{\lambda \in (0, c^{-1})}\left(\lambda t - \frac{v\lambda^2}{2(1 - c\lambda)}\right) = \frac{v}{c^2}h\left(\frac{ct}{v}\right),$$

where $h(s) = 1 + s - \sqrt{1 + 2s}$. By Lemma A.37 we have $h(s) \geq \frac{s^2}{2(1+s)}$ for $s > 0$ and the claim follows. $\qquad\square$

The following Corollary explains what happens when $c = 0$ in (66).

**Theorem A.34.** *Let $X$ be a random variable such that $v > 0$ we have*

$$\log \mathbb{E}[e^{\lambda(X - \mathbb{E}X)}] \leq \frac{v\lambda^2}{2} \tag{67}$$

*for every $\lambda \geq 0$. Then for $t \geq 0$ we have*

$$\mathbb{P}(X - \mathbb{E}X > t) \leq \exp\left(\frac{-t^2}{2v}\right).$$

*Proof.* Using Markov inequality, for $\lambda \geq 0$ we have

$$\begin{aligned}
\mathbb{P}(X - \mathbb{E}X > t) &\leq \frac{\mathbb{E}[e^{\lambda(X - \mathbb{E}X)}]}{e^{\lambda t}} \leq \exp\left(\frac{v\lambda^2}{2} - \lambda t\right) \\
&= \exp\left(\frac{v\lambda^2}{2} - \lambda t + \frac{t^2}{2v} - \frac{t^2}{2v}\right) \\
&= \exp\left(\frac{1}{2}\left(\lambda v^{1/2} - tv^{-1/2}\right)^2 - \frac{t^2}{2v}\right).
\end{aligned}$$

Setting $\lambda = t/v$ the claim follows. $\qquad\square$

**Proposition A.35.** *If for a given $p > 0$ and $c > 0$ the inequality*

$$\mathbb{E}[e^{\lambda(X - \mathbb{E}X)}] \leq \frac{e^{-\lambda p}}{(1 - c\lambda)^{p/c}} \tag{68}$$

*holds for all $\lambda < c^{-1}$, then the following claims hold:*

*(a) For all $\lambda \in (0, c^{-1})$*

$$\log \mathbb{E}[e^{\lambda(X - \mathbb{E}X)}] \leq \frac{pc\lambda^2}{2(1 - c\lambda)}. \tag{69}$$

*(b) For all $\lambda \leq 0$*

$$\log \mathbb{E}[e^{\lambda(X - \mathbb{E}X)}] \leq \frac{pc\lambda^2}{2}. \tag{70}$$

*Proof.* (a) Using the inequality $-\log(1 - s) - s \leq \frac{s^2}{2(1-s)}$ for $s \in (0, 1)$ (see Lemma A.38 (a)) we have:

$$\begin{aligned}
\log \mathbb{E}[e^{\lambda(X - \mathbb{E}X)}] &\leq -\lambda p - \frac{p}{c}\log(1 - c\lambda) \\
&= \frac{p}{c}(-\log(1 - c\lambda) - c\lambda) \leq \frac{p}{c} \cdot \frac{(c\lambda)^2}{2(1 - c\lambda)} = \frac{pc\lambda^2}{2(1 - c\lambda)}.
\end{aligned}$$

(b) Using inequality $-\log(1-s) - s \leq \frac{s^2}{2}$ for $s < 0$ (see Lemma A.38 (b)) we have:

$$\log \mathbb{E}[e^{\lambda(X - \mathbb{E}X)}] \leq -\lambda p - \frac{p}{c} \log(1 - c\lambda)$$

$$= \frac{p}{c}(-\log(1 - c\lambda) - c\lambda) \leq \frac{p}{c} \cdot \frac{(c\lambda)^2}{2} = \frac{pc\lambda^2}{2}.$$

$\square$

**Corollary A.36.** *If for a given $p > 0$ and $c > 0$ we have*

$$\log \mathbb{E}[e^{\lambda(X - \mathbb{E}X)}] \leq \frac{e^{-\lambda p}}{(1 - c\lambda)^{p/c}} \tag{71}$$

*for all $\lambda < c^{-1}$, then for all $t \geq 0$*

$$\mathbb{P}(X - \mathbb{E}X > t) \leq \exp\left(\frac{-t^2}{2(pc + ct)}\right), \tag{72}$$

*and*

$$\mathbb{P}(X - \mathbb{E}X < -t) \leq \exp\left(\frac{-t^2}{2pc}\right). \tag{73}$$

*Proof.* From Proposition A.35 we know that (69) holds. Now, by Theorem A.33, inequality (72) holds.

From Proposition A.35 we know that (70) holds. If we substitute $\lambda \mapsto -\lambda$ and $X \mapsto -X$ in (70) we have

$$\log \mathbb{E}[e^{\lambda(-X + \mathbb{E}X)}] \leq \frac{pc\lambda^2}{2}$$

for $\lambda \geq 0$. Hence, using Theorem A.35 we get (73). $\square$

### A.V.2 TECHNICAL INEQUALITIES

**Lemma A.37.** *For $s \geq 0$ we have*

$$1 + s - \sqrt{1 + 2s} \geq \frac{s^2}{2(1 + s)}.$$

*Proof.* For $s > 0$ we have:

$$1 + s - \sqrt{1 + 2s} = (1 + s - \sqrt{1 + 2s}) \cdot \frac{1 + s + \sqrt{1 + 2s}}{1 + s + \sqrt{1 + 2s}}$$

$$= \frac{(1 + s)^2 - 1 + 2s}{1 + s + \sqrt{1 + 2s}} = \frac{s^2}{1 + s + \sqrt{1 + 2s}} \geq \frac{s^2}{2(1 + s)}.$$

The last inequality follows from the fact that $1 + s = \sqrt{1 + 2s + s^2} > \sqrt{1 + 2s}$. $\square$

**Lemma A.38.** *(a) For $s \in (0, 1)$ we have $-\log(1 - s) - s \leq \frac{s^2}{2(1-s)}$.*

*(b) For $s < 0$ we have $-\log(1 - s) - s \leq \frac{s^2}{2}$.*

*Proof.* Not that for $f(s) = -\log(1 - s) - s$ we have $f'(s) = \frac{1}{1-s} - 1 = \frac{s}{1-s}$. Hence, in the case of (a), it follows for $s \in (0, 1)$

$$-\log(1 - s) - s = f(s) - f(0) = \int_0^s f'(t) \, dt$$

$$= \int_0^s \frac{t}{1 - t} \, dt \leq \int_0^s \frac{t}{1 - s} \, dt = \frac{1}{1 - s} \int_0^s t \, dt = \frac{s^2}{2(1 - s)}.$$

In the case of (b), for $s < 0$ we can apply a similar argument:

$$-\log(1-s) - s = f(s) - f(0) = \int_0^s f'(t)\,dt$$

$$= \int_0^s \frac{t}{1-t}\,dt \leq \int_s^0 \frac{-t}{1-t}\,dt \leq \int_s^0 -t\,dt = \frac{s^2}{2}.$$

$\square$

**Lemma A.39.** *For a given $t > 0$, $v > 0$ and $c > 0$ the function*

$$f(\lambda) = \lambda t - \frac{v\lambda^2}{2(1 - c\lambda)}$$

*has a maximum on interval $(0, c^{-1})$ at point $\lambda_{\max} = \frac{\sqrt{v+2tc} - \sqrt{v}}{c\sqrt{v+2tc}}$. and it equals*

$$\frac{v}{c^2}\left(1 + \frac{tc}{v} - \sqrt{1 + 2\cdot\frac{tc}{v}}\right).$$

*Proof.* When we take the first derivative we get

$$f'(\lambda) = t - \frac{v\lambda}{(1 - c\lambda)} - \frac{vc\lambda^2}{2(1 - c\lambda)^2}.$$

Setting $f'(\lambda) = 0$ and using the substitution $x = \frac{\lambda}{(1-c\lambda)}$, we get

$$-\frac{vc}{2}\cdot x^2 - vx + t = 0.$$

Solving this quadratic equation by $x$ we get

$$x_{1,2} = -\frac{1}{c} \pm \sqrt{\frac{1}{c^2} + \frac{2t}{vc}}.$$

Since the solution needs to be non-negative,

$$x = -\frac{1}{c} + \sqrt{\frac{1}{c^2} + \frac{2t}{vc}} = -\frac{1}{c} + \sqrt{\frac{v}{vc^2} + \frac{2tc}{vc^2}}$$

$$= \frac{-\sqrt{v} + \sqrt{v + 2tc}}{c\sqrt{v}} = \frac{\sqrt{v + 2tc} - \sqrt{v}}{c\sqrt{v}}.$$

Now, when we calculate $\lambda$ from $x$, we get

$$\lambda_{\max} = \frac{\sqrt{v + 2tc} - \sqrt{v}}{c\sqrt{v + 2tc}}.$$

Note that, $\lambda_{\max}$ is the minimum since $f'$ is a strictly decreasing function on $(0, c^{-1})$. The value of the minimum is

$$f'(\lambda_{\max}) = \lambda_{\max}\left(t - \frac{v}{2}\cdot\frac{\lambda_{\max}}{1 - c\lambda_{\max}}\right) = \lambda_{\max}\left(t - \frac{v}{2}\cdot x\right)$$

$$= \frac{\sqrt{v + 2tc} - \sqrt{v}}{c\sqrt{v + 2tc}}\left(t - \frac{v}{2}\cdot\frac{\sqrt{v + 2tc} - \sqrt{v}}{c\sqrt{v}}\right)$$

$$= \frac{\sqrt{v + 2tc} - \sqrt{v}}{c\sqrt{v + 2tc}}\left(t - \frac{\sqrt{v}\sqrt{v + 2tc} - v}{2c}\right)$$

$$= \frac{\sqrt{v + 2tc} - \sqrt{v}}{c\sqrt{v + 2tc}}\cdot\frac{2tc + v - \sqrt{v}\sqrt{v + 2tc}}{2c}$$

$$= \frac{(\sqrt{v + 2tc} - \sqrt{v})}{c}\cdot\frac{\sqrt{v + 2tc} - \sqrt{v}}{2c}$$

$$= \frac{(\sqrt{v + 2tc} - \sqrt{v})^2}{2c^2} = \frac{2v + 2tc - 2\sqrt{v^2 + 2vtc}}{2c^2}$$

$$= \frac{v}{c^2}\left(1 + \frac{tc}{v} - \sqrt{1 + 2\cdot\frac{tc}{v}}\right).$$

$\square$

**Lemma A.40.** *For $\varepsilon \in (0, 1)$ and $\rho^2 \in [\varepsilon, 1]$ we have*

$$(\rho\sqrt{1-\varepsilon} - \frac{\varepsilon}{2}\sqrt{1-\rho^2})^2 + 1 - \rho^2 \leq 1 - \varepsilon\rho^2.$$

*Proof.* It is easy to show that $\rho \geq \varepsilon$ and $1 - \varepsilon \geq 1 - \rho^2$, hence

$$0 \leq \rho\sqrt{1-\varepsilon} - \frac{\varepsilon}{2}\sqrt{1-\rho^2} \leq \rho\sqrt{1-\varepsilon}. \tag{74}$$

Now, we have:

$$(\rho\sqrt{1-\varepsilon} - \frac{\varepsilon}{2}\sqrt{1-\rho^2})^2 + 1 - \rho^2 \overset{(74)}{\leq} (\rho\sqrt{1-\varepsilon})^2 + 1 - \rho^2 = 1 - \varepsilon\rho^2$$

$\square$

**Lemma A.41.** *For $\varepsilon \in (0, 1)$ we have*

$$\left[1 + \frac{\varepsilon^2}{2(1+\varepsilon\sqrt{2})}\right]^{-\frac{p}{2}} \leq e^{-\frac{p}{4} \cdot \frac{\varepsilon^2}{(1+\varepsilon)^2}}$$

*Proof.* Using Lemma A.38(b) we have

$$\log\left(\left[1 + \frac{\varepsilon^2}{2(1+\varepsilon\sqrt{2})}\right]^{-\frac{p}{2}}\right) = -\frac{p}{2} \cdot \log\left[1 + \frac{\varepsilon^2}{2(1+\varepsilon\sqrt{2})}\right]$$

$$\leq \frac{p}{2}\left[-\frac{\varepsilon^2}{2(1+\varepsilon\sqrt{2})} + \frac{1}{2}\left(\frac{\varepsilon^2}{2(1+\varepsilon\sqrt{2})}\right)^2\right] \tag{75}$$

Looking at the difference:

$$\frac{\varepsilon^2}{2(1+\varepsilon\sqrt{2})} - \frac{\varepsilon^2}{2(1+\varepsilon)^2} = \frac{\varepsilon^2[(1+\varepsilon)^2 - (1+\varepsilon\sqrt{2})]}{2(1+\varepsilon)^2(1+\varepsilon\sqrt{2})} = \frac{\varepsilon^2[\varepsilon(2-\sqrt{2}) + \varepsilon^2]}{2(1+\varepsilon)^2(1+\varepsilon\sqrt{2})}$$

$$\geq \frac{\varepsilon^2[\varepsilon(2-\sqrt{2}) + \varepsilon^2]}{2(1+\varepsilon)^2(1+\varepsilon\sqrt{2})} \geq \frac{\varepsilon^3(2-\sqrt{2})[1 + \frac{\varepsilon}{2-\sqrt{2}}]}{2(1+\varepsilon)^2(1+\varepsilon\sqrt{2})}$$

$$\geq \frac{\varepsilon^3(2-\sqrt{2})[1 + \varepsilon\sqrt{2}]}{2(1+\varepsilon)^2(1+\varepsilon\sqrt{2})} \geq \frac{\varepsilon^3 \cdot \frac{1}{2}}{2(1+\varepsilon)^2} = \frac{\varepsilon^3}{4(1+\varepsilon)^2}$$

$$\geq \frac{1}{2}\left(\frac{\varepsilon^2}{2(1+\varepsilon\sqrt{2})}\right)^2$$

Hence, we have $(75) \leq -\frac{p}{4} \cdot \frac{\varepsilon^2}{(1+\varepsilon)^2}.$

$\square$

### A.V.3 SPECIAL CASE OF MEAN VALUE THEOREM

**Lemma A.42.** *Let $\psi(h) = \frac{h}{\sqrt{1+h^2}}$, then $\psi'(h) = \frac{1}{(1+h^2)^{3/2}}$. Furthermore, we have the following inequalities:*

*(a) For $0 \leq a < b$ we have*

$$\psi(b) - \psi(a) \leq \psi'(a)(b-a).$$

*(b) For $a < 0 < b$ we have*

$$\psi(b) - \psi(a) \leq b - a.$$

*Proof.* Using product rule for derivatives

$$\psi'(h) = \frac{(h)'\sqrt{1+h^2} - h(\sqrt{1+h^2})'}{1+h^2} = \frac{\sqrt{1+h^2} - h \cdot \frac{h}{\sqrt{1+h^2}}}{1+h^2}$$

$$= \frac{1+h^2-h^2}{(1+h^2)^{3/2}} = \frac{1}{(1+h^2)^{3/2}}.$$

For $a < b$ we have

$$\psi(b) - \psi(a) = \int_a^b \psi'(h)\, dh \leq \max_{s \in [a,b]} \psi'(s) \int_a^b du = \max_{s \in [a,b]} \psi'(s)(b - a).$$

It is not hard to see

$$\max_{h \in [a,b]} \psi'(h) = \begin{cases} \psi'(a) & 0 \leq a < b \\ \psi'(0) = 1 & a < 0 < b \end{cases}$$

$\square$

## B   RESULTS FOR RP DOT PRODUCT WHEN $P = A$

### B.I   RP DOT PRODUCT WHEN $P = A$

This section contains analogous results to §2.1 of the main part of the paper. We will show that inner-product similarity for $P = A$ will produce especially poor approximations for vertices $u \in H_c^\gamma$ and $v \in L_c$.

**Theorem B.1.** *Let $X = AR^\top$. Then, the following claims hold:*
*(a) Asymptotic result. For $u, v \in V$*

$$X_{u*} X_{v*}^\top \overset{a}{\sim} \mathcal{N}\left(n_{uv}, \frac{n_{uu} n_{vv} + n_{uv}^2}{q}\right), \tag{76}$$

*(b) Finite-sample result. For $\varepsilon \in (0, 1)$ and $\delta \in (0, 1)$ if $q \geq 4\frac{1+\varepsilon}{\varepsilon^2} \log\left[\frac{n(n-1)}{\delta}\right]$ then*

$$|X_{u*} X_{v*}^\top - n_{uv}| < \varepsilon \sqrt{n_{uu} n_{vv}} \tag{77}$$

*holds with probability at least $1 - \delta$.*

*Proof.* From part (a) of Theorem A.6, we can represent $X_{u*} X_{v*}^\top$ as sum of $q$ i.i.d random variables and a convenient term. From this, we can show asymptotic normality and get concentration results that lead to a JL Lemma-type result. These are described in the Proposition A.1.

Since $X_{u*} = A_{u*} R^\top$ and $X_{v*} = A_{v*} R^\top$, claim (a) follows from (15) and parts (a) and (b) of Lemma 2.2. Since $n = k = \|V\|$, claim (b) follows from part (b) of Proposition A.1. $\square$

Let us fix $v \in L_c$ and look at the set $\{X_{u*} X_{v*}^\top : u \in H_c\}$. We will show that these calculations can severely overvalue (or undervalue) the relevance values $\{A_{u*} A_{v*}^\top : u \in H_c\}$ they approximate. This will be a consequence of the two following lemmas.

**Lemma B.2.** *If $v \in L_c$ and $u \in H_c^\gamma$, then the standard deviation in (76) is greater than $\gamma d_v$, i.e.:*

$$\gamma d_v \leq \sqrt{\frac{n_{uu} n_{vv} + n_{uv}^2}{q}} \tag{78}$$

*Proof.* We have $\frac{n_{uu} n_{vv} + n_{uv}^2}{q} \overset{(2)}{\geq} \frac{d_u d_v + n_{uv}^2}{q} \overset{u \in H_c}{\geq} \frac{\gamma^2 c q d_v + n_{uv}^2}{q} \geq \gamma^2 c d_v \overset{v \in L_c}{\geq} (\gamma d_v)^2.$ $\square$

**Lemma B.3.** *If $v \in L_c$ and $u \in H_c$, the standard deviation in (76) is greater than its expectation, i.e.:*

$$n_{uv} \leq \sqrt{\frac{n_{uu} n_{vv} + n_{uv}^2}{q}}. \tag{79}$$

*Proof.* We have $\sqrt{\frac{n_{uu} n_{vv} + n_{uv}^2}{q}} \overset{(78)}{\geq} \gamma d_v \overset{(4)}{\geq} n_{uv}.$ $\square$

Under the conditions of Lemma B.3, we can produce the approximate one-sigma confidence interval

$$\left[ n_{uv} - \sqrt{\frac{n_{uu}n_{vv} + n_{uv}^2}{q}}, n_{uv} + \sqrt{\frac{n_{uu}n_{vv} + n_{uv}^2}{q}} \right]$$

in which we can expect $X_{u*}X_{v*}^\top$ to take values with less than 69% probability, based on (76). This means that getting a value outside that interval is not unlikely. In that case, from (79), we know such values will more than double $n_{uv}$ or be less than 0, a poor approximation. On the other hand, if we look at the values $(X_{w*}X_{v*}^\top : w \in L_c)$, we can see that the variance in (76) is bounded as

$$\frac{n_{ww}n_{vv} + n_{wv}^2}{q} \overset{(2)}{\leq} \frac{d_w^2 d_v^2 + d_w^2 d_v^2}{q} \overset{w,v \in L_c}{\leq} c^4 \frac{2}{q}.$$

Hence, in this case we can produce the three-sigma interval $\left[ n_{wv} - 3c^2\sqrt{\frac{2}{q}}, n_{wv} + 3c^2\sqrt{\frac{2}{q}} \right]$, where $X_{w*}X_{v*}^\top$ will take values with more than 99% probability, which is a small deviation.

Further, setting the random projection dimension $q$ according to (A.1), i.e., $q = \left\lceil \frac{1+\varepsilon}{\varepsilon} \log \left[ \frac{n(n-1)}{\delta} \right] \right\rceil$, we fulfill the finite-sample result in part (b) of Theorem B.1. However, for large values of $\sqrt{n_{uu}n_{vv}}$ (c.f., the denominator of 77), we note that the bounds in (77) become looser, thus larger approximation errors ensue. This will happen for vertices $u \in H_c^\gamma$, since $n_{uu} \geq d_u$. As a simple numerical illustration, assume $A \in \{0,1\}^{n \times n}$ such that $n_{uu} = d_u$ and $n_{vv} = d_v$. Let $\varepsilon = 10^{-2}$, $d_u = 10^7$, $d_v = 10$ and $n_{uv} = 1$. Then, (77) yields $X_{u*}X_{v*}^\top \in (1 - 100, 1 + 100)$.

The following is a technical lemma that we refer to in §2.1.

**Lemma B.4.** *(a) For $u, v \in V$ and $v$ satisfying* (4) *we have*

$$\frac{n_{uv}}{d_u d_v} \leq \frac{\gamma}{d_u}; \qquad (80) \qquad\qquad \frac{\sqrt{n_{vv}}}{d_v} \frac{1}{\sqrt{d_u}} \leq \frac{\sqrt{n_{uu}n_{vv}}}{d_u d_v} \leq \sqrt{\frac{\gamma}{d_v}} \frac{1}{\sqrt{d_u}}. \qquad (81)$$

*(b) For $u \in H_c^\gamma$ and $v \in L_c$ we have* $\frac{\gamma}{d_u} \leq \sqrt{\frac{1}{q}\left[ \frac{n_{uu}n_{vv}}{d_u^2 d_v^2} + \left( \frac{n_{u,v}}{d_u d_v} \right)^2 \right]}$.

*Proof.* For (80), we multiply the inequality (4) with $(d_u)^{-1}$. We have $d_u \overset{(2)}{\leq} n_{uu} \overset{(4)}{\leq} \gamma d_u$ and multiplying the last inequality by $\frac{n_{vv}}{d_u^2 d_v^2}$ and taking square roots (81) follows.

Part (b) follows from (78) by multiplying this inequality with $(d_u d_v)^{-1}$. $\qquad\square$

## B.II Instability of Ranking for Dot Product when $P = A$

This section is analogous to §3.1. Recall, $\mathrm{rel}_{uv} = A_{u*}A_{v*}$ and $\mathrm{rel}_{uv}^R = A_{u*}R^\top R A_{v*}^\top$.

The following lemma will help us in our considerations.

**Lemma B.5.** *If $x$ and $y$ are orthogonal vectors in $\mathbb{R}^n$, we have*

$$\frac{\cos(x, x-y)}{\sqrt{1 - \cos^2(x, x-y)}} = \frac{\|x\|}{\|y\|}. \qquad (82)$$

*Proof.* Using $(x, y) = 0$, we have

$$\cos(x, x-y) = \frac{(x, x-y)}{\|x\|\|x-y\|} = \frac{\|x\|^2}{\|x\|\sqrt{\|x\|^2 + \|y\|^2}} = \frac{\|x\|}{\sqrt{\|x\|^2 + \|y\|^2}}.$$

Now, it is not difficult to show $\sqrt{1 - \cos^2(x, x-y)} = \frac{\|y\|}{\sqrt{\|x\|^2 + \|y\|^2}}$. Hence, (82) follows. $\qquad\square$

Let $w = v \in L_c$. The relevance value will have a bound.

**Proposition B.6.** *For all $u \in V$ we have* $\mathrm{rel}_{vu} = n_{vu} \leq \gamma d_v$.

*Proof.* The result follows from definition of $n_{uv}$ and (4). $\qquad\square$

For a node of low degree, the relevance has to be low. However, this might not be the case for their approximation.

**Corollary B.7.** *For $v \in L_c$ and $u \in H_c^\gamma$, we have $\mathrm{rel}_{vv}^R \overset{a}{\sim} \mathcal{N}\left(n_{vv}, \frac{2n_{vv}^2}{q}\right)$ and $\mathrm{rel}_{vu}^R$ will have a nonnegative expectation with the standard deviation greater than $\gamma d_v$.*

*Proof.* Follows from Theorem B.1 and Lemma B.2 $\qquad\square$

The last result tells us that $\mathrm{rel}_{vv}^R$ will have values in $\left[n_{vv}\left(1 - 3\sqrt{\frac{2}{q}}\right), n_{vv}\left(1 + 3\sqrt{\frac{2}{q}}\right)\right]$ with more than 99%. This means, for example if $q \geq 100$, $\mathrm{rel}_{vu}^R \in [1.5\gamma d_v, \infty) \subset \left[n_{vv}\left(1 + 3\sqrt{\frac{2}{q}}\right), \infty\right)$ can happen with probability $\mathbb{P}(\mathcal{N}(0,1) > 1.5) \approx 6.7\%$. For higher $q$, this probability will be higher. Roughly, we can anticipate that $\mathrm{rel}_{vv}^R < \mathrm{rel}_{vu}^R$ when $\mathrm{rel}_{vv} > \mathrm{rel}_{vu}$ happens more frequently.

The following result will show that this can happen more often than we want.

**Corollary B.8.** *Let $v \in L_c$ and $u \in H_c^\gamma$ be two vertices with no common neighbors, i.e., such that $\mathrm{rel}_{vv} > 0$ and $\mathrm{rel}_{vu} = 0$. Then, $\mathbb{P}(\mathrm{rel}_{vv}^R < \mathrm{rel}_{vu}^R) > \mathbb{P}\left(\mathcal{T}_q > \gamma^{-1/2}\right) \geq \mathbb{P}(\mathcal{T}_q > 1) \approx 15.8\%$.*

*Proof.* Since $P_{v*}P_{u*}^\top = 0$, we have $\frac{\cos(P_{w*}, P_{u*} - P_{v*})}{\sqrt{1 - \cos^2(P_{w*}, P_{u*} - P_{v*})}} = \frac{\|P_{v*}\|}{\|P_{u*}\|}$ by (82). Hence, Theorem 3.1 gives us

$$\mathbb{P}(\mathrm{rel}_{vv}^R < \mathrm{rel}_{vu}^R) = \mathbb{P}\left(\mathcal{T}_q > \frac{\|P_{v*}\|}{\|P_{u*}\|}\sqrt{q}\right). \tag{83}$$

Further, $\frac{\|P_{v*}\|}{\|P_{u*}\|}\sqrt{q} = \sqrt{\frac{n_{vv}q}{n_{uu}}} \overset{(4)}{\leq} \sqrt{\frac{\gamma d_v q}{n_{uu}}} \overset{v \in L_c}{\leq} \sqrt{\frac{\gamma c q}{n_{uu}}} \overset{(2)}{\leq} \sqrt{\frac{\gamma c q}{d_u}} \overset{u \in H_c^\gamma}{\leq} \frac{1}{\sqrt{\gamma}}$. From the last inequality, we have $\mathbb{P}\left(\mathcal{T}_q > \frac{\|P_{v*}\|}{\|P_{u*}\|}\sqrt{q}\right) \geq \mathbb{P}\left(\mathcal{T}_q > \gamma^{-1/2}\right)$ and the claim follows from (83). Finally, from (4) and (2), we note that $\gamma \geq 1$. $\qquad\square$

Similar results would follow if we took two vertices $w, v$ of low degree with many common neighbors and a high degree vertex $u$ that has no common neighbors with $w$ and $v$. For simpler calculations, we took $w = v$.

