# OpenReview forum: "Node Similarities under Random Projections: Limits and Pathological Cases"
_ICLR.cc/2025/Conference — ICLR 2025 Poster_

### Official Review · Reviewer_YfdX · 2024-10-30

**Soundness:** 2
**Presentation:** 1
**Contribution:** 2
**Rating:** 3
**Confidence:** 5

**Summary:**

In this paper, the authors investigate low dimensional embeddings of graphs via random projections, and in particular investigate dot product/cosine similarity preservation under such a setting, and apply it to a ranking application. The main contributions of this paper are theoretically motivated, as the authors attempt to show that under a natural definition of relevance, the cosine similarity will have better approximation properties under random projections than the dot product, and better stability in the case of ranking.

**Strengths:**

originality: the paper produces some original results theoretically on the analysis of dot product vs cosine similarity in the random projection setting for graphs. This line of investigation is original, to the best of my knowledge.
quality:  I discuss the quality of this paper in the next section. Overall, I believe the mathematical derivations concluded in this paper are sound, but the quality of the exposition, motivation, and logical flow has much room for improvement.
clarity: Overall, clarity is not a strength of this paper. However, The first two pages introducing the paper and the contributions are written clearly.
significance: The broader topic of analyzing graph embeddings is certainly a significant direction of investigation. The application of ranking is also a significant avenue.

**Weaknesses:**

I raise questions and remarks along several front：

1. Motivation: The the elephant in the room when considering the paper's line of investigation in terms of motivation and significance is this: is if higher order connectivity of graphs is a main motivation, what is the advantage of considering random projections versus spectral embedding approaches (It is clear that the eigendecomposition based approaches capture connectivity and matrix powers very well from spectral graph theory). I understand that this is a theory paper and I do not expect experiments comparing the two, but some sort of high level explanation/discussion seems warranted to convince the reader that it is worth the effort to develop theory for random projections.

There are many details that the authors do not provide sufficient motivation for. For example, Also, on the top of page 4, q = O(log n) is set to explain large graphs' johnson lindenstrauss type results, which is natural given the well known log scaling of random projection techniques. However, in the same page in theorem 2.2a, the asymptotic results are considered under a root q scaling. I understand that this is what makes the normal approximation machinery goes through, but there seems to be a disconnect between the motivation of this research and some of the asymptotic results provided.

Minor point: -  the introduction of the NDCG metric was without citation/reference/background motivation in page 6. I suggest adding motivation.

2. The mathematical results are not presented in a very organized/clear way.

In the statement of the theorems, e.g. theorem 2.2, there is little indication of what setting the theorem corresponds to. In theorem 2.2a, the term "large q" is used in this asymptotic result, presumably indicating q is being sent to infinity, but the term q also appears on the variance term on the right hand side...one has to look at the remark below to see that this is explained by the somewhat nonstandard notation \sim_a to indicate a root q scaling.  Results like these could be stated a lot more clearly. I highly suggest the authors expand the statements in the theorem to state exactly the definitions/constructions used in the setting and the regularity/structural assumptions that they are referring to for clarity.

I say this NOT to indicate that the mathematical derivation is wrong (even though I did not check proofs in appendix line by line, the mathematical results here overall look ok to me), but to indicate that the lack of clarity in the write up and the statements of the theorems make it very difficult for readers to properly check the math and keep track of the exact setting/assumptions that the particular theorems are referring to. Connecting the theoretical results, to the recommendations in the discussion section, becomes even harder given the somewhat disorganized nature of the presentation.

3. The organization of the paper can be improved:
- the subsection titles are not well organized. Section 2.1 is RP DOT PRODUCT WHEN P = T, and seeing this one naturally expects 2.2 to be P = A, but turns out that is delegated to the appendix, and 2.2 goes to cosine similarity where P = A and T are both considered since they don't make a difference. I suggest making 2.1 heading just RP dot product, and then consider the subcases P = T etc within the section rather than on the title. Ditto for the stability subsection titles in section 3.

Given the above, my belief is that the paper can be substantially improved in the presentation, clarity and organization front.

**Questions:**

See comments above.

---

> ### Author Response · Authors · 2024-11-20
> **Response to Reviewer YfdX**
>
> We appreciate the reviewer recognizing the strengths of the paper in terms of its originality and significance, as well as the soundness of the mathematical derivations. Next, we address each of the comments raised by the reviewer.
>
> 1.1 - *Motivation of Random Projections in comparison with Spectral Methods*. The main advantages of random projections lie in their favorable algorithmic requirements (a single matrix multiplication over a sparse adjacency matrix) and their flexibility.
>
> Whereas computation of spectral embeddings requires the decomposition of the adjacency matrix of the whole graph for meaningful eigen-/singular vectors, computation of random projection embeddings can be performed independently *and only for an arbitrary subset of nodes of interest*. This is decisive for the adoption of random projections many practical applications (e.g., large/billion-scale graphs, graphs with attributes, dynamic graphs) which warrants the need for characterization of their approximation quality.
>
> Random projections also present favorable computational complexity: for a graph with $n$ nodes and $m$ edges, computing $q$-dimensional RP embeddings for the whole graph requires a single-pass sparse matrix multiplication, with computational complexity $\mathcal{O}(mq)$. Here, $m  = \bar d n \ll n^2$ for most graphs, where $\bar d := \frac{m}{n}$ is the graph's average degree. Spectral embeddings typically require power iterations with multiple matrix multiplication passes for convergence, with the number of iterations and precision being both graph and algorithm dependent.
>
> We understand this context was missing in paper. Following the suggestion of the reviewer, we will incorporate this discussion to the manuscript to further motivate our contributions to the theory for random projections.
>
> 1.2 - *Scaling of $q$ and $n$*. We show that the dot product is a sum of $q$ independent terms, and that for large values of $q$, the distribution will be approximately normal. In the case of the JL Lemma, for large $n$ (for example, $n>10^6$), from Theorem 2.2.b, we have $q \geq \frac{1}{\varepsilon^2}$. This is large enough for the dot product to be in the domain of attraction of the Central Limit Theorem and gives valuable insight into where we can expect the dot product under random projections values to be. To further connect the asymptotic and finite-sample results, an analog role of $\frac{1}{q}$ in the variance term for normality is played by $\varepsilon$ in the JL lemma, but for all dot products. Following the suggestion of the reviewer, we will add a remark to clarify this in the text.
>
> 2.1 - *Notation for asymptotic results*. We agree that the notation for asymptotic convergence in distribution can be presented earlier in the paper, before it is used in Theorem 2.2. We thank the reviewer for this observation and will update the paper accordingly.
>
> 2.2 - *Implicit definitions and assumptions*. We will revise the theorem statements for any implicit definitions or assumptions, as recommended by the reviewer.
>
> 2.3 - *Connecting recommendations to theoretical results*. To address the point raised by the reviewer, we will add in the conclusion explicit references to the theorems and results leading to the recommendations.
>
> 3 - *Organization of the paper*.  We appreciate the suggestions for renaming of the subsections given by the reviewer and will update the paper accordingly.

---

### Official Review · Reviewer_qkog · 2024-11-01

**Soundness:** 3
**Presentation:** 3
**Contribution:** 3
**Rating:** 6
**Confidence:** 2

**Summary:**

The paper examines the problem of embedding an adjacency/transition matrix using random projection through the lens of preserving dot product and cosine similarity. The paper provide both asymptotic and finite sample results that compares the two, and furthermore investigates this problem in a ranking configuration.

**Strengths:**

1. The paper provides an interesting analysis supported by theorems alongside their interpretations that make the results more clear.
2. The ranking application is interesting, and is supported by proofs as well.

**Weaknesses:**

See questions.

**Questions:**

A. As the authors wrote, random projection has guarantees with respect to the Euclidean distance preservation (Based on the JL lemma). I wonder how well would the regular random projection perform (in terms of dot product preservation) if the rows of the adjacency/translation matrix would have been normalized (using the L2 norm) before the projection. If space permits, I would suggest the authors to add a brief discussion or even analysis of this case. This could provide valuable context on the robustness of the findings
and a more comprehensive comparison of approaches.


B. The assumption that the adjacency matrix is composed of only integers numbers should appear at the beginning of section 2. Currently it first appears in the proof of Lemma 2.1 in line 138. I recommend to add a clear statement about the integer assumption for the adjacency matrix in the opening paragraph or definitions of Section 2. Also, I would suggest to note or even briefly discuss the implications of this assumption, given that adjacency matrices in some applications may be continuous.


C. The definition of  rank_w^R(h) that was used in line 315 is missing. I suggest to add a brief definition of it where it is first introduced, perhaps with a simple example to illustrate its meaning in the context of the ranking application.


Insights: The adjacency matrix configuration that you analyzed can be applied to problems in Natural Language Processing.

---

> ### Author Response · Authors · 2024-11-25
> **Response to Reviewer qkog**
>
> We thank the reviewer for providing valuable feedback and appreciating the strengths of our paper.
>
> Next, we address each of the questions raised:
>
> A. *L2 normalization*. Although we did not include this case in the main paper, we would like to refer the reviewer to the asymptotic and finite-sample results in Theorem A.10 and Theorem A.17 in the Appendix, respectively, which did consider the specific case of L2 normalization *before* random projection.
>
> The reasons why we did not promote these results to the main paper are two-fold:
>
> 1. The asymptotic variance $\frac{1 + \rho^2}{q} $ obtained for this case (c.f. Theorem A.10) is higher than the variance $\frac{(1 - \rho^2)^2}{q}$ obtained for cosine similarity (c.f. Theorem A.21).
>
> 2. Obtaining the rows of a matrix polynomial $p(A)$ needed to compute the denominators for normalization involves computing $p(A)$ before random projection. Such a computation is often impractical, as it can typically lead to the powers of $A$ resulting in dense $n \times n $ matrices. On the other hand, computing the polynomial $p(A)R^T$ after random projection (using the association $A(\ldots(A(AR^T))$) takes advantage of the intermediate factors, which are lower-dimensional projected matrices $n \times q$, with $q \ll n$.
>
> We agree with the reviewer that the case of L2 normalization is a natural direction of consideration as an intermediate case to L1 normalization before RP, when $P=T$, and L2 normalization after RP, as done for cosine similarity. We can capture the suggestion of the reviewer as a remark for a more comprehensive comparison of the results in the paper.
>
> B. *Entries of the adjacency matrix* We will state more clearly that the matrix can be assumed as $A\in (\mathbb{Z}_0^+)^{n\times n}$. Proofs in the paper would also work for $A\in ([1,\infty]\cup \{0\})^{n\times n}$ unchanged. Results for the general case $A\in (\mathbb{R}_0^+)^{n\times n}$ would also hold by incorporating a scaling constant $c > 0$ such that $cA\in ([1,\infty]\cup \{0\})^{n\times n}$, but we preferred not do it to avoid unnecessary clutter. We can comment on this in the text of the paper.
>
> C. We will provide an explicit definition of $\textrm{rank}_w^R(h)$ with a simple example in the context of the ranking application, as suggested by the reviewer.
>
> Lastly, we are keen on learning more about the application of the adjacency matrices to natural language processing, as it can further extend the scope of our results. We would appreciate any specific references that the reviewer may be willing to provide us.

---

### Official Review · Reviewer_LuRP · 2024-11-02

**Soundness:** 3
**Presentation:** 3
**Contribution:** 3
**Rating:** 6
**Confidence:** 4

**Summary:**

In this paper, the authors investigate random projections (RP) for graph matrices, specifically adjacency and transition matrices, to produce 'low-dimensional' node embeddings. They derive both asymptotic and finite-sample results in the spirit of the Johnson-Lindenstrauss (JL) lemma, assessing the distortion of two node similarity measures—dot products and cosine similarities—under Gaussian random projections. The authors demonstrate that RP can significantly distort dot product similarity, with the extent depending on the degrees of the nodes involved. In contrast, cosine similarity proves more robust, exhibiting less distortion under RP. They apply these findings to a ranking problem, theoretically establishing that normalized discounted cumulative gain (NDCG) is more stable when cosine similarity is used as the similarity measure compared to dot product similarity. Finally, the authors provide numerical evidence supporting their theoretical claims using a real-world dataset.

**Strengths:**

The main strengths, in my opinion, are:

1. **Relevance**: Node embedding is highly relevant within graph machine learning and representation learning, and RP offer a promising approach to obtaining these embeddings. The study focuses on well-established similarity measures widely used in practice, and the application to ranking is equally pertinent.

2. **Rigorous Mathematical Treatment**: The paper is technically sound, with proofs that appear correct and rigorous.

3. **Clear Presentation**: The paper is well-written and easy to follow, aside from a few issues outlined below.

**Weaknesses:**

1- **Claims of Novelty and Improvements upon Known Results**: To clarify, I do consider the paper to make a valid contribution with its theoretical analysis of random projections (RP) for node embeddings and the resulting distortions in well-known similarity measures. However, it is sometimes challenging to determine which results the authors claim as novel. It would be helpful if the authors clearly delineate what they consider to be novel contributions versus extensions or refinements of existing work. For instance, in the introduction, Section 2, and the appendix, the authors appear to suggest that their rotation argument—which follows directly from well-known properties of Gaussian distributions—constitutes a novel contribution. They state that "we use the rotation argument to provide a new representation for the dot product and cosine similarity under random projections." However, if I understand correctly, the dot product representation could be derived using the distribution of the off-diagonal entries of a Wishart matrix, with a correlation coefficient corresponding to their parameter $\rho$.

The inclusion of detailed proofs in the appendix is appreciated for self-containment, but much of the content presented there either derives from well-established results or can be directly obtained through known concentration or asymptotic arguments. If my interpretation is accurate, the paper’s genuine technical novelty lies in the more refined bounding of relevant quantities following classical concentration or asymptotic approaches. This does sometimes lead to improved results, but in certain sections, such as Section A.III.5, the enhancements seem only marginal.

2- **Difficulty to extend to other sparse RP settings**: The arguments presented here, based on rotation symmetries of the multivariate Gaussian distributions, seem difficult to adapt to the more practically relevant case of spare RP.  In my view, it would be beneficial to discuss the potential challenges and limitations of extending their approach to sparse random projections, or discussing avenues for future work in the conclusion.

**Questions:**

1. The notation $ X_{u*}$ for the rows of a matrix is not explained prior to its use on line 115. It would help to define this notation when it first appears.

2.  On line 108, the authors suggest that computing $p(A)R^\top $ is faster than $p(A)$ alone. Could the authors clarify why this is the case?

3. It may improve clarity to introduce the power law either before or immediately following line 155.

4.  In line 281, the authors state that the interval endpoints are independent of the node degrees. Is this statement exact, or is it an approximation? Additional clarification would be beneficial here.

5. Proposition 2.6 seems straightforward. Is it necessary to present it formally as a proposition?

6.  For the ranking application, a more detailed explanation of the overarching goal would be helpful. Also, citations for the Normalized Discounted Cumulative Gain (NDCG) metric would add valuable context.

7. The definition of  $\text{rank}^R_w(h)$ is missing, and notation in this section could be streamlined to be consistent with the rest of the paper. For example, $NDCG(r^K_i, \hat{r}^K_i)$ is not defined.

8. There are some citation formatting issues in the appendix, such as on line 1079, where several references are improperly displayed.

9. It would be helpful to more clearly indicate which results in the appendix are novel contributions.

10. You establish a comparision with the results about cosine similarity of (Arpit et al. 2014) in section A.IV.5, but this not mentioned in the related work section. Indeed, the related work section, it could be mentioned which of the cited articles, and how, can be compared with the results here. For instance, you compare in Section A.III.5 the concentration results for $\frac{\langle Rx,Ry\rangle}{\|x\|\|y\|}$ with the  resullts in (Vempala, 2004) and (Kaban, 2015).

11-Could you please address the point I raised in the weaknesses section?

---

> ### Author Response · Authors · 2024-11-23
> **Response to Reviewer LuRP (Weaknesses)**
>
> We would like to thank the reviewer for the thorough consideration and valuable feedback given to our paper.
>
> Response to weaknesses:
>
> *Novelty and refinements*. Thank you for referring us to the results on the off-diagonal entries of the Wishart matrix; we had not made that connection before and are grateful to learn about it. We will adapt the text to make sure it is pointed out where the result is used in the Appendix. Whereas prior literature has focused on analyzing the dot product through the difference $RP_{u*}^TP_{v*}R^T=\frac{1}{4}(\|P_{u*}R^T+P_{v*}R^T\|-\|P_{u*}R^T-P_{v*}R^T\|)$, this angular representation allows us to reveal fundamental differences between dot product and cosine similarity in quality of the approximation under RP. Our novel results include the specific analysis considering the graph structure; the exact formula for sign change given in Proposition A.8, from which Theorem 3.1 was developed; and the exact estimation of flipping probability in Corollary 3.4. The remaining results were improvements or refinements of existing results, which also followed from the angular representation of the marginal distributions. We will make sure to clarify this in the paper.
>
> *Other sparse RP settings*. We agree with the reviewer that the potential extension to sparse RP setting an interesting point for discussion. From our early explorations after writing the current paper, we can point to the following. Due to similar properties between Rademacher and normal random variables, we expect similar qualitative behavior. More specifically, in the sparse case the entries of $R$ are often taken to satisfy $$Pr (R_{ij}=\\frac{k\\sqrt{s}}{\\sqrt{q}})=\\left\\{\\begin{array}{cll}
> \\frac{1}{2s} & \\textrm{for} & k =-1\\\\
> (1- \frac{1}{s}) & \\textrm{for} & k = 0\\\\
> \\frac{1}{2s} & \\textrm{for} & k = 1
> \\end{array}\\right.$$
> for $s>0$. In this case, the dot product will have the following properties: $$\\mathbb{E}[RP_{u*}^TP_{v*}R^T]=P_{u*}^tP_{v*}$$ and
> $$\\mathrm{Var}[RP_{u*}^TP_{v*}R^T=\\frac{1}{q}\\left[\|P_{u*}\\|^2\\|P_{v*}\\|^2+ (P_{u*}^TP_{v*})^2+(s-3)\\sum_{w\\in V}^nP_{uw}^2P_{vw}^2\\right].$$
>
> Using a Central Limit Theorem argument similar to parts (a) of Theorem 2.2 and Theorem A.6, we will have
> $$RP_{u*}^TP_{v*}R^T\stackrel{a}{\sim} \mathcal{N}\left(P_{u*}^TP_{v*}, \mathrm{Var}[RP_{u*}^tP_{v*}R^T]\right).$$
> Note that $(s-3)$ in the variance expression can be negative, however since all entries of matrix $P$ are non-negative, we have $$0\leq \sum_{w\in V}P_{uw}^2P_{vw}^2\leq (P_{u*}^tP_{v*})^2.$$
>
> If the product of the norms is high compared to the dot product, the challenge of controlling the variance will remain a problem. This will lead to similar issues with respect to the structure of the graph. For $s\geq 3$ and $P=T$, if $u$ is a high-degree node and $v$ a low-degree node, the standard deviation will be greater than the expectation (using similar arguments as in Corollary 2.3). The discussion about confidence intervals would also hold. For $s<3$, some adaption of the argument would be necessary. This points that the we expect the similar qualitative behavior in the case of dot product.
>
> For the cosine similarity, using the $\delta$-method we should be able to get part (a) Theorem 2.5, potentially with some modifications in the variance.
>
> Getting analogs of non-asymptotic results as we did in the text of paper will require restating and modifying proofs. They were simpler to prove for Gaussian matrices, but we believe we can expect similar outcomes for the sparse case. We plan to develop these results in future work that may lead to a new and complementary manuscript.
>
> We will add a comment about this in the conclusion.

---

> > ### Author Response · Authors · 2024-11-23
> > **Response to Reviewer LuRP (Questions)**
> >
> > Responses to questions:
> >
> >  1. We clarify in the text that $X_{u*}$ represents the row that corresponds to the vertex $u$.
> >  2. Computing $p(A)R^T$ is generally faster, since a power $A^kR^T$ is calculated as $A(\ldots(A(AR^T))$. We have $A\in (\mathbb{Z}_0^+)^{n\times n}$ and $R\in \mathbb{R}^{q\times n}$, with $A$ sparse and $q \ll n$ by assumption. Computing $A^k$ without $R$ runs the risk of the resulting full-dimensional $n \times n$ matrix $A^k$ becoming dense, depending on the graph's connectivity. We can add this to the appendix.
> >  3. We will introduce the power law as suggested.
> >  4. We will revise the comment on line 281 to state that the interval end points only depend on the cosine similarity we want to estimate. We thank the reviewer for raising this issue.
> >  5. We used it to make a statement on the stability of the cosine similarity. We can drop the proof for conciseness.
> >  6. We will add an explanation for ranking and citations for NDCG as suggested.
> >  7. We will streamline the notation used in the Computational Experiments section as suggested.
> >  8. Thank you for pointing the formatting glitches on those citations in the appendix; they will be addressed.
> >  9. We will address the novelty and improvement of the results in the appendix.
> >  10. Thank you for this suggestion. We will promote the referred comparison in the Appendix to the "related work" section in the main paper, as suggested.
> >  11. We addressed point I of the weaknesses in our previous response.

---

> > > ### Comment · Reviewer_LuRP · 2024-11-26
> > > **Some comments on the authors response**
> > >
> > > I would like to sincerely thank the authors for their detailed response and apologize for my delayed reply. Regarding the claims of novelty, I am still unconvinced by the authors' arguments. For instance, they cite the sign-change property as an example of novelty, supported by a series of results (Proposition A8, Theorem 3.1, and Corollary 3.4). In my opinion, Proposition A8(a) is straightforward, and part (b) follows directly. Specifically, given that result (23) in Proposition A6 is known (as I highlighted in my previous comment), Proposition A8(b) appears to be a simple consequence. I would consider such properties to be "folklore" in the context of Wishart matrices. Therefore, I maintain my opinion regarding this weakness of the paper, particularly in relation to my earlier statement: "If my interpretation is accurate, the paper’s genuine technical novelty lies in the more refined bounding of relevant quantities following classical concentration or asymptotic approaches. This does sometimes lead to improved results, but in certain sections, such as Section A.III.5, the enhancements seem only marginal."
> > >
> > > I also appreciate the authors' effort to address the sparse RP setting. As they note, the asymptotic case could potentially be addressed, but the non-asymptotic case appears more challenging. One intuition supporting my main concern is that the rotation argument, which relies on the rotational invariance of Gaussian matrices, may no longer apply in this context. Do the authors have any insights regarding this limitation? While I do not expect a complete resolution, as this could indeed be a topic for future work (as the authors suggest), I believe this limitation, albeit perhaps minor, represents a weakness of the proposed approach and will likely require new arguments to overcome.
> > >
> > > Given these points, I maintain my previous assessment and score for the paper.

---

> ### Author Response · Authors · 2024-11-30
> **On the rotational invariance of the Gaussian distribution and concentration results for cosine similarity**
>
> We again thank the reviewer for their comments.
>
> One of the goals this paper was to clarify how *cosine similarity* behaves under random projections. Although there existed prior results in the literature -- for example as we mention in Remark A.30 (based on the results from (Arpit et al)) -- those concentration inequality results looked like
> $$|\cos(Rx,Ry)-\cos(x,y)| \leq \varepsilon (1+|\cos(x,y)|).$$
>
> This fails to capture the fundamental fact that the closer $|\cos(x,y)|$ is to 1, the more precise the estimate with random projections will be. This was indicated by the asymptotic result (Theorem A. 21), and confirmed by concentration result (Theorem A. 23):
> $$ |\cos(Rx,Ry)-\cos(x,y)| \leq \varepsilon (1-\cos^2(x,y)).$$
> This fact also enabled us to prove JL Lemma version for cosine similarity as stated in Theorems A. 28 as well as the version for graphs in part (b) of Theorem 2.5.
>
> The decomposition provided by the rotational invariance of the Gaussian distribution played a crucial role in obtaining this estimate, as it can be seen in Lemma A.27. The elements from the representation (24)
> $$\frac{(Rx,Ry)}{\|Rx\|\|Ry\|}= \frac{\rho\|N\| + M_1\sqrt{1-\rho^2}}{\sqrt{(\rho\|N\|+M_1\sqrt{1-\rho^2})^2 + (1-\rho^2)\|M_{2\ldots q}\|^2}}$$
> and their distributions were essential in proving this result.
>
> For cosine similarity -- in the sparse case, a similar concentration result may be difficult to obtain. However, we do believe that the version in the literature of the concentration result can still be obtained, as long as we can get the concentration results for the dot product or the norm -- both of which seem likely.

---

> > ### Author Response · Authors · 2024-11-30
> > **On transferable results and the sparse case specifics**
> >
> > Next, we will address the question on what can be transferred and what will be specific for the sparse case.
> >
> > Again we are assuming:
> > $$Pr (R_{ij}=\\frac{k\\sqrt{s}}{\\sqrt{q}})=\\left\\{\\begin{array}{cll}
> > \\frac{1}{2s} & \\textrm{for} & k =-1\\\\
> > (1- \frac{1}{s}) & \\textrm{for} & k = 0\\\\
> > \\frac{1}{2s} & \\textrm{for} & k = 1
> > \\end{array}\\right.$$
> >
> > As discussed already, using Central Limit Theorem and possibly $\delta$-method, we will have the similar asymptotic results as those in parts (a) of Theorem 2.2 and Theorem 2.5.
> >
> > **Corollary 3.4** for $P=T$ will hold in a *weaker form*. Namely, if $u$ is a high-degree node and $v$ low-degree node with no common neighbor (recall, in this case $P_{u*}P_{v*}^t=0$), we have
> > $$(rel^R_{uu}<rel^R_{uv}) = (P_{u*}R^tRP_{u*}^t<P_{u*}R^tRP_{v*}^t)$$
> > $$
> > =\left(\frac{P_{u*}R^tRP_{v*}^t}{\|P_{u*}R^t\|^2}>1\right) =\left(\frac{\hat{P}\_{u*}R^tR\hat{P}\_{v*}^t}{\|\hat{P}\_{u*}R^t\|^2}>\frac{\|P_{u*}\|}{\|P_{v*}\|}\right)
> > $$
> > $$=\left(\frac{\hat{P}\_{u*}R^tR\hat{P}\_{v*}^t/\sqrt{q}}{\|\hat{P}\_{u*}R^t\|^2/q}>\frac{\|P\_{u*}\|}{\|P\_{v*}\|}\sqrt{q}\right),$$
> > where $\hat{P}\_{h}=\frac{P_{h*}}{\|P_{h*}\|}$ for $h=u,v$.
> >
> > Using the same arguments as in the proof of Corollary 3.4 (see arguments after (12)), we have
> > $\frac{\|P_{u*}\|}{\|P_{v*}\|}\sqrt{q}\leq \gamma^{-1/2}$. Therefore,
> > $$\Pr(rel^R_{uu}<rel^R_{uv}) \geq \Pr\left(\frac{\hat{P}_{u*}R^tR\hat{P}_{v*}^t/\sqrt{q}}{\|\hat{P}_{u*}R^t\|^2/q}>\gamma^{-1/2}\right).$$
> >
> > Using Central Limit Theorem, we have $\hat{P}\_{u*}R^tR\hat{P}\_{v*}^t/\sqrt{q}\stackrel{d}{\to} N(0,1)$, from the Law of Large Numbers $\|\hat{P}\_{u*}R^t\|^2/q\to 1$ almost surely as $q\to \infty$.
> >
> > Therefore, $\frac{\hat{P}\_{u*}R^tR\hat{P}\_{v*}^t/\sqrt{q}}{\|\hat{P}\_{u*}R^t\|^2/q}\stackrel{d}{\to} N(0,1)$.
> >
> > Hence, since $q$ is large and $\gamma \geq 1$
> > $$\Pr(rel^R_{uu}<rel^R_{uv})\geq \Pr\left(\frac{\hat{P}_{u*}R^tR\hat{P}_{v*}^t/\sqrt{q}}{\|\hat{P}_{u*}R^t\|^2/q}>\gamma^{-1/2}\right) \approx \Pr(N(0,1) > \gamma^{-1/2}) \geq \Pr(N(0,1) > 1).$$
> > This is essentially the statement of Corollary 3.4 --- showing how often the random projection will flip the ranking order in this case.
> >
> > **Dot product** estimates for $P=T$. In the case of sparse matrices, we have a different situation:
> > - $T$ has nonnegative entries whose rows sum to 1;
> > - $R\in \\{0,1,-1\\}^{p\times n}$.
> >
> > Therefore, elements of the sum
> > $$S = P\_{u*}R^tRP\_{v*}^t-P_{u*}P_{v*}^t=\sum_{j=1}^q \left(T_{u*}R_{\*j}^tR_{\*j}T_{v*}^t-\frac{1}{q}T_{u*}T_{v*}^t\right)$$
> > are all in the in the interval $[-\frac{s+1}{q},\frac{s}{q}]$. Many concentration inequalities could be used now (Hoeffding, Bennett, ...). A first look at Bennett's inequality gives similar estimates as those used to prove the dot product results for $P=T$ in the Gaussian case (formula (2.10) in (Bucheron et al.)):
> > $$\Pr(S>\varepsilon)\leq \exp\left(\frac{-\varepsilon^2q}{2(\nu+\varepsilon s/3)}\right),$$
> > $$\Pr(S<-\varepsilon)\leq \exp\left(\frac{-\varepsilon^2q}{2(\nu+\varepsilon (s+1)/3)}\right),$$
> > where $\nu =q {\rm Var} [S]= \|T_u\|^2\|T_v\|^2+(T_u,T_v)^2+(s-3)\sum_{w\in V}^nP_{uw}^2P_{vw}^2$ is the variance.
> > This is similar to formula (35) in our paper. Using this or some more precise result will lead to JL type result for the dot product.
> >
> > As mentioned before, the dot product estimate can be used to estimate the **cosine similarity** by applying the inequality to each term in the fraction $\frac{P_{u*}R^tRP_{v*}}{\|P_{u*}R^t\|\|P_{u*}R^t\|}$. As done in the literature, this will likely lead to the inequality like
> > $$| \cos (P_{u*}R^t,P_{u*}R^t)-\cos (P_{u*},P_{u*})|\leq \varepsilon (1+|\cos (P_{u*},P_{u*})|),$$
> > since a similar approach was used for the Gaussian case in (Arpit et al.) and led to Lemma A. 29 and Remark A. 30 in the Appendix of the paper.
> >
> > In summary, we believe that all results from this paper will hold in some (possibly modified) form for the case of sparse random projections. Although the rotation argument which allowed us to prove tighter results for cosine similarity may be specific to the Gaussian case, the above insights tell us that the approaches, qualitative results, and essential differences between dot product and cosine similarity for graphs are transferrable to the sparse case. In this respect, the current paper makes the essential contributions of laying out the problem of the difference in behavior (and degree-related pathologies) of random projections over graphs according to different notions of similarity and extensively addressing the Gaussian case as its foundation. Extension to the sparse case is a natural next step, whose motivation is also justified as a contribution of the current paper. With the above elaborations, we hope to have provided the necessary elements to fully substantiate the contributions of our paper.

---

> > > ### Comment · Reviewer_LuRP · 2024-12-01
> > >
> > > I would like to thank the authors once again for their response and for sharing their insights regarding the sparse RP case. I encourage them to further explore this promising direction in their future work.

---

### Official Review · Reviewer_Nudr · 2024-11-02

**Soundness:** 3
**Presentation:** 2
**Contribution:** 2
**Rating:** 6
**Confidence:** 4

**Summary:**

This paper investigates how well dot product and cosine similarity are preserved by random projections when these are applied over the rows of the graph matrix, concluding that the variance of the dot product estimation is worse than cosine similarity estimation.

**Strengths:**

1. Each mathematical description is carefully written.
2. The specific theoretical results, Theorems 2.2 and 2.5 are novel, as far as I know. They successfully associate the variance in estimating relevance (dot product or cosine similarity) with the graph structure.

**Weaknesses:**

1. Johnson-Lindenstrauss Lemma is roughly stated in Line 118 without the reference for proof, but not formally introduced as a Theorem. As there are some variants in the statement, it is essential to specify which form to consider with specific reference with proof.
2. The further discussion associating JL Lemma and Theorem 2.2. would be preferable. What aspects of JL Lemma previous researchers have tended to miss and what aspects of the problem setting lead to a higher variance in estimation? Would it not be possible to give an intuitive explanation so readers can understand it without reading the proof of Theorem 2.2?
3. While each theorem in this paper is interesting, the direction of the whole discussion is not justified by those theorems.
Specifically, in Conclusion section, the Authors states that "we propose that practitioners of random projection methods first use the embeddings $u \\mapsto \\frac{X\_{u*}}{\\|X\_{u*}\\|}$." However, easiness in estimating an index (in this case, a similarity measure) does not directly mean we can recommend it. Rather, such a measure is problem-dependent. We cannot choose a similarity measure arbitrarily even if we know one is easy to estimate than another. For example, if we set $\\mathrm{rel}\_{uv} = 1$, which is a constant function, then we can trivially estimate it from any random projection (we can ignore the projection and simply always output 1). Nevertheless, this relevance measure would not be useful in real applications. If we propose to use cosine similarity, we need to discuss its meaning from a graph perspective, also.
4. Related to the above, the Authors should have discussed the difficulty of the estimation problem itself. The reason why a trivial algorithm can estimate the above trivial similarity measure $\\mathrm{rel}\_{uv} = 1$ is that the similarity measure estimation problem itself is easy as the variance of the true similarity $\\mathrm{rel}\_{uv}$ is zero (here, variance is defined when we sample $u$ and $v$ randomly), not that the trivial algorithm is excellent. Likewise, the Authors should have discussed the difference between the difficulties in the dot product similarity and cosine similarity. In this sense, Proposition 3.2 and 3.6 are partially helpful but not sufficient, as $\\mathrm{rel}\_{uv} = 10000$ is still an easy problem, while the upper bound of the value is large. One could evaluate the variance of $\\mathrm{rel}\_{uv}$ for both the dot-product case and cosine case, showing its dependency on the graph structure. If the problem difficulties are almost the same but the estimation performance through the random projection is much worse for dot-product than for cosine similarity, it is worth alerting. However, if the variance of $\\mathrm{rel}\_{uv}$ is higher for dot-product case, which means that the estimation problem itself is more difficult, then your comparison between Theorems 2.2 and 2.5 may not be practically meaningful.

----
**After the discussion period**:
The concern on Weakness 4 has not been solved as the specific distribution of cosine similarity and dot product has not been given. However, the concern on Weakness 3 has been solved by the Authors' agreeing to revise the conclusion. Even if the Weakness 4 remains, as long as the conclusion is not in the wrong direction, the current paper is meaningful as an analysis of random projection performance from a cosine similarity perspective and dot product similarity perspective. The analyses themselves are beneficial to the community. I have raised the score.

**Questions:**

See Weaknesses.

---

> ### Author Response · Authors · 2024-11-21
> **Response to Reviewer Nudr**
>
> We thank the reviewer for appreciating the strengths of our paper and for providing feedback for clarification, which we address next:
>
> 1. We stated the general JL Lemma in Theorem A.31 of the Appendix, but did not reference it in the main paper. Following the suggestion of the reviewer, we will place it in the main part of the paper and provide the requested reference with the associated proof.
>
> 2. We can expand the discussion on the JL and Theorem 2.2 as follows. Prior research has focused on applying the JL Lemma simultaneously to both $\|P_{u*}R^t+P_{v*}R^t\|$ and $\|P_{u*}R^t-P_{v*}R^t\|$ in the expansion $RP_{u*}^tP_{u*}R^t=\frac{1}{4}(\|P_{u*}R^t+P_{v*}R^t\|-\|P_{u*}R^t-P_{v*}R^t\|)$. This form (a difference of terms) limits the derivation of advanced results, since the associated terms are not independent.
> On the other hand, the representation (23) of the dot product, given by
> $$RP_{u*}^tP_{u*}R^t=\|x\|\|y\|(\rho \|N\|^2+\|N\|M_1\sqrt{1-\rho^2}),$$
> where $N$ is the $q$-dimensional standard normal vector and $M_1$ independent standard normal random variable, is easier to analyze. More specifically:
>
> - Since $\|N\|^2$ is sum of $q$ terms, we get asymptotic normality, which would be difficult with the usual approach.
>
> - We can get an exact formula for the sign change (Proposition A.8), which was useful to show that in ranking a reordering might happen with high probability, as stated in Corollary 3.4.
>
> - We also get more precise concentration results as given in  Theorem 2.2(b) and Theorem 2.5(b). A comparison is discussed in Sections A.III.5 and A.IV.5 of the Appendix.
>
> This approach is even more critical in analyzing the cosine similarity, in which case getting any concrete results would be considerably more difficult. Cosine similarity under random projections involves a ratio of functions of random matrices (which may explain the scarcity of theoretical results in this particular subject) and the transformation in (23) is a crucial step in their characterization.
>
> 3. We agree that a similarity measures can be problem-dependent, and many variations of them have been considered in the literature of specific applications. Dot products of rows of adjacency matrices have a clear interpretation as the overlap of neighbors. Transition probabilities and cosine similarity are normalizations of the dot product, and have also been widely used. Our focus was on these similarity measures. When researchers are considering any of the aforementioned measures, we point out benefits and drawbacks of using them in the context of random projections.
>
> 4. Researchers have experimented with random projections as embeddings, which have been presented as competent approach for approximating node similarity based on guarantees of the JL Lemma. The scope of this paper was to point what challenges might appear using this approach. We focused on the quality with respect to the degree distribution. Other methods for producing embeddings might have different types of challenges with respect to different structures. We hope to have addressed the point raised by the reviewer and will gladly come back to provide further clarifications if needed.

---

> > ### Comment · Reviewer_Nudr · 2024-11-23
> > **Appreciate the detailed comments but there is still a gap between theorems and the conclusions**
> >
> > Thank you for the detailed responses. As there seems to be some misunderstanding in our communications, I would like to restate my concerns, responsing to your rebuttal comments. Let me know if you find my understanding in the following responses, to make sure I evaluate your manuscripts correctly.
> >
> > **About 1**: I appreciate your response. Yes, I encourage the Authors to refer to the formal form of the JL lemma.
> >
> > **About 2**: I appreciate your detailed explanation. This is what I wanted. I encourage the authors to summarise the discussion and emphasize the essence of the difference between the previous usage of the JL lemma and your usage. Readers would not easily see your novel insight since both previous work and your work use the JL lemma.
> >
> > **About 3**:
> > Thank you for your comments. However, the comments did not solve my concerns. I still feel the current manuscript's conclusion section statements, e.g., "we propose that practitioners of random projection methods first use the embeddings...," are no longer in the scope of the Authors' research. As the Authors agreed, some have used the dot product and others have used the cosine similarity for different application-side reasons. Whether or not random projection can give a good estimate is an important property, but this does not mean one should be replaced by the other, because they measure different things. Therefore, we would not be able to say that "we propose that practitioners of random projection methods first use the embeddings..." as the Authors do in the Conclusion section. The authors' research would not be capable of conclusions other than that "different similarity metrics have different estimation difficulties", and this version would still be an acceptable and publishable-level conclusion to the community.
> >
> > Taking the dot product and cosine similarity as examples, the cosine similarity loses the information about the norm of the vectors, compared to the dot product. In applications where we are not interested in the norm information, we might choose the cosine similarity since it is estimated easily, but in applications where we are interested in the norm information, we cannot avoid the dot product even when it is difficult to estimate.
> >
> > **About 4**:
> > Thank you for your comments. However, it seems that we miscommunicated here. My concern was that the authors' current discussion did not separately discuss the intrinsic difficulty due to each similarity metric's fluctuation itself and the difficulty due to the random projection.
> >
> > If the difference in the error magnitudes of estimating the dot product similarity and estimating the cosine similarity is due to the difference in the magnitude of the fluctuations of the respective similarities themselves, we would say it is unclear why the authors limit their discussion to estimation from random projections. No matter what estimation methods we select, it is obvious that estimation from variables with large fluctuations is more difficult than estimation from variables with small fluctuations.
> > In such cases, it is natural to discuss why differences in the magnitude of fluctuations of the original vectors arise, rather than in the context of random projections.
> > Of course, the Authors might argue that there may be situations in which one would still wish to discuss estimation from random projections in such cases, but if so, the need for that discussion would need to be justified in the Introduction. However, there is no such justification in the manuscript. This is why the current manuscript is unacceptable.
> >
> > On the other hand, if you can show that the fluctuations of the dot product similarity and cosine similarity themselves are comparable but the difficulty of estimation from random projection is different, then the author's direction, i.e., to focus the discussion on random projection, would be convincing.
> > Hence, if there is a result that the intrinsic fluctuations of the dot product similarity and the cosine similarity themselves are comparable, it would be appreciated if you could show it.
> >
> > Just to be sure, my original review comments did NOT mean the Authors need to consider other metrics or other specific estimation methods. The examples, e.g., rel=1 (constant), are just to show the importance of separately discussing the difficulty owing to the fluctuation of the similarity metrics and the difficulty owing to the estimation methods' issues.

---

> ### Author Response · Authors · 2024-11-25
> **We acknowledge the reviewer's point and suggestion for point 3**
>
> 3. We appreciate the reviewer's point of view in limiting the extent of the recommendations and agree with the proposed correction. Our previous conclusions were primarily focused on the different characteristics of cosine and dot product with respect to their estimation noise under random projections. In that direction, we showed that a fundamental property of relative ordering under different similarities and normalizations can be significantly affected by random projections (with cosine similarity showing a more stable behavior). When considering the broader context of applications, we agree that this may not be the only factor to be examined towards the adoption of a particular similarity metric (e.g., dot product vs. cosine). As the reviewer appropriately pointed out, if a certain application relies on specific norm information, then cosine similarity may not be applicable, and dot product may be required -- even if more difficult to estimate. In that respect, we hope that our characterization of the different estimation behavior under RP of these two similarities (including the identification of potential pathologies for nodes of low and high degrees) is a contribution towards the understanding of the trade-offs imposed. We will reformulate our conclusion to limit the extent of the recommendations given and thank the reviewer for emphasizing this point.

---

> > ### Author Response · Authors · 2024-11-25
> > **We provide further analyses for point 4**
> >
> > 4. Thank you for the detailed elaboration of your concern. When separately discussing the intrinsic difficulty due to each similarity metric's fluctuation itself and the difficulty, the analysis in our paper corresponds to the second alternative that you pointed out. That is, the fluctuations of the dot product similarity and cosine similarity themselves are comparable but the difficulty of estimation from random projection is different. That can be shown by the following discussion about the distribution of the values of $rel_{uv}$.
> >
> > In **Section 3.1**, for dot product with $P=T$, we can look at three cases for the nodes $u$ and $v$:
> >
> > - *(a)* *No common neighbor* ($n_{uv}=0$). In this case $rel_{uv}=0$.
> >
> > - *(b)* They have a common neighbor ($n_{uv}\geq 1$) and *both nodes are low degree*. Now we have $d_u,dv\leq c$, so that
> > $$ \frac{1}{c^2}\leq \frac{n_{uv}}{d_ud_v} =rel_{uv}\leq 1.$$
> >
> > - *(c)* They have a common neighbor ($n_{uv}\geq 1$) and *one node is high degree node*. Let $u\in H_c$, by definition $d_u\geq \gamma^2 cq$. Therefore, using Proposition 3.2
> > $$\frac{1}{d_{max}^2}\leq \frac{n_{uv}}{d_ud_v}=rel_{uv}\leq \frac{\gamma}{d_u}\leq \frac{1}{\gamma c q}$$
> > where $d_{max} =\max_{h\in V} d_h$.
> >
> > Note that $rel_{uv}$ will have most of the values $0$, then since most of the nodes are of low degree the interval $[\frac{1}{d_{max}^2},\frac{1}{\gamma c q}]$ and similarities involving high degree node will be in $[\frac{1}{c^2},1].$ Note that since $\gamma \geq 1$ and $q\gg c$ the intervals are all disjoint.
> >
> > The estimation $rel^R_{uv}$ variance does not depend on $rel_{uv}$ only, but also on $rel_{uu}$ and $rel_{vv}$ that can be significantly larger values (Theorem 2.2):
> >
> > $$rel_{uv}^R\stackrel{a}{\sim} \mathcal{N}\left(rel_{uv}, \frac{1}{q}[rel_{uu}rel_{vv}+rel_{uv}^2]\right).$$
> >
> > Also note that, if $rel_{uv}=rel_{wh}$, the estimate $rel_{wh}^R$ can have a dramatically different variance depending on the values of $rel_{ww}$ and $rel_{hh}$.
> >
> > In **Section 3.2**, we can also expand it to include the following observations (on top of those already stated in Proposition 3.6). For *cosine similarity* and $P\in\{A,T\}$, we can look at three cases for the nodes $u$ and $v$:
> >
> > - *(a)* Nodes $u$ and $v$ have no common neighbor. Then, $rel_{uv}=0$.
> >
> > - *(b)* They have a common neighbor ($n_{uv}\geq 1$) and *both nodes are low degree*.
> > Since $d_u,d_v\leq c$ and $n_{uu}\leq d_u^2$, we have:
> > $$\frac{1}{c^2}\leq \frac{1}{d_ud_v}\leq rel_{uv}=\frac{n_{uv}}{\sqrt{n_{uu}n_{vv}}}\leq 1.$$
> >
> > - *(c)* They have a common neighbor ($n_{uv}\geq 1$) and *one node is of low degree and another is of high degree*. Let $u\in H_c$ and $v\in L_c$. Then
> > $$\frac{1}{d_{max}^2}\leq rel_{uv}=\frac{n_{uv}}{\sqrt{n_{uu}n_{vv}}} \stackrel{(2),(4)}{\leq} \frac{\gamma d_v}{\sqrt{d_ud_v}}=\frac{\gamma \sqrt{d_v}}{\sqrt{d_u}}\leq \frac{\gamma \sqrt{c}}{\sqrt{\gamma^2 c q}}=\frac{1}{\sqrt{q}},$$
> > where (2) corresponds to $d_u \leq n_{uu}$ and (4) to $n_{uv}\leq \gamma d_v$.
> >
> >
> > Most values of $rel_{uv}$ will be $0$; however we see that many will be in the interval $\left[\frac{1}{c^2},1\right]$, and there will be some with lower values in the interval $\left[\frac{1}{d_{max}^2},\frac{1}{\sqrt{q}}\right]$.
> >
> > Note that in this case variance of the estimate only depends on $rel_{uv}$, since we have
> > $$rel^R_{uv}\stackrel{a}{\sim}\mathcal{N}\left(rel_{uv}, \frac{(1-rel_{uv}^2)^2}{q}\right).$$
> >
> > If $rel_{uv}=rel_{wh}$, the variance of the estimates will be the same. The closer the relevance $rel_{uv}$ is to 1, the more precise the estimate $rel_{uv}^R$ is. The variance of $rel_{uv}^R$ is the highest in the case $rel_{uv}$ is $0$.
> >
> > As shown above, the distribution of relevance values for from 3.1.(*a-c*) (dot product) is comparable to the ones for 3.2.(*a-c*) (cosine), and the difference in the estimation of their values can be largely attributed to their different behavior with respect to random projections as described in the paper.

---

> > > ### Comment · Reviewer_Nudr · 2024-11-26
> > > **Appreciate your detailed concrete examples!**
> > >
> > > I thank the Authors for the detailed response. In particular, the specific examples to show the difference between the fluctuations of the dot product with P=T and cosine similarity are quite helpful.
> > >
> > > Verifying the Authors' comments needs time as it does not show the fluctuation's variance but instead intervals. However, if I can confirm the claim is reasonable during the reviewer discussion period, I will raise the score accordingly.

---

> > > > ### Author Response · Authors · 2024-12-01
> > > > **Let us know if any additional information or clarification can help**
> > > >
> > > > We appreciate the reviewer's efforts in verifying our claims. If there is any additional information or clarification that we can provide to assist you during the reviewer discussion period, please let us know.

---

### Official Review · Reviewer_RUkx · 2024-11-03

**Soundness:** 3
**Presentation:** 4
**Contribution:** 4
**Rating:** 8
**Confidence:** 3

**Summary:**

The paper focuses on the practice of using random projections (RP) to learn graph embeddings. In particular, they motivate the advantage of using Cosine Similarity over Euclidean Distance (which is widely popular), grounded in theoretical evidence. The authors show that a nodes with low/high degree enjoy weaker guarantees under Euclidean-based RP. They further extend their scope to the ranking context, where they build upon the general setting results and conclude that Cosine Similarity-based RP yield more robust rankings.

**Strengths:**

The paper studies an important and (much to the author's credits) well-motivated problem: random projection (RP) for dimensionality reduction on large-scale graph representation learning. Given the prevalence of RP and JL Lemma under many graph representation learning methods today, I believe their analysis on a fundamental design choice (Euclidean vs. Cosine Similarity) will be of great influence to the community. In particular, the theoretical question of how node degree affects graph representation learning is one with clear motivation --- given the influence highly-connected nodes have in many real-world applications. As such, I believe both their theoretical results and techniques will be of interest to the community broadly. Overall, the paper is very clearly written and the narrative is centered.

**Weaknesses:**

For Figure 1, it may be helpful to the audience if the axis and legends are accompanied by a text description as opposed to just the (paper-defined) mathematical notations. For example, it would help clarity if the blue dots are clearly labeled "Cosine" and the y-axis states "node degree".

In the theoretical sections, the presentation of the theorems and their accompanied analysis is helpful for interpreting the theorems. However, I felt that the contrast between results in Section 2.1 and 2.2 can be made stronger (or a subsection itself). See Q2 in the questions section for an example.

For the experimental section, I feel that including simple experiments independent of the ranking application can better support the theoretical analysis in Section 2. For example, would it be possible to design a simple example (synthetic network) where the two different RPs generate substantially different results for a high-degree node? This could also be in the form of a written example.

**Questions:**

1. Can the authors clarify, in simple terms, what difference between the Euclidean and Cosine dissimilarity led to one generating substantially poorer guarantees for high-degree nodes --- based on the theorems?

2. What is the approximate three-sigma interval, and how is it relevant? I see that the "interval endpoints do not depend on the node degrees" in the cosine case, but it is unclear to me the relationship between this interval and the projected features/learned embeddings. I think it will be helpful to not only highlight this distinction (as the authors have by italicizing), but also motivate it in the broader context clearly.

---

> ### Author Response · Authors · 2024-11-25
> **Response to Reviewer RUkx**
>
> We would like to thank the reviewer appreciating our paper and providing valuable feedback.
>
> Weaknesses:
>
> 1. Thanks for pointing out how we can improve Figure 1. We will update its labels as you suggested.
>
> 2. Addressed in the Questions block below.
>
> 3. We will produce an example independent of the ranking application that illustrates the deviation of the estimate from the value with respect to the nodes degree/connectedness.
>
> Questions:
>
> 1. We can add some discussion as follows:
>
> For dot product, the variance or the estimate of relevance of a pair of nodes using random projections depends both on their relevance as well as their node degrees. This adds a side effect that is more pronounced for nodes of low and high degrees, depending on whether the adjacency or transition matrix is adopted, respectively.  On the other hand, with cosine similarity the variance of error does not depend on the node degrees.
>
> More specifically, we can express the above in terms of the node relevance values. In the case of the dot product (Theorem 2.2), both $rel_{uu}$ and $rel_{vv}$ play a role in addition to $rel_{uv}$:
>
> $$rel_{uv}^R\stackrel{a}{\sim} \mathcal{N}\left(rel_{uv}, \frac{1}{q}(rel_{uu}rel_{vv} + rel_{uv}^2)\right)$$
>
> while in the case of cosine similarity (Theorem 2.5), only $rel_{uv}$ matters:
> $$rel^R_{uv}\stackrel{a}{\sim}\mathcal{N}\left(rel_{uv}, \frac{(1-rel_{uv}^2)^2}{q}\right).$$
>
> 2. The $3\sigma$ interval (property) refers to the corresponding Gaussian confidence interval within which the random projections will generate an estimate with approximately $99.73$% confidence. We also expressed this interval in terms of the node degrees and relevance values for the case of $P=A$ (see text following Lemma B.3 in the Appendix). For the case of cosine similarity, the passage "endpoints do note depend on the node degrees" was not precise, and we will correct it to state that the interval end points only depend on the cosine similarity being estimated. We can further highlight this difference in Sections 2.1 and 2.2, especially on their degree dependence. Such a distinction is also motivated and explained by the discussion provided for Q2 above.

---

> > ### Comment · Reviewer_RUkx · 2024-11-26
> >
> > Thank you to the authors for addressing my concerns.
> >
> > I feel that the pitched discussion in text: "For dot product, the variance... depends on both their relevance as well as their node degrees.. adds a side effect that is more pronounced.." is relatively imprecise (or wordy). However, I am very supportive of including the distribution of the node relevance values, since they capture the differences concisely (in particular, your remark that "only $rel_\mathrm{uv}$ matters").
> >
> > Thank you for clarifying that the $3\sigma$ interval is in the context of Gaussian confidence intervals, that makes a lot of sense.

---

### Meta-Review · Area_Chair_xdzV · 2024-12-20

**Metareview:**

This paper studies random projection methods for graph embedding: in particular embeddings produced by AR or TR where R is a Johnson-Lindenstrauss random projection matrix and A and T are the graph adjacency matrices and transition matrices in particular. They argue that due to the high variance introduced by the graph degree distribution, it is more appropriate to compare these embeddings via cosine similarity (i.e. inner product after normalization) rather than via direct inner product. This is perhaps not surprising -- and it may be an interesting observation outside of the graph context in other settings where random projection is used on data points with highly varying norms. The reviewers overall felt that the paper was well presented and that the theoretical results were interesting and valuable.

As the AC, I feel that a major weakness is that the paper only considers embeddings of the form AR and TR which are very far from what is employed in practice. To really capture graph structure, higher degree polynomials in these matrices (possibly with other modifications such as entrywise nonlinearities) are needed. It does not feel that the techniques of the paper would extend well to these more realistic settings. Thus, I feel that the paper gives a in-depth study of a very niche topic which in reality is far from the practice of graph embedding. Since its theoretical results follow fairly standard approaches, to me this makes the paper below bar for acceptance. However, after discussing with reviewers, they did not feel that this was a major weakness and still supported acceptance -- hence the recommendation.

**Additional Comments On Reviewer Discussion:**

NA. The rebuttal was useful in clarifying some points raised by the reviewers but nothing changed substantially.

---

### Decision · Program_Chairs · 2025-01-22

Accept (Poster)